# Improving deep neural network generalization and robustness to background bias via layer-wise relevance propagation optimization

Pedro R. A. S. Bassi [1,2] ✉, Sergio S. J. Dertkigil[3] & Andrea Cavalli[1,4] ✉

Features in images' backgrounds can spuriously correlate with the images' classes, representing background bias. They can influence the classifier's decisions, causing shortcut learning (Clever Hans effect). The phenomenon generates deep neural networks (DNNs) that perform well on standard evaluation datasets but generalize poorly to real-world data. Layer-wise Relevance Propagation (LRP) explains DNNs' decisions. Here, we show that the optimization of LRP heatmaps can minimize the background bias influence on deep classifiers, hindering shortcut learning. By not increasing run-time computational cost, the approach is light and fast. Furthermore, it applies to virtually any classification architecture. After injecting synthetic bias in images' backgrounds, we compared our approach (dubbed ISNet) to eight state-of-the-art DNNs, quantitatively demonstrating its superior robustness to background bias. Mixed datasets are common for COVID-19 and tuberculosis classification with chest X-rays, fostering background bias. By focusing on the lungs, the ISNet reduced shortcut learning. Thus, its generalization performance on external (out-of-distribution) test databases significantly surpassed all implemented benchmark models.

Deep neural networks (DNNs) revolutionized image classification. Counting on millions of trainable parameters, the models proved capable of analyzing entire images, becoming a new standard in many different fields. However, the features contained in images' backgrounds may have a strong and detrimental effect on the classification process. Background features can unintentionally correlate with the images' classes, thus representing background bias, also called spurious correlations. Trained in such environments, a DNN can learn to base its decisions not only on relevant image regions, but also on background features. The influence of background bias over the classifier deteriorates its capacity to analyze the images' relevant features and reduces the trustworthiness of its decisions. The biased model will perform artificially well on the training dataset, and on evaluation databases that contain the same background biases. This is a common condition for test datasets that are independent and identically distributed (i.i.d.) in relation to the training data (e.g., evaluation databases created by randomly slipping a dataset into a training and a testing partition). Nevertheless, the DNN will not be adequate for a practical application, which may not portray the same background biases. This scenario favors shortcut learning (or Clever Hans effect)[1], a condition where deep neural networks learn decision rules that perform well on standard benchmarks, but poorly on real-world applications. Unlike an overfitted DNN, with shortcut learning the model will perform well on an i.i.d. test dataset, but it will fail to

[1]Alma Mater Studiorum - University of Bologna, Bologna, Italy. [2]Center for Biomolecular Nanotechnologies, Istituto Italiano di Tecnologia, 73010 Arnesano (LE), Italy. [3]School of Medical Sciences, University of Campinas (UNICAMP), Campinas (SP), Brazil. [4]Istituto Italiano di Tecnologia, 16163 Genova (GE), Italy. ✉e-mail: pedro.salvadorbassi2@unibo.it; andrea.cavalli@unibo.it

generalize and be accurate on out-of-distribution (o.o.d.) databases. Furthermore, the image features that cause shortcut learning can be difficult for a person to identify[1]. This study presents a DNN architecture and training strategy to more efficiently deal with one of the main causes of shortcut learning: background bias.

Layer-wise relevance propagation (LRP)[2] is an explanation technique designed to create heatmaps for deep classifiers. LRP heatmaps are graphics that explain the model's behavior by making explicit how each part of an input image influenced the DNN output. We can create a heatmap for an input image, explaining the classifier score for a given class. Positive and negative heatmap values (dubbed relevance) indicate image areas that increased or decreased the classifier's confidence in the class, respectively. Areas with relevance closer to zero were less important for the classifier's decision. Thus, high relevance magnitude in the image background indicates strong background bias influence over the classifier's output.

This study proposes producing differentiable LRP heatmaps during training and optimizing them with a loss function. The function, named heatmap loss, employs ground-truth semantic segmentation masks to identify heatmap regions that correspond to the input image's background, and minimizes the LRP relevance magnitude in these regions. We called the minimization of the heatmap loss, alongside a classification loss, Background Relevance Minimization (BRM). BRM hinders classifier's decision rules based on background features. Since the training technique uses segmentation masks and LRP heatmaps to teach the classifier which image regions it must focus on, it can be regarded as an explanation- and segmentation-based spatial attention mechanism. A DNN trained with BRM can implicitly distinguish the input image's foreground and background features (i.e., implicitly segmenting a figure), and make decisions based only on the foreground. Accordingly, we named a classifier trained with BRM an Implicit Segmentation Neural Network (ISNet).

By minimizing the influence of background features over the DNN's outputs, the ISNet hinders the shortcut learning caused by background bias, improving out-of-distribution (o.o.d.) generalization. Despite its name, the ISNet is a classifier, not a segmenter. It does not produce segmentation outputs. After the training procedure, the ground-truth foreground masks and the creation of LRP heatmaps are no longer needed. Thus, the ISNet introduces no computational cost at run-time. We use BRM to train a classifier, defined by a backbone architecture, and the resulting ISNet has the same structure as its backbone. The ISNet's run-time efficiency is important for deploying DNNs in mobile or less powerful devices. Furthermore, the ISNet is versatile, because virtually any classification architecture can be used as its backbone.

A segmentation-classification pipeline is a standard strategy to avoid the influence of background bias over a classifier's decisions. It is a sequence of two networks. The first DNN segments the image's foreground. Afterwards, its output is used to erase the background, creating a segmented image that the second DNN classifies. Running two large DNNs sequentially leads to heavy memory and time requirements for the pipeline. We consider it as a benchmark in this study. Moreover, we also compare the ISNet to DNNs optimizing alternative explanation heatmaps. Hierarchical Attention Mining[3] (HAM, in Supplementary Note 7.2) and Guided Attention Inference Network[4] (GAIN) optimize Grad-CAM[5], Right for the Right Reasons[6] (RRR) optimizes input gradients[7], and the ISNet Grad*Input is an ablation experiment where we substitute the ISNet's LRP heatmaps by Gradient*Input[8] explanations. We also contrast the ISNet to a standard deep classifier, and to other state-of-the-art DNN architectures designed to minimize shortcut learning or control classifier attention. They are: Attention Gated Networks[9] (AG-Sononet) and the Vision Transformer[10], exemplifying attention mechanisms that do not learn from semantic segmentation masks; and a multi-task U-Net, which simultaneously produces classification scores and semantic segmentation outputs.

We present multiple classification experiments, designed to assess the ISNet's capacity of hindering the shortcut learning caused by background bias. Currently, the most popular open databases of COVID-19 chest X-rays contain either no or few COVID-19 negative/control cases[11,12]. Due to this limitation, to this date most studies resorted to mixed source datasets to train DNNs to differentiate between COVID-19 patients, healthy people, and patients with other diseases (e.g., non-COVID-19 pneumonia). In these datasets, different classes come from different databases assembled in distinct hospitals and cities. The dissimilar sources may contain different biases, which may help DNNs to classify the images according to their source dataset, rather than the disease symptoms. One study[13] accurately determined the source datasets after removing all lung regions from the images, proving the presence of background bias. A review[14] concluded that, if models are allowed to analyze the whole X-ray or a rectangular bounding box around the lungs, they tend to strongly focus on areas outside of the lungs. Thus, they fail to generalize or achieve satisfactory performance on external, o.o.d. datasets, with dissimilar sources in relation to the training images. Moreover, the review identified that the problem is a cause of shortcut learning[1]. A paper[15] utilized external datasets to evaluate traditional COVID-19 detection DNNs, whose reported results had been highly positive, and it demonstrated a strong drop in their performances. For these reasons, researchers have resorted to testing on o.o.d. datasets to understand the bias and generalization capability of DNNs trained on mixed databases. The approach shows reduced and more realistic performances in relation to the standard (i.i.d.) evaluation[14,16,17]. A common conclusion of these works is that using a segmentation-classification pipeline (segmenting the lungs before classification) improves generalization capability, reducing the bias induced by mixed training datasets[16,17]. The ISNet shall be able to focus only on the lungs, and we consider that the task of COVID-19 detection using the usual mixed datasets of chest X-rays will be useful to demonstrate its benefits.

We trained the ISNet on a mixed dataset based on one of the largest open COVID-19 chest X-ray databases[12], with the objective of distinguishing COVID-19 positive cases, normal images, and non-COVID-19 pneumonia. The two diseases have similar symptoms, making their differentiation non-trivial, and both produce signs in chest X-rays. Examples of COVID-19 signs are bilateral radiographic abnormalities, ground-glass opacity, and interstitial abnormalities[18]. Examples of pneumonia signals are poorly defined small centrilobular nodules, airspace consolidation, and bilateral asymmetric ground-glass opacity[19]. We evaluate the optimized DNN in a cross-dataset approach, using images collected from external locations in relation to the training samples. Evaluation with an o.o.d. dataset[14] shall allow us to assess whether the ISNet can reliably ignore the image background, reducing shortcut learning and increasing generalization performance.

The dataset mixing issue is not exclusive to COVID-19 detection. Instead, the technique is necessary whenever researchers need to use a classification database that does not contain all the required classes. Sometimes, such databases are the largest ones, a desirable quality for deep learning. This is the case in COVID-19 detection, and we find another example in tuberculosis (TB) detection. To the best of our knowledge, the tuberculosis X-ray dataset from TB Portals[20] is the open-source database with the largest number of tuberculosis-positive X-rays to date. However, the dataset is composed of TB-positive cases only. Thus, dataset mixing is required to use the TB Portals data for training tuberculosis detection DNNs. Moreover, a recent review[21] showed that many studies that classify tuberculosis with neural networks use dataset mixing. Furthermore, very few works evaluate DNN generalization to o.o.d. test datasets[21], and a study suggested that TB classification performance may strongly drop when DNNs are tested with datasets from external sources (o.o.d.)[22]. Another paper exposed

strong DNN attention outside of the lungs in TB-detection with mixed databases and advised the utilization of lung segmentation before classification[23]. Although the World Health Organization states that chest radiography is essential for the early detection of TB, they provide no recommendation on the use of computer aided detection as of 2016[24]. The reasons for this decision were the small number of studies, methodological limitations, and limited generalizability of the findings[24]. Consequently, we hypothesize that the tuberculosis detection task is prone to producing background bias and shortcut learning, especially when mixed datasets are employed. For this reason, we included the application in this study. We classify X-rays as normal or tuberculosis-positive. Our training dataset mixes the TB Portals database[20] with healthy X-rays from the CheXPert dataset[25]. To analyze the extent of shortcut learning, we evaluate the DNNs on an i.i.d. test dataset, and on an o.o.d. database[26]. We utilize the TB detection task to demonstrate that background bias and shortcut learning are not exclusive to COVID-19 detection, and to assess the ISNet's capacity of addressing the problem in diverse unfavorable scenarios.

Before showing the aforementioned applications, our "Results" section begins with a set of experiments using artificial bias. They consist of training the neural networks in natural and medical image datasets containing synthetic background bias, defined as a geometrical shape whose format (square, triangle or circle) is correlated with the image's classification label. The strong artificial background bias attracts the attention of standard classifiers, making them lose focus on the image's true region of interest and causing evident shortcut learning. The artificial bias is controllable, allowing the creation of diverse evaluation scenarios. We employ them to quantitatively compare the different neural networks' capacity of hindering the shortcut learning induced by background bias. Moreover, LRP heatmaps should be able to clearly show attention to the geometrical shapes in the models affected by shortcut learning. We experimented with adding synthetic background bias to 3 diverse databases. The first is the aforementioned COVID-19 dataset, also used in experiments without synthetic bias. The second is a facial attribute estimation dataset, extracted from CelebA[27]. Since a study[27] indicated that the classification of more attributes causes the classifiers to naturally focus more on the faces, we opted to classify only 3 facial attributes (rosy cheeks, high cheekbones, and smiling), increasing the difficulty of learning an adequate attention profile. The third database is a subset of the Stanford Dogs[28] dog breed classification dataset, comprising the following breeds: Pekingese, Tibetan Mastiff, and Pug. Stanford Dogs contains bounding-boxes, but no ground-truth dog segmentation target[28] (necessary for ISNet training). Thus, we converted its bounding-boxes to foreground masks, employing a pretrained general purpose semantic segmenter, DeepMAC[29]. Upon visual inspection, the quality of the masks was high. The three datasets allow us to assess the ISNet's background bias resistance in very diverse scenarios.

Supplementary Note 5 explains all datasets and their limitations. Supplementary Note 6 displays implementation details for the diverse tasks. Moreover, Supplementary Note 7 introduces two additional applications, CheXPert[25] (a large single-source X-ray database displaying multiple conditions) classification, and the facial attribute estimation task without synthetic bias. Both datasets do not present strong background bias. Thus, the experiments better delimit the ISNet use-case scenario.

The ISNet PyTorch code is available at https://github.com/PedroRASB/ISNet[30]. Summarizing, in this study, we:

1. Proposed a classifier architecture (ISNet) that, without a segmenter at run-time, reliably ignored images' backgrounds. In relation to the eight implemented state-of-the-art DNNs, the ISNet displayed superior capacity of hindering the shortcut learning caused by background bias, improving generalization. We justified this empirical result (section "Results") with an in-depth

theoretical comparison between the DNNs (Supplementary Note 2). Moreover, the ISNet is flexible (accepting virtually any classifier backbone) and introduces no extra computational cost at run-time.

2. Proposed the optimization of LRP heatmaps to improve and control DNNs, introducing the concept of background relevance minimization. The technique produced a theoretically founded (section "ISNet theoretical fundamentals") explanation-based spatial attention mechanism that learns from foreground segmentation masks.

3. Presented an efficient and automatically differentiable PyTorch implementation of LRP.

## Results
### Synthetic background bias
In these experiments, training images contained synthetic background bias designed to cause shortcut learning. We tested the neural networks on 3 datasets: a biased set, which contains the same background geometrical shapes found in training; the standard set, with no synthetic bias; and a deceiving bias dataset, which has geometrical shapes, but the correlation between them and the classification labels deceivingly change (e.g., a circle that was correlated with class 1 in training will be associated to class 2 during testing). Supplementary Note 5 provides more details. A comparison of a DNN's performance on the three testing scenarios provides a quantitative assessment of shortcut learning. In relation to the biased test performance, the higher the influence of the geometrical shapes on the classifier's outputs, the larger the F1-Score reduction when they are removed from the evaluation dataset (standard test) or substituted by deceiving bias (deceiving bias test). Table 1 presents the test average F1-Scores for all DNNs in the three testing environments.

Data augmentation in COVID-19 detection and facial attribute estimation consisted of random rotations, translations, and flipping (Supplementary Note 6.1). Thus, the applications exemplify standard data augmentation practices. Meanwhile, we utilized no data augmentation in Stanford Dogs. As the augmentation procedures may totally or completely remove the synthetic background bias from the image, this choice makes the Stanford Dogs experiment a scenario of more extreme background bias. The ISNet is resistant to image flipping, rotations, and translations. These operations did not negatively affect the training procedure, nor did they worsen the validation error during preliminary tests with an augmented validation dataset.

The three synthetic bias applications represent very distinct scenarios, all of them considering high-resolution images (224x224) and deep classifiers. COVID-19 detection is a challenging and contemporary biomedical classification task, with images' foregrounds defined as the lungs. Stanford Dogs and CelebA are natural image RGB datasets, where the foregrounds are the dogs or faces. CelebA presents challenging in-the-wild pictures, where faces can appear in multiple poses[27]. Stanford Dogs represents a difficult fine-grained classification task, with large intra-class variance, low inter-class variance, high background variety, and possible occlusion[28]. Like the CelebA faces, dogs can appear in multiple poses and distances. Thus, in the two datasets, foregrounds strongly vary throughout the figures. Dataset sizes also substantially change, with 501 images in the dogs dataset, 13932 in COVID-19 detection, and 30000 for facial attribute estimation. As the three tasks represent very diverse scenarios, the DNNs' performances vary across the experiments. However, the distinct applications allow us to draw more reliable conclusions by analyzing repeating patterns in the experiments' results.

First, all datasets have a strong tendency to cause shortcut learning, as intended. This is proved by the large performance drops seen for the standard classifiers in Table 1. The baseline model represents a common DNN, without any mechanism to avoid background attention. In the most extreme case, Stanford Dogs, the model's

**Table 1 | Test macro-average F1-Scores for neural networks trained in datasets with synthetic background bias[a]**

| Model | Biased test maF1 | Standard test maF1 | Deceiving bias test maF1 |
|---|---|---|---|
| Stanford dogs with synthetic background bias | | | |
| ISNet | 0.548 ± 0.035 | 0.553 ± 0.035 | 0.548 ± 0.035 |
| ISNet Grad*Input | 0.55 ± 0.034 | 0.545 ± 0.034 | 0.545 ± 0.034 |
| Standard classifier | 0.926 ± 0.019 | 0.419 ± 0.034 | 0.071 ± 0.017 |
| Segmentation-classification pipeline | 0.519 ± 0.035 | 0.519 ± 0.035 | 0.518 ± 0.035 |
| Multi-task U-Net | 0.522 ± 0.036 | 0.455 ± 0.036 | 0.38 ± 0.035 |
| AG-Sononet | 0.956 ± 0.015 | 0.214 ± 0.027 | 0.019 ± 0.009 |
| Extended GAIN | 0.935 ± 0.017 | 0.445 ± 0.034 | 0.1 ± 0.019 |
| RRR | 0.851 ± 0.025 | 0.548 ± 0.034 | 0.299 ± 0.025 |
| Vision transformer (ViT-B/16) | 0.637 ± 0.034 | 0.419 ± 0.032 | 0.399 ± 0.032 |
| Standard classifier reference (trained without synthetic bias) | – | 0.556 ± 0.035 | – |
| COVID-19 detection with synthetic background bias | | | |
| ISNet | 0.775 ± 0.008 | 0.775 ± 0.008 | 0.775 ± 0.008 |
| ISNet Grad*Input | 0.542 ± 0.01 | 0.544 ± 0.01 | 0.417 ± 0.01 |
| Standard classifier | 0.775 ± 0.008 | 0.434 ± 0.01 | 0.195 ± 0.004 |
| Segmentation-classification pipeline | 0.618 ± 0.009 | 0.619 ± 0.009 | 0.618 ± 0.009 |
| Multi-task U-Net | 0.667 ± 0.01 | 0.341 ± 0.007 | 0.156 ± 0.004 |
| AG-Sononet | 0.943 ± 0.005 | 0.386 ± 0.008 | 0.047 ± 0.003 |
| Extended GAIN | 0.41 ± 0.009 | 0.306 ± 0.006 | 0.219 ± 0.003 |
| RRR | 0.464 ± 0.009 | 0.458 ± 0.008 | 0.426 ± 0.008 |
| Vision transformer (ViT-B/16) | 0.685 ± 0.009 | 0.496 ± 0.009 | 0.327 ± 0.009 |
| Standard classifier reference (trained without synthetic bias) | – | 0.546 ± 0.01 | – |
| Facial attribute estimation with synthetic background bias | | | |
| ISNet | 0.807 ± 0.027 | 0.807 ± 0.027 | 0.807 ± 0.027 |
| ISNet Grad*Input | 0.496 ± 0.02 | 0.499 ± 0.02 | 0.503 ± 0.021 |
| Standard classifier | 0.974 ± 0.012 | 0.641 ± 0.054 | 0.398 ± 0.019 |
| Segmentation-classification pipeline | 0.794 ± 0.031 | 0.794 ± 0.031 | 0.794 ± 0.031 |
| Multi-task U-Net | 0.985 ± 0.008 | 0.665 ± 0.129 | 0.351 ± 0.015 |
| AG-Sononet | 0.985 ± 0.009 | 0.616 ± 0.094 | 0.326 ± 0.016 |
| Extended GAIN | 0.886 ± 0.023 | 0.773 ± 0.034 | 0.633 ± 0.03 |
| RRR | 0.794 ± 0.024 | 0.77 ± 0.032 | 0.557 ± 0.025 |
| Vision transformer (ViT-B/16) | 0.675 ± 0.023 | 0.645 ± 0.03 | 0.531 ± 0.023 |
| Standard classifier reference (trained without synthetic bias) | – | 0.802 ± 0.028 | – |

[a]In the multi-class single-label experiments (Stanford Dogs and COVID-19 detection), scores are reported as mean and standard deviation. In facial attribute estimation (multi-label problem), they are displayed as mean and 95% confidence intervals. Supplementary Note 10 provides more details about the statistical analysis in this study.

macro-average F1-Score (maF1) falls from 0.926 ± 0.019 (with geometrical shapes in the test dataset) to 0.546 ± 0.034 (with no synthetic bias), and finally to 0.071 ± 0.017 (with deceiving background bias). Putting these results into perspective, from the 201 evaluation

samples, in the biased test the model correctly classifies 192, in the unbiased test, 89, and, in the deceiving test, 10.

Second, the segmentation-classification pipeline and the ISNet were the only models not affected by background bias in any of the three experiments in Table 1. They show no reduction in maF1 when the synthetic background bias was removed or substituted by deceiving bias. On the other hand, all other models display maF1 drop across the columns in Table 1. Even considering the interval estimates, none of the other benchmark DNNs have overlapping maF1 intervals in all three testing scenarios. Thus, the bias influence over these classifiers is evident. Conversely, the ISNet successfully minimized background attention and the consequent shortcut learning. Moreover, the ISNet's resistance to background bias is not accompanied by an accuracy drop: it has the highest average F1-Scores in the three diverse tasks (standard and deceiving bias tests in Table 1). It could even surpass the segmentation-classification pipeline, a much larger and slower model (Supplementary Note 9 presents a speed and size comparison).

As further proof of the ISNet capacity of avoiding background bias attention while retaining high accuracy, its test maF1 scores, when trained with the synthetically biased data, match or surpass a standard classifier trained in datasets without any synthetic background bias (Table 1). We did not find relevant natural background bias in Stanford Dogs or CelebA. Accordingly, in these two cases, the ISNet matched a standard classifier trained in an unbiased environment, indicating that the insertion of synthetic bias could not reduce its accuracy. Meanwhile, the ISNet also hindered the shortcut learning caused by the naturally occurring background bias in the COVID-19 dataset. This result explains why it strongly surpassed the standard classifier trained in the COVID-19 database without any synthetic bias. I.e., in this scenario the standard model suffers shortcut learning caused by non-synthetic background bias (resulting from dataset mixing), which the ISNet hinders (section "COVID-19 detection").

In Stanford Dogs, the standard classifier is a VGG-19[31]. This architecture is also used as the classification backbone for the ISNet, ISNet Grad*Input, segmentation-classification pipeline (defined as a U-Net[32] segmenter followed by VGG-19 classifier), extended GAIN, and RRR. In the two other datasets (and in the remaining experiments in this study), a DenseNet121[33] substitutes the VGG-19 as the standard classifier and the backbone for the aforementioned DNNs. The remaining benchmark networks (multi-task U-Net, AG-Sononet, and Vision Transformer) utilize their original architectures in all tasks (Supplementary Note 2.1). The DenseNet and VGG show how the diverse DNN architectures behave with different types of backbone. A Densely Connected Convolutional Network[33] (DenseNet121) exemplifies very deep (121 layers) architectures with skip connections, and a VGG-19[31] represents DNNs with fewer layers (19) and no skip connections. The VGG was chosen as the shallower model because it is an influential and popular architecture, with a simple yet effective design. DenseNets are very deep but efficient, carrying a small number of parameters in relation to their depth[33]. This characteristic resonates with the efficiency focus of the ISNet design. Another reason for the DenseNet121 choice is that the architecture is among the most popular for the classification of lung diseases in chest X-rays[34], a task that is considered in multiple experiments throughout this work. Evidencing the ISNet's versatility, it surpassed all benchmark DNNs and was resistant to shortcut learning with both backbones (Table 1).

## COVID-19 detection

For COVID-19 detection we employ an external (o.o.d.) test dataset, whose images come from distinct hospitals and cities in relation to the training database (Supplementary Note 5.1). Tables 2 and 3 show the DNNs' test performances.

The ISNet obtained the best o.o.d. test performance in Tables 2 and 3, surpassing all benchmark DNNs' average performance metrics.

Moreover, the ISNet results' 95% highest density intervals do not overlap with any other network for any average performance measurement. The ground-truth foreground masks used for training were automatically generated by a U-Net, trained for lung segmentation in another study[16]. Therefore, the performances achieved by the ISNet and by the alternative segmentation-classification pipeline could possibly increase even more, provided a dataset containing a large amount of chest X-rays accompanied by manually segmented lungs.

Tables 2 and 3 results may seem worse than other COVID-19 detection studies, which report remarkably high performances, strongly surpassing expert radiologists (e.g., F1-Scores close to 100%). However, evidence suggests that, currently, such results are obtained when the training and test datasets come from the same distribution/sources (i.i.d. datasets)[14]. Moreover, studies showed that these strong performances may be boosted by bias and shortcut learning, preventing the neural networks from generalizing, or achieving comparable results in the real-world[14,16,15]. Instead, the performances in Table 2 are comparable to other works that evaluate their DNNs in external (o.o.d.) databases[14,16,15]. For example, an article[15] reported AUC of 0.786 ± 0.025 on an external COVID-19 X-ray dataset, considering a DenseNet121 and no lung segmentation. Here, the DenseNet121 obtained 0.808 AUC, which falls into their reported confidence interval. Another paper[16] evaluates COVID-19 detection and utilizes lung segmentation before classification with a DenseNet201[33]. They achieved maF1 of 0.754, with 95% HDI of [0.687,0.82], evaluating on an external dataset. The ISNet maF1 95% HDI, [0.755,0.791], fits inside their reported 95% HDI. We must note that the aforementioned studies use different databases. Thus, caution is required when directly comparing the numerical results.

COVID-19 detection with mixed datasets is a task known for background bias and common shortcut learning, which results in subpar o.o.d. generalization[15,14]. Accordingly, the results in Tables 2 and 3 are consistent with our findings from the synthetic bias experiments. Firstly, the standard classifier (DenseNet121) displayed unimpressive generalization, achieving only 0.546 ± 0.01 o.o.d. maF1. Moreover, in Table 1, the ISNet and the alternative segmentation-classification pipeline consistently were the two models with the highest resistance to background bias attention and the best generalization. Accordingly, in Tables 2 and 3, the two models had superior o.o.d. maF1 in COVID-19 detection. Therefore, the results in the COVID-19 detection confirm that the task is prone to shortcut learning, which the ISNet and pipeline could better mitigate. However, the ISNet could surpass even the large pipeline, showing no HDI superposition with its results. We analyze this finding in Supplementary Note 2.2, where we also theoretically justify why the ISNet surpassed all other benchmark models.

To illustrate that it is not possible to simply train a model with segmented images, and then use it without segmentation at run-time, we tested the segmentation-classification pipeline after removing its segmenter (U-Net). Thus, we simulated a DenseNet121 trained on segmented images and used to classify unsegmented ones. As expected, this resulted in a dramatic performance drop: maF1 fell from 0.645 ± 0.009 to 0.217 ± 0.003 (changing its 95% HDI from [0.626,0.663] to [0.211,0.224]). Therefore, unlike the ISNet, the pipeline needs a segmenter at run-time. Table 4 displays the confusion matrices for all DNNs.

## Tuberculosis Detection

Table 5 reports performances for tuberculosis detection, using an external (o.o.d.) test dataset (Supplementary Note 5.2). On the i.i.d. test dataset all models had mean AUC over 0.9. Moreover, considering i.i.d. evaluation, the segmentation-classification pipeline achieved maF1 (with 95% confidence interval) of 0.955 ± 0.016, the ISNet 0.974 ± 0.012, the extended GAIN 0.982 ± 0.009, the Vision Transformer 0.926 ± 0.02, RRR 0.839 ± 0.028, and all other DNNs' mean maF1 scores surpassed 0.985. Our i.i.d. test results are in line with other studies that detected tuberculosis with DNNs, most of which report very high AUC and F1-Score[21]. We could not find studies employing a training dataset like ours and o.o.d. testing, as the evaluation methodology is rare in tuberculosis detection.

Like in COVID-19 detection and all experiments with synthetic bias, the standard classifier (DenseNet121) o.o.d. generalization was underwhelming in TB detection (0.566 ± 0.05 maF1), and the ISNet was the best performing model on the tuberculosis o.o.d. dataset.

**Table 2 | Test F1-Scores and ROC-AUC for the deep neural networks in COVID-19 detection (o.o.d. evaluation)[a]**

| Model and Metric | Normal | Pneumonia | COVID-19 | Mean (macro-average) |
|---|---|---|---|---|
| ISNet F1-Score | 0.555 ± 0.022, [0.512,0.597] | 0.858 ± 0.007, [0.844,0.871] | 0.907 ± 0.006, [0.896,0.918] | 0.773 ± 0.009, [0.755,0.791] |
| U-Net+DenseNet121 F1-Score | 0.571 ± 0.018, [0.535,0.607] | 0.586 ± 0.013, [0.561,0.611] | 0.776 ± 0.008, [0.76,0.792] | 0.645 ± 0.009, [0.626,0.663] |
| DenseNet121 F1-Score | 0.444 ± 0.02, [0.403,0.482] | 0.434 ± 0.015, [0.405,0.463] | 0.76 ± 0.008, [0.744,0.775] | 0.546 ± 0.01, [0.527,0.565] |
| Multi-task U-Net F1-Score | 0.419 ± 0.025, [0.369,0.469] | 0.119 ± 0.011, [0.098,0.14] | 0.585 ± 0.009, [0.566,0.602] | 0.374 ± 0.01, [0.355,0.394] |
| AG-Sononet F1-Score | 0.124 ± 0.015, [0.096,0.153] | 0.284 ± 0.015, [0.255,0.312] | 0.659 ± 0.009, [0.641,0.676] | 0.356 ± 0.008, [0.34,0.372] |
| Extended GAIN F1-Score | 0.203 ± 0.019, [0.166,0.24] | 0.485 ± 0.013, [0.46,0.511] | 0.711 ± 0.009, [0.693,0.728] | 0.466 ± 0.009, [0.449,0.485] |
| RRR F1-Score | 0.36 ± 0.018, [0.325,0.394] | 0.552 ± 0.013, [0.526,0.577] | 0.737 ± 0.009, [0.72,0.755] | 0.55 ± 0.009, [0.532,0.568] |
| Vision Transformer (ViT-B/16) F1-Score | 0.382 ± 0.017, [0.348,0.415] | 0.474 ± 0.013, [0.448,0.499] | 0.525 ± 0.011, [0.503,0.548] | 0.46 ± 0.009, [0.443,0.478] |
| ISNet AUC | 0.931 ± 0.01 | 0.962 ± 0.006 | 0.976 ± 0.005 | 0.952 |
| U-Net+DenseNet121 AUC | 0.888 ± 0.019 | 0.78 ± 0.016 | 0.846 ± 0.013 | 0.833 |
| DenseNet121 AUC | 0.804 ± 0.023 | 0.805 ± 0.015 | 0.86 ± 0.013 | 0.808 |
| Multi-task U-Net AUC | 0.721 ± 0.034 | 0.412 ± 0.019 | 0.487 ± 0.02 | 0.553 |
| AG-Sononet AUC | 0.451 ± 0.028 | 0.681 ± 0.019 | 0.658 ± 0.018 | 0.591 |
| Extended GAIN AUC | 0.7 ± 0.025 | 0.756 ± 0.016 | 0.806 ± 0.016 | 0.724 |
| RRR AUC | 0.782 ± 0.02 | 0.736 ± 0.017 | 0.835 ± 0.014 | 0.775 |
| Vision Transformer (ViT-B/16) AUC | 0.755 ± 0.032 | 0.645 ± 0.019 | 0.619 ± 0.019 | 0.683 |

[a]Class ROC-AUC scores are calculated with a one-versus-rest approach and accompanied by 95% confidence intervals. Mean AUC is provided as point estimates and we calculate it with a pairwise technique[48], instead of averaging the class scores. Other metrics are reported as: mean ± std, [95% HDI]. Both mean and standard deviation (std) are extracted from the metric's probability distribution, according to Bayesian estimation. 95% HDI indicates the 95% highest density interval, an interval containing 95% of the metric's probability mass. Furthermore, any point inside the interval has a probability density that is higher than that of any point outside the interval. Supplementary Note 10 explains the statistical methods in detail.

**Table 3 | Test precision, recall and specificity for the deep neural networks in COVID-19 detection (o.o.d. evaluation)[a]**

| Model and Metric | Normal | Pneumonia | COVID-19 | Mean (macro-average) |
|---|---|---|---|---|
| ISNet precision | 0.544 ± 0.026, [0.494,0.594] | 0.794 ± 0.01, [0.774,0.814] | 0.993 ± 0.002, [0.988,0.997] | 0.777 ± 0.009, [0.759,0.795] |
| U-Net+DenseNet121 precision | 0.446 ± 0.019, [0.408,0.483] | 0.791 ± 0.015, [0.763,0.82] | 0.723 ± 0.011, [0.702,0.744] | 0.653 ± 0.009, [0.636,0.67] |
| DenseNet121 precision | 0.364 ± 0.02, [0.324,0.402] | 0.827 ± 0.018, [0.792,0.861] | 0.649 ± 0.01, [0.629,0.67] | 0.614 ± 0.009, [0.594,0.631] |
| Multi-task U-Net precision | 0.552 ± 0.033, [0.488,0.617] | 0.232 ± 0.02, [0.194,0.272] | 0.469 ± 0.01, [0.449,0.489] | 0.418 ± 0.013, [0.392,0.444] |
| AG-Sononet precision | 0.104 ± 0.013, [0.079,0.129] | 0.665 ± 0.025, [0.616,0.715] | 0.549 ± 0.01, [0.528,0.569] | 0.439 ± 0.01, [0.419,0.459] |
| Extended GAIN precision | 0.189 ± 0.019, [0.152,0.225] | 0.603 ± 0.016, [0.571,0.636] | 0.642 ± 0.011, [0.62,0.664] | 0.478 ± 0.009, [0.461,0.496] |
| RRR precision | 0.262 ± 0.015, [0.232,0.293] | 0.728 ± 0.016, [0.697,0.758] | 0.723 ± 0.011, [0.701,0.745] | 0.571 ± 0.008 [0.555,0.587] |
| Vision transformer (ViT-B/16) precision | 0.268 ± 0.015, [0.239,0.297] | 0.552 ± 0.016, [0.521,0.584] | 0.572 ± 0.014, [0.544,0.598] | 0.464 ± 0.009, [0.447,0.481] |
| ISNet recall | 0.566 ± 0.026, [0.515,0.616] | 0.933 ± 0.007, [0.919,0.946] | 0.835 ± 0.01, [0.816,0.853] | 0.778 ± 0.009, [0.76,0.796] |
| U-Net+DenseNet121 recall | 0.796 ± 0.021, [0.756,0.837] | 0.466 ± 0.014, [0.439,0.494] | 0.838 ± 0.009, [0.819,0.856] | 0.7 ± 0.009, [0.683,0.717] |
| DenseNet121 recall | 0.57 ± 0.026, [0.518,0.618] | 0.294 ± 0.013, [0.27,0.32] | 0.916 ± 0.007, [0.902,0.93] | 0.594 ± 0.01, [0.574,0.612] |
| Multi-task U-Net recall | 0.338 ± 0.024, [0.29,0.386] | 0.08 ± 0.008, [0.066,0.095] | 0.776 ± 0.011, [0.755,0.797] | 0.398 ± 0.009, [0.38,0.416] |
| AG-Sononet recall | 0.156 ± 0.019, [0.12,0.192] | 0.18 ± 0.011, [0.16,0.201] | 0.824 ± 0.01, [0.805,0.843] | 0.387 ± 0.008, [0.371,0.402] |
| Extended GAIN recall | 0.22 ± 0.021, [0.178,0.261] | 0.406 ± 0.014, [0.379,0.432] | 0.796 ± 0.01, [0.775,0.816] | 0.474 ± 0.009, [0.456,0.492] |
| RRR recall | 0.574 ± 0.025, [0.524,0.624] | 0.445 ± 0.014, [0.417,0.471] | 0.753 ± 0.011, [0.731,0.775] | 0.59 ± 0.01, [0.57,0.611] |
| Vision transformer (ViT-B/16) recall | 0.665 ± 0.024, [0.616,0.712] | 0.415 ± 0.014, [0.388,0.442] | 0.486 ± 0.013, [0.461,0.511] | 0.522 ± 0.01, [0.501,0.542] |
| ISNet specificity | 0.937 ± 0.005, [0.928,0.946] | 0.834 ± 0.009, [0.817,0.851] | 0.995 ± 0.002, [0.991,0.998] | 0.922 ± 0.003, [0.916,0.928] |
| U-Net+DenseNet121 specificity | 0.869 ± 0.006, [0.857,0.882] | 0.915 ± 0.006, [0.903,0.928] | 0.708 ± 0.011, [0.686,0.73] | 0.831 ± 0.004, [0.823,0.839] |
| DenseNet121 specificity | 0.869 ± 0.006, [0.856,0.881] | 0.958 ± 0.005, [0.949,0.967] | 0.549 ± 0.012, [0.525,0.573] | 0.792 ± 0.004, [0.784,0.8] |
| Multi-task U-Net specificity | 0.964 ± 0.004, [0.957,0.971] | 0.818 ± 0.009, [0.801,0.835] | 0.201 ± 0.01, [0.182,0.22] | 0.661 ± 0.004, [0.653,0.669] |
| AG-Sononet specificity | 0.823 ± 0.007, [0.808,0.837] | 0.938 ± 0.006, [0.927,0.948] | 0.384 ± 0.012, [0.361,0.408] | 0.715 ± 0.004, [0.707,0.722] |
| Extended GAIN specificity | 0.875 ± 0.006, [0.863,0.888] | 0.817 ± 0.009, [0.799,0.834] | 0.597 ± 0.012, [0.573,0.62] | 0.763 ± 0.004, [0.754,0.772] |
| RRR specificity | 0.787 ± 0.008, [0.772,0.802] | 0.886 ± 0.007, [0.872,0.9] | 0.737 ± 0.011, [0.716,0.758] | 0.803 ± 0.004, [0.795,0.812] |
| Vision transformer (ViT-B/16) specificity | 0.761 ± 0.008, [0.745,0.776] | 0.769 ± 0.01, [0.75,0.788] | 0.669 ± 0.012, [0.646,0.692] | 0.733 ± 0.005, [0.723,0.742] |

[a]Metrics are reported as: mean ± std, [95% HDI], according to Bayesian estimation. Supplementary Note 10 provides more details about the statistical analysis.

Furthermore, the proposed model showed no confidence interval overlap with the other DNNs' AUCs. Here, the segmentation-classification pipeline results were not as promising as in COVID-19 detection. Supplementary Note 2.3 analyses this finding. The ISNet performance on the i.i.d. evaluation database is among the lowest, while it was the best performing model on the o.o.d. dataset. Shortcut learning is characterized by high accuracy on standard benchmarks (i.i.d. datasets), but impaired o.o.d. generalization, and poor real-world performance[i]. Thus, the ISNet quantitative results are coherent with a reduction in shortcut learning, without overall accuracy degradation. Therefore, TB classification is an additional example of a real-world application that can be heavily affected by background bias and shortcut learning, representing another notable use-case for the ISNet.

## Heatmaps
Figure 1 presents heatmaps for the experiments with synthetic background bias. They show that only the ISNet and the segmentation-classification pipeline consistently and effectively minimized the influence of the background bias over the classifier. The results in Table 1 support the heatmaps, by quantitatively proving that the two models were the only ones never influenced by the artificial bias.

Figure 2 shows heatmaps for the COVID-19 and TB detection applications (without synthetic bias). The tasks use mixed training datasets, which are known to cause background bias and shortcut learning[35,13,15,14]. Accordingly, the LRP heatmaps for a standard classifier (DenseNet121) demonstrate a significant influence of background features over the classifier's decisions, indicating shortcut learning. Supporting this finding, Tables 2 and 5 show that the model's generalization performance was impaired. It achieved only 0.546 ± 0.01 and 0.566 ± 0.05 average F1-Scores in the COVID-19 and TB o.o.d. test

datasets, respectively. Conversely, the heatmaps in Fig. 2 indicate that the ISNet is the DNN with the least amount of background attention in the two tasks. Quantitatively supporting the information in the heatmaps, the ISNet's o.o.d. generalization performance surpassed all other models in TB and COVID-19 detection (Tables 2 and 5), indicating that it could better minimize the influence of background bias over the classification decisions.

Supplementary Note 2.3 thoroughly analyses the heatmaps in view of the quantitative results in Tables 1–5, and uses this investigation to compare the ISNet and the benchmark DNNs in more detail. Moreover, it also presents Grad-CAM explanations for the ISNet. Finally, Supplementary Note 3 compares the ISNet's LRP heatmaps to X-rays where a radiologist, who had no access to the DNN or the heatmaps, marked the lung diseases' lesions. The comparison demonstrates a correlation between the marked regions and the areas that influenced the ISNet's decisions. In summary, the ISNet diverted DNN focus from background bias to the lesions. The high-resolution marked X-rays and ISNet heatmaps are available and individually analyzed in Supplementary Data 1[36], and exemplified in Supplementary Fig. 2.

## Discussion
In three synthetic bias applications, considering diverse tasks, dataset sizes, and classifier backbones, we quantitatively demonstrated that the artificial background bias could not influence the ISNet's decisions. Therefore, the model hindered shortcut learning and improved generalization. The COVID-19 and Tuberculosis classification tasks exemplify realistic and contemporary scenarios where dataset mixing is commonly employed, frequently causing background bias, shortcut learning, and impaired o.o.d. generalization[35,13,15,14,23]. The two

**Table 4 | Test confusion matrices for the deep neural networks in COVID-19 detection (o.o.d. evaluation)**

| True class | Predicted class | | |
|---|---|---|---|
| | Normal | Pneumonia | COVID-19 |
| ISNet | | | |
| Normal | 210 | 157 | 3 |
| Pneumonia | 81 | 1210 | 4 |
| COVID-19 | 93 | 156 | 1266 |
| U-Net + DenseNet121 | | | |
| Normal | 296 | 9 | 65 |
| Pneumonia | 271 | 604 | 420 |
| COVID-19 | 95 | 149 | 1271 |
| DenseNet121 | | | |
| Normal | 211 | 16 | 143 |
| Pneumonia | 306 | 381 | 608 |
| COVID-19 | 63 | 62 | 1390 |
| RRR | | | |
| Normal | 213 | 84 | 73 |
| Pneumonia | 355 | 576 | 364 |
| COVID-19 | 243 | 130 | 1142 |
| Multi-task U-Net | | | |
| Normal | 125 | 42 | 203 |
| Pneumonia | 62 | 103 | 1130 |
| COVID-19 | 38 | 300 | 1177 |
| Attention Gated Sononet | | | |
| Normal | 57 | 2 | 311 |
| Pneumonia | 346 | 233 | 716 |
| COVID-19 | 151 | 114 | 1250 |
| Extended GAIN | | | |
| Normal | 81 | 145 | 144 |
| Pneumonia | 241 | 526 | 528 |
| COVID-19 | 108 | 200 | 1207 |
| Vision Transformer | | | |
| Normal | 247 | 50 | 73 |
| Pneumonia | 279 | 538 | 478 |
| COVID-19 | 393 | 385 | 737 |

applications displayed shortcut learning in this study, as indicated by the standard classifiers' (DenseNet121) unimpressive o.o.d. generalization and strong background attention. Consequently, being able to hinder background attention and shortcut learning, the ISNet was consistently the model with the best o.o.d. generalization in the two tasks, like in the experiments with synthetic background bias. Accordingly, the ISNet's LRP heatmaps indicate a minimization of the influence of background features on classifier's outputs.

In the applications with artificial bias and on COVID-19 detection, the segmentation-classification pipeline was surpassed only by the ISNet. However, the pipeline's generalization capacity was less promising in tuberculosis detection. The synthetic bias experiments quantitatively demonstrated that, besides the ISNet and the segmentation-classification pipeline, the remaining benchmark models could not effectively hinder the shortcut learning caused by background bias. Accordingly, their generalization capacity did not match the ISNet's, and their heatmaps showed significant background attention.

We justify these empirical findings with a theoretical analysis in Supplementary Note 2 and section "ISNet theoretical fundamentals". It explains the benchmark models' drawbacks, which the ISNet does not share. In summary, the segmentation-classification pipeline is robust to background bias, but it is computationally expensive even at run-time, and it may form decision rules based on images' foreground shape, possibly hindering generalization. In the multi-task DNN, foreground features can guide the creation of the segmentation output, while background features heavily influence the DNN's classification output. Therefore, the model can precisely segment the foreground, while background bias influences its classification scores. Attention mechanisms that do not learn from foreground segmentation masks (e.g., AG-Sononet[9] and Vision Transformer[10]) cannot reliably differentiate background and foreground features. Thus, they may not hinder background bias attention and shortcut learning. Input gradients and Gradient*Input explanations are much noisier than LRP for deep neural networks. Accordingly, considering deep backbones and high-resolution images, the ISNet can more effectively and stably minimize background attention in relation to DNNs optimizing input gradients (RRR[6]) and Gradient*Input (ISNet Grad*Input). Finally, Grad-CAM optimization (GAIN[4]) can produce spurious Grad-CAM heatmaps, which deceivingly display no background attention, while background bias influences the classifier's outputs. Thus, minimization of Grad-CAM backgrounds may not suppress shortcut learning. Summarizing, in relation to the state-of-the-art, we empirically and theoretically demonstrate the ISNet's superior capacity of avoiding the influence of background bias, thus hindering shortcut learning and improving generalization.

Besides its superior resistance to background bias, the ISNet introduces no increment in run-time computational cost to its backbone classifier. E.g., by replacing a U-Net followed by a DenseNet121 (segmentation-classification pipeline) with an ISNet containing a DenseNet121 backbone, we obtain a model that is about 70% to 108% faster at run-time and has almost 80% less parameters. Indeed, the ISNet matches a standard DenseNet121 run-time speed and size (Supplementary Note 9). Accordingly, the proposed architecture is an efficient technique to suppress background attention and shortcut learning, increasing confidence in a DNN's decisions. Moreover, it accepts virtually any backbone classifier and represents an interpretable attention mechanism: we know that it works by hindering attention outside of a region of interest, which is clearly defined by the ground-truth segmentation targets used during training. In summary, this study empirically and theoretically demonstrated that the ISNet's optimization of LRP heatmaps is a flexible approach, which produces deep classifiers that are resistant to background bias while retaining high accuracy and efficiency.

## Methods
### Layer-wise relevance propagation
Since DNNs are complex and nonlinear structures with millions of parameters, it is difficult to explain their decisions. Layer-wise relevance propagation[2] (LRP) is an explanation technique tailored for deep models, providing heatmaps to interpret DNNs. A past work qualitatively and quantitatively demonstrated that LRP explanations provide higher resolution heatmaps and more interpretable information than attention mechanisms and the corresponding attention heatmaps[37]. Furthermore, in relation to standard attention, LRP heatmaps can reveal additional evidence used by the classifier to make a decision[37]. Other studies compared explanation techniques and found LRP to be among the most robust, surpassing Grad-CAM and Gradient*Input[38,39]. From a theoretical perspective, LRP is rooted in the Deep Taylor Decomposition framework[40,41]: it explains a classifier's decision by approximating a series of local Taylor expansions, performed at each DNN neuron (section "ISNet theoretical fundamentals").

For each class, an LRP heatmap explains the influence of the input image regions on the classifier's confidence for that class. LRP is based on a semi-conservative propagation[2] of a quantity called relevance

**Table 5 | Performance metrics for the deep neural networks in tuberculosis detection (o.o.d. evaluation)[a]**

| Model and metric | Normal | Tuberculosis | Mean (macro-average) |
|---|---|---|---|
| ISNet precision | 0.744 ± 0.045 | 0.734 ± 0.043 | 0.739 ± 0.044 |
| U-Net+DenseNet121 precision | 0.63 ± 0.061 | 0.573 ± 0.043 | 0.601 ± 0.052 |
| DenseNet121 precision | 0.578 ± 0.055 | 0.564 ± 0.046 | 0.571 ± 0.05 |
| Multi-task U-Net precision | 0.515 ± 0.048 | 0.539 ± 0.053 | 0.527 ± 0.05 |
| AG-Sononet precision | 0.731 ± 0.06 | 0.599 ± 0.041 | 0.665 ± 0.05 |
| Extended GAIN precision | 0.576 ± 0.04 | 0.766 ± 0.06 | 0.671 ± 0.05 |
| RRR precision | 0.663 ± 0.049 | 0.664 ± 0.046 | 0.663 ± 0.048 |
| Vision Transformer (ViT-B/16) precision | 0.52 ± 0.044 | 0.56 ± 0.059 | 0.54 ± 0.052 |
| ISNet recall | 0.714 ± 0.046 | 0.762 ± 0.042 | 0.738 ± 0.044 |
| U-Net+DenseNet121 recall | 0.409 ± 0.05 | 0.768 ± 0.042 | 0.589 ± 0.046 |
| DenseNet121 recall | 0.479 ± 0.05 | 0.659 ± 0.047 | 0.569 ± 0.048 |
| Multi-task U-Net recall | 0.586 ± 0.05 | 0.468 ± 0.05 | 0.527 ± 0.05 |
| AG-Sononet recall | 0.406 ± 0.05 | 0.855 ± 0.035 | 0.631 ± 0.042 |
| Extended GAIN recall | 0.883 ± 0.032 | 0.372 ± 0.048 | 0.627 ± 0.04 |
| RRR recall | 0.642 ± 0.049 | 0.685 ± 0.046 | 0.663 ± 0.048 |
| Vision Transformer (ViT-B/16) recall | 0.679 ± 0.047 | 0.395 ± 0.049 | 0.537 ± 0.048 |
| ISNet F1-Score | 0.729 ± 0.046 | 0.748 ± 0.043 | 0.738 ± 0.044 |
| U-Net+DenseNet121 F1-Score | 0.496 ± 0.056 | 0.656 ± 0.044 | 0.576 ± 0.05 |
| DenseNet121 F1-Score | 0.524 ± 0.052 | 0.608 ± 0.047 | 0.566 ± 0.05 |
| Multi-task U-Net F1-Score | 0.548 ± 0.049 | 0.501 ± 0.052 | 0.524 ± 0.05 |
| AG-Sononet F1-Score | 0.522 ± 0.057 | 0.704 ± 0.04 | 0.613 ± 0.048 |
| Extended GAIN F1-Score | 0.697 ± 0.039 | 0.501 ± 0.056 | 0.599 ± 0.048 |
| RRR F1-Score | 0.652 ± 0.049 | 0.674 ± 0.046 | 0.663 ± 0.048 |
| Vision Transformer (ViT-B/16) F1-Score | 0.589 ± 0.046 | 0.463 ± 0.054 | 0.526 ± 0.05 |
| ISNet AUC | 0.809 ± 0.031 | 0.809 ± 0.031 | 0.809 ± 0.031 |
| U-Net+DenseNet121 AUC | 0.667 ± 0.039 | 0.667 ± 0.039 | 0.667 ± 0.039 |
| DenseNet121 AUC | 0.576 ± 0.04 | 0.576 ± 0.04 | 0.576 ± 0.04 |
| Multi-task U-Net AUC | 0.549 ± 0.041 | 0.549 ± 0.041 | 0.549 ± 0.041 |
| AG-Sononet AUC | 0.717 ± 0.037 | 0.717 ± 0.037 | 0.717 ± 0.037 |
| Extended GAIN AUC | 0.676 ± 0.038 | 0.676 ± 0.038 | 0.676 ± 0.038 |
| RRR AUC | 0.728 ± 0.036 | 0.728 ± 0.036 | 0.728 ± 0.036 |
| Vision Transformer (ViT-B/16) AUC | 0.558 ± 0.041 | 0.558 ± 0.041 | 0.558 ± 0.041 |

[a]The cells display the metrics' mean and 95% confidence intervals.

through the DNN layers, starting from one of the outputs of DNN's last layer neurons (logits), and ending at the input layer, where the heatmap is produced. The meaning of the relevance in the heatmap is determined by the choice of the logit where the propagation starts. Positive values indicate that an input image pixel positively influenced the logit, increasing the classifier confidence in the class it represents. Meanwhile negative values indicate areas that reduced this confidence (e.g., regions that the classifier related to other classes in a multi-class, single-label problem). The relevance's magnitude indicates how important the image regions were for the classifier's decision.

After choosing the explained output neuron, we define its relevance as its output (the logit, prior to nonlinear activation), and set the relevance of all other last-layer neurons to zero. Afterwards, diverse rules propagate the relevance through each DNN layer, one at a time, until the model's input. The choice of propagation rules influences the heatmap's interpretability, noisiness, and the stability of the propagation[40]. The most basic rule is called LRP-0. We define the k-th output of a fully-connected layer, $z_k$, before the layer's non-linear activation, as:

$$z_k = \sum_j w_{jk} a_j \tag{1}$$

Where $w_{jk}$ represents the layer's weight connecting its input j ($a_j$) to output k ($z_k$). The output k bias parameter is represented as $w_{0k}$, and the equation assumes $a_0 = 1$. LRP-0 propagates the relevance from the layer output, $R_k$, to its input, $R_j$, according to the following equation[40]:

$$R_j = \sum_k \frac{w_{jk} a_j}{z_k} R_k \tag{2}$$

LRP-0 redistributes the relevance from the layer's k-th output ($R_k$) to its inputs ($a_j$) according to how much they contributed to the k-th output ($z_k$). A second rule, LRP-$\varepsilon$, changes LRP-0 to improve the relevance propagation stability, noisiness, and the explanation's contextualization and coherence (section "ISNet theoretical fundamentals"). It adds a small constant, $\varepsilon$, to $z_k$. Being sign($\cdot$), a function evaluating to 1 for positive or zero arguments, and to -1 otherwise, LRP-$\varepsilon$ is defined as:

$$R_j = \sum_k \frac{w_{jk} a_j}{z_k + \text{sign}(z_k)\varepsilon} R_k \quad \text{,where } \varepsilon > 0 \tag{3}$$

It is possible to adopt a different rule for the DNN input layer, taking into account the input space domain. For images, a rule called LRP-z[B40] considers the maximum and minimum pixel values allowed in the figures (Supplementary Note 4.3). LRP has defined rules for the most common DNN layers. The technique is scalable and can be efficiently implemented and applied to virtually any neural network architecture. Since convolutions have equivalent fully-connected layers, the rules explained here directly apply to them. However, efficient implementations of LRP for convolutions and other layers are presented in Supplementary Note 4.

**Background relevance minimization and ISNet**
Layer-wise Relevance Propagation was created to show how input features influenced classifier's decisions[2]. With the ISNet, we suggest directly optimizing LRP to improve classifiers' behavior. During training, background relevance minimization (BRM) penalizes undesired relevance in the training images' LRP heatmaps, constraining the classifier to learn decision rules that do not rely on background features. Algorithm 1 explains BRM, the ISNet training procedure. The heatmap loss ($L_{LRP}$, Algorithm 1, step 2d, explained in section "ISNet loss function") employs gold standard foreground segmentation masks to identify and penalize the background relevance in LRP heatmaps. The masks are only necessary for training. They are figures valued one in the image's foreground, and zero in the background. When not available, a pretrained segmenter can create them. E.g., a U-Net[32] pretrained to segment a specific type of foreground, or a general model pretrained for novel class segmentation (like DeepMAC[29]). Supplementary Note 6 details training settings, hyper-parameters, data processing, and augmentation used for the applications in this study, considering both the ISNet and the benchmark models.

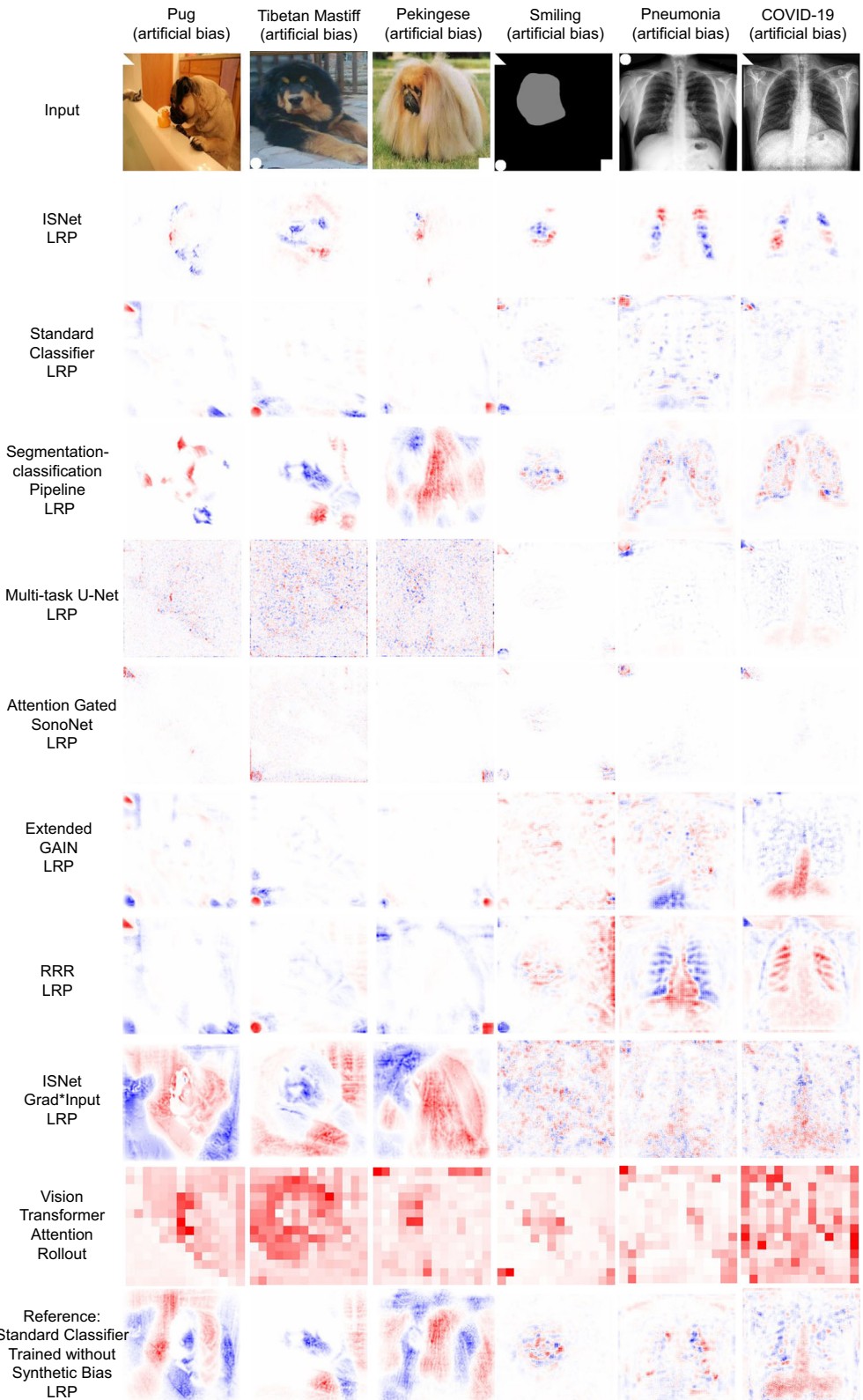

**Fig. 1 | Heatmaps (Layer-wise Relevance Propagation/LRP for convolutional networks and attention rollout for Vision Transformer) for positive COVID-19 and Pneumonia X-rays and photographs, extracted from the synthetically biased test datasets (biased test).** Last row displays classifier trained without the synthetic bias (and analyzing images without the bias), for reference. The image's true class is stated above the figures, and the DNN that produced the heatmap is identified on the left. The triangle (background bias) indicates the classes COVID-19, smiling or Pug. The circle pneumonia, high cheekbones, and Tibetan Mastiff. The square rosy cheeks and Pekingese. Red colors in the LRP maps indicate areas the DNN associated to the image's true class, while blue colors are areas that reduced the network confidence for the class. For attention rollout, red shows the DNN attention. White represents areas with little influence over the classifiers. DNN focus on the images' foregrounds (dogs, faces, or lungs), which results in whiter heatmap backgrounds, is desirable. For privacy, the face picture was substituted by a representation of the face (gray) and bias (white) locations, but classifiers received the real picture.

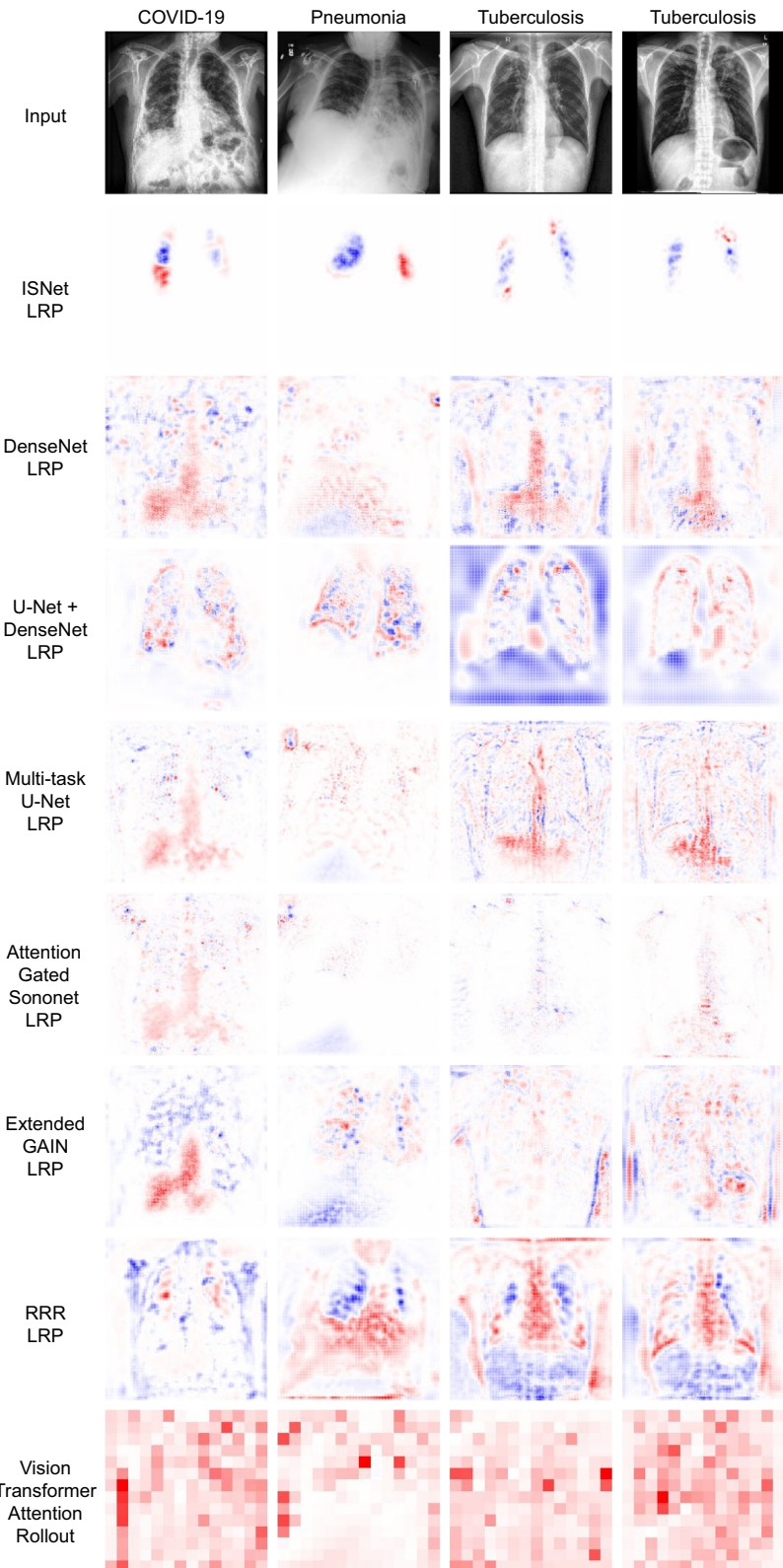

**Fig. 2 | Heatmaps (Layer-wise Relevance Propagation/LRP for convolutional networks and attention rollout for Vision Transformer) for positive COVID-19, Pneumonia, and tuberculosis.** The image's true class is stated above the figures, the DNN that produced the heatmap is identified on the left. For LRP, red colors indicate areas that the DNN associated to the true class, while blue colors are areas that decreased the network confidence for the class. For attention rollout, red indicates the DNN attention. White represents areas with little influence over the classifiers. DNN focus on the images' foregrounds (lungs), which results in whiter heatmap backgrounds, is desirable. Examples of background bias are markings over the right shoulder in the pneumonia X-ray, a letter R in the neck region of the left TB X-ray, and an L over the left shoulder in the other tuberculosis X-ray. Only the heatmaps for the ISNet and the U-Net + DenseNet show no attention to these biases. Body regions outside of the lungs also represent background bias, which the ISNet ignored as well.

**Algorithm 1. Background relevance minimization: the ISNet training procedure**

1: Initialization: randomly initialize the backbone classifier's trainable parameters, $\boldsymbol{\theta}$.
2: Training epoch: for every randomly drawn mini-batch (images $\mathbf{X}$, segmentation ground-truth foreground masks $\mathbf{M}$, and classification labels $\mathbf{Y}$) in the training dataset:
   a: Preprocess and (optional) augment the mini-batch.
   b: Classifier forward pass: classify the B (mini-batch size) images with the backbone classifier, assigning probabilities ($\hat{\mathbf{Y}}$) for the K possible classes.
   c: Layer-wise Relevance Propagation: create $B \times K$ differentiable LRP heatmaps in parallel, $\mathbf{H}$, explaining the K classifier's logits for the B input images. Employ LRP-$\varepsilon$ (we set $\varepsilon = 0.01$) throughout the entire DNN, except for its first layer, where we use LRP-$z^B$.
   d: Loss calculation: calculate classification loss (e.g., cross-entropy) according to the classifier output and the classification labels, $L_C(\hat{\mathbf{Y}}, \mathbf{Y})$. Calculate the heatmap loss according to the LRP heatmaps and the foreground masks, $L_{LRP}(\mathbf{H}, \mathbf{M})$. Linearly combine both to produce the ISNet loss, using a balancing hyper-parameter, P: $L_{IS} = (1 - P).L_C + P.L_{LRP}$, where $0 \leq P \leq 1$. Increasing P increases the ISNet resistance to background bias, but it may reduce training speed.
   e: Gradient backward pass: use automatic backpropagation to calculate the gradient of the ISNet loss with respect to the backbone classifier's trainable parameters, $\Delta_{\boldsymbol{\theta}} L_{IS}$.
   f: Optimizer step: update the backbone classifier's trainable parameters according to the gradient $\Delta_{\boldsymbol{\theta}} L_{IS}$. We use stochastic gradient descent with momentum as the optimizer.
3: Validation epoch. For every mini-batch (images $\mathbf{X}$, segmentation ground-truth foreground masks $\mathbf{M}$, and classification labels $\mathbf{Y}$) in the hold-out validation dataset:
   a: Perform the image processing, classifier forward pass, LRP, and loss calculation as described in steps 2a (except for data augmentation), 2b, 2c, and 2d. Monitor the average validation ISNet loss at the end of each epoch.
4: Repeat steps 2 and 3 until the maximum number of epochs (N) is reached. At the end of the training procedure, return the backbone classifier with the parameters ($\boldsymbol{\theta}$) that minimized the hold-out validation ISNet loss.

We introduce an efficient implementation of Layer-wise Relevance Propagation in PyTorch, dubbed LRP Block, which produces differentiable LRP heatmaps in parallel. The LRP Block allows automatic backpropagation through LRP. Thus, the heatmap loss gradient can be backpropagated from the loss output until the backbone classifier's parameters, allowing the minimization of the ISNet loss (Algorithm 1, step 2e). The block is presented in detail in Supplementary Note 4, which explains its LRP implementation for multiple types of classifier layers. The structure can be deactivated or removed after the training procedure, representing no run-time computational cost. Alternatively, it can explain the trained ISNet's decisions with LRP: minimal background relevance indicates the success of background relevance minimization. Since LRP can be applied to virtually any DNN[40], the ISNet can accept virtually any classifier backbone architecture.

We need to produce one heatmap for each possible class, starting the LRP relevance propagation at the output neuron that classifies it (Algorithm 1, step 2c). We cannot minimize background LRP relevance for a single class (e.g., the winning one). Imagine that we have a bias in the image background, associated with class C, and we minimize only the background LRP relevance for class C. In this case, the classifier can negatively associate all other classes with the bias, using it to lower their outputs, making the class C output neuron the winning one. This negative association is expressable as negative relevance in the other classes' heatmaps. Consequently, the penalization of positive and

negative background relevance in all maps is a solution to the problem. Accordingly, the ISNet training time increases with the number of categories in the classification task (K). For efficiency, the LRP Block propagates relevance in batches, producing multiple independent heatmaps in parallel. If a memory limit is reached, heatmaps will need to be produced in series, making training time linearly increase with K. Future ISNet implementations may reduce this drawback. However, as heatmaps are not necessary after training, the run-time ISNet is as fast and efficient as its backbone classifier. Thus, the ISNet exchanges training time for run-time performance, especially when compared to the benchmark segmentation-classification pipeline (Supplementary Note 9). This trade-off may be very profitable, given that DNNs can be trained with powerful computers, then later deployed in less expensive or portable devices.

## ISNet loss function

The ISNet loss, $L_{IS}$, is the function minimized during ISNet training (Algorithm 1, step 2d). It is a linear combination of two terms (Equation (4)): $L_C$, a standard classification loss (e.g., cross-entropy), and $L_{LRP}$, the heatmap loss. Their influence over the loss gradient is balanced by a hyper-parameter P. The heatmap loss, $L_{LRP}$, is also a linear combination of two functions, the background ($L_1$) and the foreground ($L_2$) losses (Equation (5)). The combination utilizes two hyper-parameters, $w_1$ and $w_2$. Both $L_1$ and $L_2$ depend on the LRP heatmaps and the foreground segmentation masks.

$$L_{IS} = (1 - P).L_C + P.L_{LRP}, \text{ where } 0 \leq P \leq 1 \quad (4)$$

$$L_{LRP} = w_1.L_1 + w_2.L_2, \text{ where } 0 < w_1 \text{ and } 0 < w_2 \quad (5)$$

Algorithms 2 and 3 describe the calculation of $L_1$ and $L_2$, respectively. The background loss quantifies and penalizes background relevance in LRP heatmaps. Meanwhile, the foreground loss is an auxiliary term, which ensures the stability of the LRP heatmaps during ISNet training, avoiding zero maps or exploding LRP relevance. Supplementary Note 1 presents a more detailed description of the functions. Supplementary Note 6.3 details which hyper-parameters require a fine search, a coarse search, or no search. It also explains our hyper-parameter tuning strategy. The ISNet loss is differentiable, and PyTorch can perform automatic gradient backpropagation through it.

**Algorithm 2**. The Background Loss, $L_1$
   Input: $B \times K$ heatmaps, where B is the mini-batch size and K the number of classes in the classification task.
   A corresponding foreground segmentation mask, $\mathbf{M_{bk}}$, for each heatmap $\mathbf{H_{bk}}$.
   Output: $L_1$, the scalar background loss for the mini-batch.
1: Absolute heatmaps: take the absolute value of the LRP heatmaps (element-wise), abs($\cdot$). This step ensures that the background loss equally penalizes positive and negative LRP relevance. $\mathbf{H_{bk}}$ is the LRP heatmap explaining the classifier output (logit) for class k, considering the mini-batch image b as the DNN's input.
   abs($\mathbf{H_{bk}}$)
2: Normalized absolute heatmaps: normalize the absolute heatmaps, dividing each map by the absolute average relevance in its foreground region. Use foreground masks, $\mathbf{M_{bk}}$, to define such regions. This step makes the background loss relative. I.e., the loss minimization makes the influence of background features on the classifier progressively smaller than the influence of foreground features. Moreover, because $L_1$ is relative, an overall reduction of both background and foreground LRP relevance (which does not represent foreground focus) cannot minimize the loss. Sum($\cdot$) adds all elements in a tensor, and $\odot$ represents element-wise multiplication.

$\mathbf{H}'_{\mathbf{bk}} = \text{abs}(\mathbf{H}_{\mathbf{bk}}) \div \{[\text{Sum}(\text{abs}(\mathbf{H}_{\mathbf{bk}}) \odot \mathbf{M}_{\mathbf{bk}})/(\text{Sum}(\mathbf{M}_{\mathbf{bk}}) + e)] + e\}$, where $0 < e \ll 1$

3: Segmented heatmaps: in the normalized absolute heatmaps, set all foreground relevance to zero, by element-wise multiplying the maps and inverted foreground masks (i.e., figures valued one in the background and 0 in the foreground, $\mathbf{1} - \mathbf{M}_{\mathbf{bk}}$). This step ensures the loss penalizes only background LRP relevance.

$\mathbf{UH}'_{\mathbf{bk}} = (\mathbf{1} - \mathbf{M}_{\mathbf{bk}}) \odot \mathbf{H}'_{\mathbf{bk}}$

4: Raw background attention scores: use Global Weighted Ranked Pooling[42] (GWRP) over the segmented heatmaps, obtaining one scalar score per map channel. GWRP can be seen as a hybrid between average pooling and max pooling. It is governed by a hyper-parameter d ($0 \le d \le 1$). If $d = 0$, GWRP matches max pooling. If d = 1, GWRP matches average pooing. The lower the d, the more the background loss penalizes the existence of small background regions with strong influence over the classifier. Thus, lowering d increases the ISNet resistance to background bias. However, smaller values can decrease training stability. GWRP outputs one scalar, $r_{bkc}$, per channel (c) in a segmented heatmap $\mathbf{UH}'_{\mathbf{bk}}$. $\mathbf{UH}'_{\mathbf{bk}}$ has 3 channels, $\mathbf{UH}'_{\mathbf{bkc}}$, when the ISNet classifies RGB images.

$r_{bkc} = \text{GWRP}(\mathbf{UH}'_{\mathbf{bkc}})$

5: Activated scores: pass the raw scores, $r_{bkc}$, through the non-linear function $f(r_{bkc}) = r_{bkc}/(r_{bkc} + E)$, where E is a constant hyper-parameter, normally set as 1. The activated scores are naturally limited between 0 and 1, being an adequate input for cross-entropy.

6: Background attention loss: calculate the cross-entropy between the activated scores and a zero target, $CE(\cdot)$. We utilize a zero objective in cross-entropy, because lower activated scores represent weaker background attention.

$CE(f(r_{bkc})) = -\ln(1 - f(r_{bkc}))$

7: $L_1$: calculate the average background attention loss for all heatmaps in the training mini-batch:

$L_1 = \frac{1}{B.K.C} \sum_{b=1}^{B} \sum_{k=1}^{K} \sum_{c=1}^{C} CE(f(r_{bkc}))$

**Algorithm 3**. The Foreground Loss, $L_2$

Input: $B \times K$ heatmaps, where B is the mini-batch size and K the number of classes in the classification task.

A corresponding foreground segmentation mask, $\mathbf{M}_{\mathbf{bk}}$, for each heatmap $\mathbf{H}_{\mathbf{bk}}$.

Output: $L_1$, the scalar background loss for the mini-batch.

1: Absolute heatmaps: take the absolute value of the LRP heatmaps (element-wise), $\text{abs}(\cdot)$. $\mathbf{H}_{\mathbf{bk}}$ is the LRP heatmap explaining the classifier output (logit) for class k, considering the mini-batch image b as the DNN's input.

$\text{abs}(\mathbf{H}_{\mathbf{bk}})$

2: Segmented heatmaps: in the absolute heatmaps, set all background relevance to zero, by element-wise multiplying them by the foreground masks. This steps avoids a direct interference of the foreground loss on the minimization of LRP background relevance, caused by the background loss ($L_1$) optimization.

$\text{abs}(\mathbf{H}_{\mathbf{bk}}) \odot \mathbf{M}_{\mathbf{bk}}$

3: Absolute foreground relevance: sum all elements in the segmented maps, $\text{Sum}(\cdot)$, obtaining the total absolute foreground relevance per-heatmap. $g(\mathbf{H}_{\mathbf{bk}}) = \text{Sum}(\text{abs}(\mathbf{H}_{\mathbf{bk}}) \odot \mathbf{M}_{\mathbf{bk}})$

4: Square losses: if a heatmap's absolute foreground relevance, $g(\mathbf{H}_{\mathbf{bk}})$, is smaller than a hyper-parameter, $C_1$, the corresponding square loss is $L_2^{bk} = (C_1 - g(\mathbf{H}_{\mathbf{bk}}))^2/C_1^2$. If it is larger than $C_2$ ($C_2 > C_1 > 0$), we have $L_2^{bk} = (C_2 - g(\mathbf{H}_{\mathbf{bk}}))^2/C_1^2$. If $C_1 > g(\mathbf{H}_{\mathbf{bk}}) > C_2$, we define $L_2^{bk} = 0$. Accordingly, the foreground loss, $L_2^{bk}$, and its gradient are zero when the heatmap's absolute foreground relevance ($g(\mathbf{H}_{\mathbf{bk}})$) is within a pre-defined range, $[C_1, C_2]$. However, the loss raises quadratically when the absolute foreground relevance exits the range. The $C_1$ and $C_2$ hyper-parameters are set to represent a natural range of absolute

relevance (Supplementary Note 6.3).

$$L_2^{bk} = \begin{cases} \frac{(C_1 - g(\mathbf{H}_{\mathbf{bk}}))^2}{C_1^2}, & \text{if } g(\mathbf{H}_{\mathbf{bk}}) < C_1 \\ 0, & \text{if } C_1 \le g(\mathbf{H}_{\mathbf{bk}}) \le C_2 \\ \frac{(g(\mathbf{H}_{\mathbf{bk}}) - C_2)^2}{C_2^2}, & \text{if } g(\mathbf{H}_{\mathbf{bk}}) > C_2 \end{cases}$$

5: $L_2$: take the average of all square losses, considering all heatmaps in the mini-batch. $L_2$ will be zero if the LRP relevance stays within normal values, but it quickly raises if it exits this natural range. Thus, $L_2$ avoids zero or exploding LRP heatmaps during ISNet training.

$L_2 = \frac{1}{B.K} \sum_{b=1}^{B} \sum_{k=1}^{K} L_2^{bk}$

## ISNet theoretical fundamentals

We could not find other works optimizing LRP explanations to improve classifiers' behavior. Layer-wise relevance propagation sits among the most robust, interpretable, and high-resolution explanations to date[38,37]. While the relevance signal is propagated through the DNN (from output to input), it extracts context and high level features from late DNN layers, and captures precise spatial information from earlier layers. Accordingly, the resulting explanation heatmap carries both high resolution and high level of abstraction. Therefore, LRP optimization can teach the ISNet to precisely identify the images' foreground features, and to form decision rules based on them (Supplementary Note 4.5).

In this Section, we theoretically justify the empirically verified (section "Results") ISNet capacity of avoiding background attention, and the shortcut learning caused by background bias. We begin by summarizing the mathematical fundamentals behind LRP, elucidating its relationship with Taylor expansions. Afterwards, we discuss how LRP optimization makes the ISNet robust to background bias.

**LRP mathematical fundamentals.** A DNN implements a function of its input, $y = f(\mathbf{X})$, where $y$ is a network's logit of interest. A first-order Taylor expansion can decompose the logit into a summation of one term per input dimension ($x_j$), plus an additional zero-order term ($f(\tilde{\mathbf{X}})$), and an approximation error ($\rho$) with respect to higher-order Taylor expansions (Eq. (6)). To minimize the approximation error (i.e., the Taylor residuum, $\rho$), the Taylor reference ($\tilde{\mathbf{X}}$, with elements $\tilde{x}_j$) should be close to the data point $\mathbf{X}$. Furthermore, by setting the reference point as a root of the function $f(\mathbf{X})$, we remove the zero-order element ($f(\tilde{\mathbf{X}}) = 0$). Thus, with a nearby root as the Taylor reference, the logit is decomposed as the summation of the first-order terms, with one term per input dimension (Eq. (7)). In this case, the terms explain the differential contribution of each input element to the logit, with respect to the logit's state of maximal uncertainty, when it is zero (50% class probability for a sigmoid function, and the ReLU function hinge)[2]. I.e., the terms explain how each input element contributed to making the logit different from zero. Accordingly, the terms can form a heatmap, explaining and decomposing the network's output, $y$. For the explanations to be more meaningful, the Taylor reference should reside in the classification problem data manifold[2].

$$f(\mathbf{X}) = f(\tilde{\mathbf{X}}) + \sum_j (x_j - \tilde{x}_j) \frac{\partial f}{\partial x_j}\bigg|_{\tilde{\mathbf{X}}} + \rho \qquad (6)$$

$$f(\mathbf{X}) \approx \sum_j (x_j - \tilde{x}_j) \frac{\partial f}{\partial x_j}\bigg|_{\tilde{\mathbf{X}}} \qquad (7)$$

Although principled, adequate Taylor explanations are difficult to compute for deep neural networks. The function $f(\mathbf{X})$ is complicated, highly non-linear, and can have noisy gradients. Finding a root $\tilde{\mathbf{X}}$ that satisfies all aforementioned requirements is a complex and analytically

intractable problem[41]. However, a DNN function, $f(\mathbf{X})$, is defined as a structure of simpler sub-functions, learned at each neuron. Finding Taylor references for the simpler functions is easier, and approximate analytical solutions can be found for the corresponding Taylor expansions[41].

The LRP relevance propagation rules we employ for the ISNet (LRP-$\varepsilon$ and LRP-$z^B$) are justified by the Deep Taylor Decomposition (DTD) framework[41,40]. DTD propagates relevance, from the classifier's logit to its inputs, according to local Taylor expansions performed at each neuron. The neuron output relevance ($R_k$, received from the subsequent layer) is viewed as a function of the neuron's inputs, $\mathbf{a}$ (composed of elements $a_j$). Accordingly, Equation (8) displays a first-order Taylor expansion of $R_k(\mathbf{a})$, considering a reference point $\tilde{\mathbf{a}}$.

$$R_k(\mathbf{a}) = R_k(\tilde{\mathbf{a}}) + \sum_j (a_j - \tilde{a}_j) \frac{\partial R_k}{\partial a_j}\bigg|_{\tilde{\mathbf{a}}} + \rho \qquad (8)$$

The relevance $R_k(\mathbf{a})$ is redistributed to the neuron's inputs $a_j$ according to the first-order terms in the expansion (summed terms in Eq. (8))[40]. Ideally, we choose a reference point ($\tilde{\mathbf{a}}$) that minimizes the zero-order term $R_k(\tilde{\mathbf{a}})$. We also want the reference to be close to the data point $\mathbf{a}$, as the proximity reduces the Taylor residuum ($\rho$). Finding this reference point is still not simple, nor computationally inexpensive, considering the complexity of the function $R_k(\mathbf{a})$[2]. Therefore, to obtain a closed-form solution for the Taylor expansion, LRP considers an approximation of $R_k(\mathbf{a})$ (dubbed approximate relevance model, $\hat{R}_k(\mathbf{a})$), and standardized choices of the reference $\tilde{\mathbf{a}}$. These choices are justified by the approximate relevance model and the neuron's input domain[40]. For a neuron with ReLU activation (Eq. (9)), a modulated ReLU activation is the most common relevance model ($\hat{R}_k(\mathbf{a})$). It is defined in Eq. (10), where $c_k$ is a constant chosen to force the model to match the true relevance at the data point $\mathbf{a}$ ($R_k(\mathbf{a}) = \hat{R}_k(\mathbf{a})$). For further explanation and theoretical justification of the relevance model, please refer to ref. 40.

$$a_k = \max(0, z_k) = \max\left(0, \sum_j w_{jk} a_j\right) \qquad (9)$$

$$\hat{R}_k(\mathbf{a}) = \max\left(0, \sum_j w_{jk} a_j\right) c_k \qquad (10)$$

The LRP-$\varepsilon$ and LRP-0 rules are derived by selecting different reference points ($\tilde{\mathbf{a}}$) for the approximate local Taylor expansions[40]. All these points satisfy $\tilde{a}_j \geq 0$, thus residing in the neuron's input domain (considering that it follows other neurons with ReLU activations). For LRP-0, we have $\tilde{\mathbf{a}} = \mathbf{0}$. For LRP-$\varepsilon$ the definition is: $\tilde{\mathbf{a}} = \frac{\varepsilon}{a_k + \varepsilon} \mathbf{a}$. The Euclidean distance between the LRP-$\varepsilon$ reference point and the actual data point, $\mathbf{a}$, is much smaller than the distance between $\mathbf{a}$ and the LRP-0 reference point[40]. Therefore, in comparison to LRP-0, LRP-$\varepsilon$ reduces the Taylor residuum ($\rho$) in relation to higher order Taylor expansions, creating heatmaps that are more faithful, less noisy, and more contextualized[40].

**Optimizing for background bias resistance: why LRP?.** There are some key qualities we expect the optimized explanation strategy to have: it must be differentiable, consider both positive and negative evidence (section "Background relevance minimization and ISNet"), and be computationally efficient. Moreover, we search for an explanation methodology that can fundamentally justify why its optimization leads to resistance to background bias. LRP-$\varepsilon$[2] satisfies these requirements. As we show in the LRP block, the strategy is differentiable. Second, it is fast, constructing an LRP heatmap requires a single backpropagation of relevance through the neural network. Efficiency and differentiability are essential for explanations that

must be created during training and optimized. Third, LRP-$\varepsilon$ considers both positive and negative relevance. Finally, the technique is principled, because it approximates the sequential application of local Taylor expansions (per-neuron) in deep neural networks with ReLU activations. Interestingly, we had no success in preliminary experiments with the optimization of LRP rules that do not consider negative evidence (e.g., LRP-$\gamma$[40]), or that are not justified by the deep Taylor framework (e.g., LRP-$\alpha\beta$[40]). They could not produce background bias resistance.

As previously explained, the terms of a first-order Taylor expansion, considering an adequate Taylor reference (a nearby root in the data manifold), indicate the contribution of a function's input elements to its output variation, when the input moves from the Taylor reference to the actual data point. Thus, the minimization of terms associated with bias should minimize the bias contribution to the function output variation, when its input moves from a point of maximal uncertainty (root) to the current data point. I.e., it minimizes the bias influence over the function's output. However, the creation of adequate Taylor explanations of a DNN is a complex and analytically intractable problem[2]. Thus, such explanations violate our requirement of computational efficiency.

However, LRP-$\varepsilon$ is a fast procedure, which explains a DNN output by approximating a sequence of local Taylor expansions. For this reason, LRP-$\varepsilon$ optimization is an efficient and justifiable alternative to minimize the influence of biased input elements on the network's logits. Recapitulating, LRP-$\varepsilon$ relevance propagation starts with the classifier logit we are explaining and uses an approximate Taylor expansion to decompose its value, and redistribute it to the inputs of the last DNN layer, according to their contribution to the logit. Repeating this procedure, the decomposition results (relevance) are further decomposed and redistributed multiples times through the DNN. I.e., an approximate Taylor expansion at each DNN neuron decomposes its output relevance and redistributes it to the neuron's inputs, according to how much they contributed to the neuron's output relevance (Eq. (8)). The procedure ends at the input layer, forming the LRP heatmap.

For the background bias to influence the logit, it needs to influence neurons (or convolutional activations) throughout the entire DNN, until its last layer. The layer L neurons that carry and process the bias information must influence neurons in layer L+1, or the bias information will not reach the DNN output. This influence will be captured by the local Taylor expansions performed at the layer L+1 neurons, affecting the LRP relevance flow. When background bias influences the classifier's decisions, it produces a flow of influence from the bias to the logit, encompassing the neurons and connections that carry and process the bias information. This influence flow will cause a corresponding flow of LRP relevance, bringing relevance from the logit to the background region of the LRP heatmap. The ISNet's background relevance minimization procedure optimizes DNN parameters to constrict and ultimately stop the relevance flow from the logits to the LRP heatmaps' background, thus minimizing the corresponding influence and information flow from the background bias to the logit. Accordingly, the ISNet hinders the background bias influence on the classifier's outputs.

We selected the LRP-$\varepsilon$ rule to propagate relevance through all DNN, except for its first layer. The input domain of the first DNN layer is diverse. In a network with ReLU activations and analyzing images, the first layer inputs range from 0 to 1 (standardized pixels), while other layers' inputs assume values in $\mathbf{R}^+$. While LRP-$\varepsilon$ is adequate for the remaining layers, the LRP-$z^B$ rule represents a more accurate choice of Taylor reference for the first layer[41]. According to preliminary tests, the ISNet works with the LRP-$\varepsilon$ in layer one, but LRP-$z^B$ produced an accuracy improvement.

Meanwhile, LRP-0 explanations are highly noisy, and less interpretable than LRP-$\varepsilon$. In the LRP-$\varepsilon$ propagation rule (Eq. (3)), the $\varepsilon$ term not only avoids division by zero, but it absorbs some of the relevance

that would have been propagated to the lower layer. The absorption mostly reduces the influence of neurons with small activations ($z_k$ in Eq. (3)) over the relevance signal being propagated. Therefore, it reduces noise and contradiction in the resulting heatmaps[40]. This improvement is justified by the DTD framework. As previously noticed, in relation to LRP-0, LRP-$\varepsilon$ represents a deep Taylor decomposition using Taylor references that are closer to the data points. Thus, LRP-$\varepsilon$ reduces the Taylor residuum, and explanations become more contextualized and coherent with the network's behavior[40].

Past studies showed that, in DNNs based on ReLU activations, LRP-0 is equivalent to Gradient*Input explanations (assuming no division by zero or numerical instabilities in LRP-0)[40]. The equality demonstrates that the advantages of LRP-$\varepsilon$ over LRP-0 directly apply when comparing LRP-$\varepsilon$ to Gradient*Input. Gradient*Input is an explanation technique proposed to improve the sharpness of input gradients[8], by multiplying them (element-wise) with the DNN input itself. Input gradients (or saliency maps) are another explanation technique[7], representing the gradient of a DNN logit with respect to the model's input. To create input gradients and Gradient*Input explanations, we backpropagate the gradient of the logit corresponding to the class we want the heatmap to explain. The ISNet Grad*Input is an ablation study, where we substituted the ISNet LRP heatmaps by Gradient*Input. Right for the Right Reasons (RRR) is a model that optimizes input gradients (alongside a standard classification loss), minimizing their background values, which are identified by ground-truth foreground masks[6]. Essentially, DNN optimizers treat the networks' inputs as constants. Consequently, background minimization in Gradient*Input minimizes the input gradient's background elements. Therefore, Gradient*Input optimization shares the solid fundamental from RRR: the minimization of input gradients' backgrounds make classifier's outputs locally invariant to changes in images' backgrounds. The learned local invariance can be generalizable: during testing, RRR also based its decisions on foreground features, instead of background bias[6]. Both input gradients and Gradient*Input are computationally efficient (similar to LRP, refer to Supplementary Note 9), differentiable, and show positive and negative evidence.

However, both input gradients and Gradient*Input are noisy for large DNNs analyzing high-resolution images[39]. RRR was originally tested in networks with few hidden layers[6]. This study considers deeper classifier backbones (DenseNet121, with 121 layers, and VGG-19, with 19 layers) and 224x224 images. Accordingly, when employing the VGG-19, the ISNet's generalization and background bias resistance significantly surpassed RRR, and slightly surpassed the ISNet Grad*Input (with confidence interval overlap, Table 1). However, with the deeper backbone, the ISNet significantly surpassed both models, without overlaps (Table 1). In summary, our empirical findings (Tables 1 to 5) indicate that the LRP-$\varepsilon$ theoretical advantages over LRP-0 or Gradient*Input (lower noise, higher coherence, and better contextualization) allow the ISNet to more effectively and stably minimize background attention, better hindering the shortcut learning caused by background bias and improving o.o.d. generalization.

Supplementary Note 2 thoroughly compares the ISNet to RRR and the ISNet Grad*Input, and it explains the alternative models in more detail. Supplementary Note 8 formally demonstrates the equivalence between LRP-0 and Gradient*Input. Afterwards, it provides an alternative view on LRP-$\varepsilon$, which makes its denoising quality clear. Finally, Supplementary Note 2.4 displays the advantages of LRP optimization over the optimization of another popular explanation technique, Grad-CAM[5].

### Reporting summary

Further information on research design is available in the Nature Portfolio Reporting Summary linked to this article.

## Data availability

Source data are provided with this paper. The X-ray data from healthy and/or pneumonia positive subjects used in this study are available in the Montgomery and Shenzen databases[26], http://archive.nlm.nih.gov/repos/chestImages.php; in ChestX-ray14[43], https://paperswithcode.com/dataset/chestx-ray14; and in CheXpert[25], https://stanfordmlgroup.github.io/competitions/chexpert/. The COVID-19 radiography data used in this study are available in The BrixIA COVID-19 project[12], https://brixia.github.io/; and in the BIMCV-COVID19+ database[44], https://bimcv.cipf.es/bimcv-projects/bimcv-covid19/. The tuberculosis-positive X-rays used in this study are available in NIAID TB Portals[20], https://tbportals.niaid.nih.gov/download-data. The Images for facial attribute estimation (along with their segmentation masks) used in this study are available in the Large-scale CelebFaces Attributes (CelebA) Dataset[27], https://mmlab.ie.cuhk.edu.hk/projects/CelebA.html. The MIMIC-CXR-JPG database (v2.0.0)[45,46,47] used in this study is available at https://physionet.org/content/mimic-cxr-jpg/2.0.0/. The Stanford Dogs[28] dataset used in this study is available at http://vision.stanford.edu/aditya86/ImageNetDogs/. X-rays with lesions marked by the radiologist and the corresponding ISNet Layer-wise Relevance Propagation heatmaps (Supplementary Data 1), generated in this study, have been deposited in https://doi.org/10.6084/m9.figshare.24243895.v2[36]. All data supporting the findings described in this manuscript are available in the article and in the Supplementary Information and from the corresponding author upon request. Source data are provided with this paper.

## Code availability

The code containing the ISNet PyTorch implementation is available at https://github.com/PedroRASB/ISNet[30]. It also presents the implementations for the benchmark deep neural network architectures.

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

## Acknowledgements
P.R.A.S.B. thanks the funding from the Center for Biomolecular Nanotechnologies, Istituto Italiano di Tecnologia (73010, Arnesano, LE, Italy). A.C. thanks for the funding from the Istituto Italiano di Tecnologia (6163, Genova, GE, Italy). We gratefully acknowledge the HPC infrastructure and the Support Team at Fondazione Istituto Italiano di Tecnologia. Data were obtained from the TB Portals (https://tbportals.niaid.nih.gov), which is an open-access TB data resource supported by the National Institute of Allergy and Infectious Diseases (NIAID) Office of Cyber Infrastructure and Computational Biology (OCICB) in Bethesda, MD. These data were collected and submitted by members of the TB Portals Consortium (https://tbportals.niaid.nih.gov/Partners). Investigators and other data contributors that originally submitted the data to the TB Portals did not participate in the design or analysis of this study.

## Author contributions
P.R.A.S.B. developed the study's concept, implemented the neural networks, and analyzed the results. S.S.J.D. annotated lesions in the X-rays. A.C. supervised and reviewed the work.

## Competing interests
The authors declare no competing interests.

## Additional information

**Peer review information** : *Nature Communications* thanks Synho Do, Ben Glocker and the other, anonymous, reviewer(s) for their contribution to the peer review of this work.

