## [Peer Review File · Nature Communications]

Reviewers' Comments:

Reviewer #1:

Remarks to the Author:

This study conducts a classification of COVID19 (X-rays) by performing a segmentation strategy together. The proposed methodology is fast and accurate, however, there are major concerns related to this paper:

-- Conducting segmentation + classification together is not a new concept, and this can be done within multi-task learning, there exists bunch of works like this. There is no unique innovation as opposed to what authors claimed.

--unsure about the importance of the problem. Detection is not really done with x-rays or ct scans, pcr is the one it gives the initial assessment, later doctors use mostly ct to see progress, in some counties x-rays were preferable but the need for such complicated framework (deep learning based segmentation, classification, etc) is not clear. perhaps not needed, maybe only for very subtle change identification only.

--heatmap analysis are qualitative. It is well known nowadays that heat maps are showing only where the network learns, they are mostly not explaining anything. Very fragile concepts. Rather, radiographic concepts should be attached to classification so that explainability is satisfied.

--data in COVID19 projects should be carefully approached. imbalance Is one problem but there are also other heterogeneity problems existing, one should approach the problem with a care and data curation part should be clear and transparent. Bias in the data (collection, annotation, etc..) should be studied carefully, as is not the case in this study.

Reviewer #2:

Remarks to the Author:

The article presents an image classification approach with implicit segmentation via layer-wise relevance propagation. Evaluation is carried out on two applications, chest X-ray disease detection, and facial attribute recognition.

Generally, an interesting objective and idea is discussed regarding the use of implicit segmentation in image classification networks. The motivation is clear and sensible, where a classification network should ideally learn what image regions to ignore to be resilient to shortcut learning.

The proposed methodology, however, is not entirely convincing. Any fully-convolutional neural network used for image classification is inherently learning some level of object segmentation/localization encoded in the spatial feature maps. This finding gave rise to the very first segmentation approaches based on fully-convolutional nets (see Long et al. 2014 Fully Convolutional Networks for Semantic Segmentation) derived from image classification backbones. The proposed LRP-based approach seems overly complicated and a sensible baseline to compare with would be to train a segmentation network with an image classification head attached to the bottleneck feature representation. This would enable a straightforward combined segmentation/classification objective, without the overhead of the LRP. There is also considerable work on attention-gated networks for both segmentation and classification, which has not been discussed and may be considered for experimental comparison.

Besides the concerns about the methodology, I believe that the structure of the article could be greatly improved. Methods and results are mixed, where segmentation performance is reported within the method description. It was very confusing that multiple results for the lung segmentation U-net were reported at different places, that also did not seem to be consistent. It remains unclear to whether the same U-net was used to provide the automated segmentations for the proposed approach as for the baseline using a sequence of segmentation and classification. Conceptually, it is unclear why the latter should perform so much worse than the proposed method that does only learn implicit segmentation. Intuitively, I would have expected the explicit

segmentation followed by classification to yield an upper bound of the performance. This somewhat surprising result should be investigated further.

The qualitative results for the CelebA data in Fig. 1 raises some questions about the claims of the paper. In fact, I thought that the heatmaps are much more convincing for the sequential segmentation/classification approach, where the positive association appears to better located around the cheeks compared to the proposed method.

Because of the issues mentioned about COVID-19 applications and the impact of biases from various data sources, I would also suggest the authors consider an in-domain application of disease detection using CheXpert or MIMIC-CXR datasets. Here, the authors could consider a subset of the 14 disease conditions to further compare the performance of different approaches, avoiding the issue of confounding factors introduced by pooling COVID and non-COVID data from different sources.

Other comments:

- The claim that the proposed approach does implicit segmentation is somewhat problematic given that no segmentation outputs are generated
- Result tables are easier to read if the same metric for different methods are reported next to each other, rather than grouping the sub tables by method.
- The very tight line spacing of the manuscript made reading very tiring and difficult. Please consider larger line spacing when preparing a manuscript for review
- The released codebase only contains parts of the code needed to replicate the results. Ideally, the implementations of the baseline methods should be added. For example, I could not verify the implementation of the U-net.

Firstly, we would like to thank the reviewers for their valuable comments, which helped us improve the quality of our manuscript. In this letter, we reply to their concerns and point out how we have changed our paper to address them. We provide a revised manuscript with all modifications marked in red (file Manuscript_R1_Red.pdf).

Reviewer 1:

- 1- *“Conducting segmentation + classification together is not a new concept, and this can be done within multi-task learning, there exists bunch of works like this. There is no unique innovation as opposed to what authors claimed.”*

The previous version of our manuscript did not contextualize our study, leading to a misunderstanding of our objective, methodology, and our architecture’s relation to the state-of-the-art. We apologize for this and thank the reviewer for his/her comment, which prompted us to reformulate a large part of our text (including the entire introduction and abstract). Furthermore, we have tested new benchmark DNNs for comparison, providing a much broader view of the state-of-the-art. We now better contextualize our proposal as a spatial attention mechanism (and the corresponding classifier architecture, the ISNet). Finally, we explain its advantages and novelties in relation to other methodologies, like multi-task learning.

Our study’s novelty is not conducting segmentation and classification together, and we did not expect to suggest that. As the reviewer pointed out, this is not a new concept, and it can be achieved with multi-task learning. Instead, the main novelty of our work is creating the first classifier that, without the need for a segmenter at run-time, reliably ignores the image’s background regardless of its sources of bias, hindering shortcut learning. To the best of our knowledge, this architecture is the first with such capability. Other models that can avoid focus on background bias always require a separate segmenter, as we have demonstrated with new experiments in this revision. Moreover, to exemplify that shortcut learning and background bias are not concerns exclusive to COVID-19 detection, we have included a new application, tuberculosis detection. We have shown that it also presented the issues, and the ISNet was once more the best model for addressing them (Table 3, **lines 313-332**).

The first and second reviewers raised concerns about the capability of a multi-task DNN (performing segmentation and classification) being similar to our proposed architecture. Accordingly, reviewer 2 suggested testing a multi-task learning model as a benchmark: he/she advised using a segmentation network (e.g., U-Net) with an image classification head attached to the bottleneck feature representation. Testing the proposed multi-task model in the COVID-19 detection and in the new tuberculosis detection task, we have observed that it does not behave like an ISNet and cannot replace it. The model achieved good segmentation performance but strongly focused on background bias for classification, failing to generalize to external datasets. There, it produced unacceptable heatmaps (Figure 1) and classification performance (please refer to **lines 211-214** and **325-329** in the manuscript). This behavior was even more evident when using the artificially biased datasets, considering both X-rays and natural images (**lines 234-237, 353-358** and Figure 2). We have demonstrated that multi-task learning cannot avoid background attention when images’ background features correlate with their classes. Therefore, unlike our proposal (the ISNet), multi-task learning is

not usable to prevent shortcut learning caused by background bias. We have formulated a logical explanation for this fact, and please find it in **lines 256-292**, which also analyze the multi-task model results in COVID-19 detection. Furthermore, **lines 51-57** introduce the multi-task DNN, **lines 1191-1207** explain its architecture, **lines 1267-1274** detail its training procedure, **lines 422-438 and 489-492** analyze its heatmaps, bias, and background attention (which confirm the conclusions indicated by the numerical results), and **lines 514-518** draw a conclusion regarding the model.

The multi-task neural network goal is to build a single DNN that acts as classifier and segmenter. Our study has a completely different objective: to present a novel attention mechanism, which creates a classifier (ISNet) that ignores the image's background regardless of the sources of bias contained in it, hindering shortcut learning. The ISNet does not produce a segmentation output, and the multi-task model does not ignore the background during classification if background bias is present, not being able to inhibit shortcut learning. The only methodological similarity between the models is the use of a loss function with multiple terms, a minor aspect of our study. Furthermore, regarding methodology, the ISNet does not employ standard image segmentation, and it is not a segmenter. Instead, it is based on the introduced concept of background relevance minimization in an explanation technique's heatmaps, which allows the creation of an explanation-based spatial attention mechanism. We have improved our introduction to more clearly explain the ISNet's novelties and advantages.

We have also added a fourth benchmark DNN to our study: an attention gated network, suggested by reviewer 2. Like ISNet, it is a classifier with an attention mechanism that should make it focus on relevant image features. However, our experiments have demonstrated that traditional attention mechanisms learn to focus on background features when they are correlated to the samples' classes (Figure 1). Therefore, such mechanisms cannot avoid shortcut learning or substitute the ISNet, showing subpar generalization in tuberculosis and COVID-19 detection (please find the results in **lines 211-214 and 325-329**), the tasks prone to shortcut learning. Again, tests with the artificially biased X-ray and photograph datasets have further confirmed the issues (**lines 237-238, 358-360** and Figure 2). We have concluded that current attention mechanisms can only substitute our model when they are helped by a separate segmenter (or segmentation masks) at run-time, which is not necessary for the ISNet, making our proposal faster and lighter (**lines 509-513**). We explain the reasoning behind this conclusion in **lines 244-255**. Moreover, we introduce the attention gated network in **lines 27-36 and 110-115**, and explain its architecture in **lines 1208-1219**. **Lines 422-445 and 489-492** analyze its heatmaps, and the revealed bias and background attention, confirming the conclusions pointed out by the numerical results. Finally, **lines 509-513** draw a conclusion.

We now summarize the main novel contributions of our paper in its introduction:

“To the best of our knowledge, this is the first study to: create a classifier architecture that, without a segmenter at run-time, reliably ignores the image's background regardless of its sources of bias, hindering shortcut learning (1); present a classifier attention mechanism that learns from segmentation targets but does not require them (or a segmenter) to control the DNN focus at run-time (2); introduce the concept of background relevance minimization in heatmaps produced by an explanation technique, which allowed the creation of an explanation-based spatial attention

mechanism (3); convert and reinterpret the process of layer-wise relevance propagation as neural network layers, allowing backpropagation through LRP (4).”

The implementation of the attention gated networks and multi-task DNN suggested by the reviewers improved our manuscript, by demonstrating the advantages and novelties of the ISNet. Therefore, after testing the models, we are positive that the validity of our claims is clear. Furthermore, we would be happy to test any other architecture that the reviewer believes can be used for the same goal as the ISNet.

2- *“unsure about the importance of the problem. Detection is not really done with x-rays or ct scans, pcr is the one it gives the initial assessment, later doctors use mostly ct to see progress, in some counties x-rays were preferable but the need for such complicated framework (deep learning based segmentation, classification, etc) is not clear. perhaps not needed, maybe only for very subtle change identification only.”*

To better contextualize the COVID-19 detection application, we must say that X-rays were pointed out as an alternative for PCR in moments when there was a shortage of the test, mostly in developing countries with low access to CT. The inherent difficulty of the images’ analysis prompted many studies to suggest artificial intelligence frameworks to help clinicians in the task of COVID-19 detection.

However, our main goal is not to create a PCR alternative. We must clarify that our study’s objective is to propose an efficient and general attention mechanism that is resistant to background bias and hinders shortcut learning. We chose COVID-19 detection as an application because it is a task where these issues are a large concern [1][2]. But our model is not only useful for this application. For example, the necessity of mixing datasets, which can lead to strong background bias, is not exclusive to COVID-19 detection. Indeed, mixing is required whenever researchers need to work with a database that does not contain all desired classes.

We have observed that the largest tuberculosis X-ray database does not have control cases. Therefore, we have also studied the task of tuberculosis detection. In **lines 146-164** of the Introduction, we explain that X-rays are essential for the early detection of TB, that many studies propose DNNs to classify TB-positive and normal cases, and that generalization capability is a known concern for the task. **Section 2.2** explains the new application results. A standard classifier, multi-task DNN, and attention gated network showed exceptionally high performance in a test dataset that was independent and identically distributed (i.i.d.) in relation to the training data, but they failed to properly generalize to an external dataset. As in COVID-19 detection, the ISNet was the best generalizing model, it minimized background attention and hindered shortcut learning, unlike the other tested DNNs (see **lines 434-460**, Figure 1 and Table 3).

The applications in our paper are demonstrative, and we do not aim to create a model that is tailored for X-ray analysis, COVID-19 detection, tuberculosis detection or facial attribute estimation. Instead, our goal is to present a novel and efficient architecture that is useful for image classification problems where background bias is a concern, thus being able to increase the confidence that a classifier's decision rules are based on the image’s region of interest. The ISNet improved

generalization in the tasks that were prone to fostering shortcut learning, tuberculosis and COVID-19 detection. We employed the facial attribute estimation task to demonstrate that our model is also capable of focusing only on the foreground when dealing with RGB natural images (containing diverse foreground shapes and sizes). Unlike the multi-task learning model, the attention gated network, and the DenseNet121, the ISNet could ignore even purposefully added background bias in the natural images and X-rays. We thank you for your comment, and we have now clearly delimited the aim and scope of our work in **lines 543-559** of the Discussion.

- 3- *“heatmap analysis are qualitative. It is well known nowadays that heat maps are showing only where the network learns, they are mostly not explaining anything. Very fragile concepts. Rather, radiographic concepts should be attached to classification so that explainability is satisfied.”*

Heatmaps reveal the input features that are important for classification, and we employ them to find and mitigate background attention. Our paper introduces the idea of minimizing background relevance in heatmaps and proves that this minimization forces a classifier to ignore even very strong background biases, like geometrical shapes correlated to the classes. Thus, we see that the heatmaps in our study were strongly related to the network’s attention. We also observed a strong correlation between our heatmap analysis and the obtained quantitative performance metrics. Specifically, models that failed to generalize and showed subpar performance scores in the external COVID-19 or tuberculosis datasets (DenseNet121, multi-task model, and attention gated network) also had strong background attention in their heatmaps, indicating bias. Additionally, the maps pointed out that these DNNs associated many background features with COVID-19, explaining their low COVID-19 specificity scores. In tuberculosis detection, the attention gated network heatmaps show the strongest association between background features and tuberculosis. Accordingly, the model had the highest number of false positives for the disease. We have explained the correlations between numerical results and heatmaps in **Section 2.5**. Moreover, heatmaps are currently being utilized to discover background bias and shortcut learning (the problems that we address in our manuscript), as exemplified in a Nature Machine Intelligence article about COVID-19 detection [1]. Finally, our heatmaps are produced with layer-wise relevance propagation (LRP), one of the most robust DNN explanation techniques [3].

Unlike state-of-the-art mechanisms, we introduced a novel spatial attention mechanism based on an explanation technique (LRP). A past work qualitatively and quantitatively demonstrated that LRP explanations provide higher resolution heatmaps and more interpretable information than attention mechanisms and their attention heatmaps [4]. Moreover, they revealed additional evidence considered by the classifier to make decisions [4]. Therefore, being based on LRP, our proposed spatial attention mechanism theoretical foundation is more solid than the state-of-the-art. Finally, our proposed mechanism is more transparent than most alternatives: we know it works by hindering attention outside of a region of interest, which is clearly defined by the ground-truth segmentation targets used during training. Typically, spatial attention mechanisms automatically consider the features that most contribute to improving classification performance relevant (refer to **lines 27-36**). Because paying attention to background bias can improve classification loss during training, alternative strategies can learn to focus on the background, as we have observed (**lines 244-255**). We have now clarified these advantages in the last paragraph of the Discussion, **lines 560-569**. We thank

the reviewer for his/her comment as it allowed us to explain the foundational advantages of our methodologies relative to the state-of-the-art more clearly.

The reviewer suggested using radiographic concepts attached to the classification. Although indeed an interesting research path, it goes against the main objective of our paper, which is to introduce a new general methodology to avoid shortcut learning and background attention in diverse applications.

- 4- *“data in COVID19 projects should be carefully approached. imbalance is one problem but there are also other heterogeneity problems existing, one should approach the problem with a care and data curation part should be clear and transparent. Bias in the data (collection, annotation, etc..) should be studied carefully, as is not the case in this study.”*

We thank the reviewer for this comment, which prompted us to improve our data analysis. We have extensively revised the **Datasets section (4.8)** of our manuscript, including a much more complete explanation of the data collection and annotation. Here, we emphasize a few points. Firstly, our training COVID-19 dataset images correspond to all triage and patient monitoring (in sub-intensive and intensive care units) samples collected between March 4th and April 4th, 2020, thus reflecting the variability found in a real clinical scenario. The test COVID-19 dataset was designed to be representative of the disease's first wave in Spain. The classes in our training dataset have similar demographics, reducing possible associated biases. Moreover, the COVID-19 demographic is coherent with demographics of COVID-19+ in Spain and Italy at the time of data collection, indicating that our database is representative [5].

To improve reproducibility, we employ open databases in this study. Furthermore, we chose the largest and most well documented COVID-19 and tuberculosis datasets we could find, increasing the statistical significance of our results. We have analyzed the data limitations and concluded that they do not compromise the study's main objective: to analyze if the ISNet can reduce shortcut learning and improve generalization by ignoring background bias. Please find in **Section 4.8.1** the analysis of the COVID-19 detection data. Specifically, **lines 1032-1054** show the bias and limitations analysis. For tuberculosis detection, **Section 4.8.2** explains the data in detail, and **lines 1125-1139** contain the bias assessment.

We believe that the main source of bias in tuberculosis and COVID-19 databases is mixing different datasets to account for more classes. Indeed, we purposefully built datasets with this characteristic. Studies [1][2] have shown that the practice, which is required to work with the largest COVID-19 or tuberculosis databases (and is also necessary for other classification problems), amplifies background bias (i.e., the existence of background features that are correlated to the classes). Accordingly, the tasks can thoroughly test the ISNet capability of ignoring background bias and mitigating shortcut learning.

We thank the reviewer again for his/her time and valuable comments.

Reviewer 2:

- 1- *“The article presents an image classification approach with implicit segmentation via layer-wise relevance propagation. Evaluation is carried out on two applications, chest X-ray disease detection, and facial attribute recognition.*

Generally, an interesting objective and idea is discussed regarding the use of implicit segmentation in image classification networks. The motivation is clear and sensible, where a classification network should ideally learn what image regions to ignore to be resilient to shortcut learning.

The proposed methodology, however, is not entirely convincing. Any fully-convolutional neural network used for image classification is inherently learning some level of object segmentation/localization encoded in the spatial feature maps. This finding gave rise to the very first segmentation approaches based on fully-convolutional nets (see Long et al. 2014 Fully Convolutional Networks for Semantic Segmentation) derived from image classification backbones. The proposed LRP-based approach seems overly complicated and a sensible baseline to compare with would be to train a segmentation network with an image classification head attached to the bottleneck feature representation. This would enable a straightforward combined segmentation/classification objective, without the overhead of the LRP. There is also considerable work on attention-gated networks for both segmentation and classification, which has not been discussed and may be considered for experimental comparison.”

Firstly, we thank the reviewer for the suggestion, as it can make the novelties and advantages of our work much clearer relative to the state-of-the-art. To better contextualize our model in the context of other studies, we have reformulated a substantial portion of our text, including the introduction. Moreover, we have added the two neural network architectures suggested in this comment (multi-task model and attention gated network) as new baselines. We have shown that they cannot substitute our proposal as they are not capable of avoiding focusing on background bias, nor hindering shortcut learning. Finally, we have added a new application to our study, tuberculosis detection, to show that background bias and shortcut learning are not issues exclusive to COVID-19 detection. In the new task, our model was again the best generalizing DNN, paying no attention to background bias.

We strongly agree with the reviewer that a fully-convolutional neural network used for classification is learning some level of object segmentation. Indeed, this CNN capacity is important to explain our model precision. In section 4.5 we have explicated the precision of our model in light of the concepts introduced by “Long et al. 2014 Fully Convolutional Networks for Semantic Segmentation”. We have drawn parallels between the pyramidal construction and skip connection introduced by “Long et al. 2014” and the ISNet architecture, explaining that analogs are naturally created by the translation of LRP rules to DNN layers.

We also agree that a sensible baseline for our work would be a segmentation DNN with a classification head attached to the bottleneck feature representation. We have implemented the proposed DNN, using a U-Net as the segmentation network. In the manuscript, **lines 51-57** introduce the model, **1191-1207** describe the architecture, and **1267-1274** explain its training procedure. We

chose the U-Net because we used it in the alternative segmentation-classification pipeline (the other baseline), and because it is heavily based on “Long et al. 2014 Fully Convolutional Networks for Semantic Segmentation” [6] (the models share all major characteristics, as being a fully-convolutional DNNs, having skip connections, and transposed convolutions). The first reviewer asked us to use multi-task learning to train a single DNN to perform segmentation (lungs or faces) and classification simultaneously. Accordingly, we trained your suggested new baseline with multi-task learning.

Testing the proposed multi-task model in COVID-19 detection and in the new tuberculosis detection task, we observed that it did not behave like an ISNet and could not replace it. It achieved good segmentation performance, but strongly focused on background bias for classification, failing to generalize to external datasets, thus producing unacceptable heatmaps (Figure 1) and classification performance (**lines 211-214, 325-329** and tables 1 and 3). This behavior was even more evident when using the artificially biased datasets, considering both X-rays and natural images (**lines 234-237, 353-358** and Figure 2). Your proposed baseline has demonstrated that multi-task learning cannot avoid background attention when images’ background features correlate with their classes. Therefore, unlike our proposal (the ISNet), multi-task learning is not usable to prevent shortcut learning caused by background bias. We have formulated a logical explanation for this fact, please find it in **lines 256-292**, which also analyze the multi-task model results in COVID-19 detection. Furthermore, **lines 51-57** introduce the multi-task DNN, **lines 1191-1207** explain its architecture, **lines 1267-1274** detail its training procedure, **lines 422-438** and **489-492** analyze its heatmaps, bias and background attention (which confirm the conclusions indicated by the numerical results), and **lines 514-518** draw a conclusion regarding the model.

The new baseline DNN goal is to build a single model that acts as classifier and segmenter. Our study has a completely different objective: to present a novel attention mechanism, which creates a classifier (ISNet) that ignores the image’s background regardless of the sources of bias contained in it, hindering shortcut learning. The ISNet does not produce a segmentation output, and the multi-task model does not ignore the background during classification if background bias is present, not being able to inhibit shortcut learning. The only methodological similarity between the models is the use of a loss function with multiple terms, a very minor aspect of our study. Furthermore, the ISNet does not employ standard image segmentation, it is not a segmenter. Instead, it is based on the introduced concept of background relevance minimization in an explanation technique’s heatmaps, which allows the creation of an explanation-based spatial attention mechanism. We have realized that the previous version of our manuscript could be improved to better contextualize our architecture and to better explain its advantages, novelties and goals. Therefore, we reformulated our introduction to clarify these aspects.

We also agree with the reviewer when he/she says that attention gated networks should be used for comparison. Indeed, we have contextualized the ISNet as an architecture based on a new spatial attention mechanism. We have included an attention gated network as the fourth benchmark DNN in our study. Like ISNet, it is a classifier with a spatial attention mechanism that should make it focus on relevant image features, which does not require a segmenter/segmentation masks (as the run-time ISNet). However, our experiments have demonstrated that traditional attention mechanisms learn to focus on background features when they are correlated to the samples’ classes. Therefore, such mechanisms cannot avoid shortcut learning or substitute the ISNet, showing subpar generalization in

tuberculosis and COVID-19 detection (**lines 211-214** and **325-329**), the tasks prone to shortcut learning. Again, tests with the artificially biased X-ray and photograph datasets have further confirmed the issues (**lines 237-238, 358-360** and Figure 2). We have concluded that current attention mechanisms can only substitute our model when they are helped by a separate segmenter (or segmentation masks) at run-time, which is not necessary for the ISNet, making our proposal faster and lighter (**lines 509-513**). We explain the reasoning behind this conclusion in **lines 244-255**. Moreover, we introduce the attention gated network in **lines 27-36** and **110-115**, and we present its architecture in **lines 1208-1219**. **Lines 422-445** and **489-492** analyze its heatmaps, and the revealed bias and background attention, confirming the conclusions pointed out by the numerical results. Finally, **lines 509-513** draw a conclusion.

With the new baselines, we expect to have addressed the reviewer's concerns about our methodology. When researchers need to use a dataset that does not contain all classes that they must study, dataset mixing is necessary. In such cases, there is a high risk of background features being correlated to the classes, creating background bias and shortcut learning. To the best of our knowledge, our study is the first to create a classifier architecture that avoids shortcut learning caused by background bias without the need for a segmenter at run-time. We have listed further novelties of our work in the introduction (**lines 102-107**).

- 2- *“Besides the concerns about the methodology, I believe that the structure of the article could be greatly improved. Methods and results are mixed, where segmentation performance is reported within the method description.”*

We thank the reviewer for the comment and have moved the information to the results section. Segmentation results for the U-Nets that we trained are now expressed in **lines 199-201, 266-268, 308-312, 340-341, and 355-358**.

- 3- *“It was very confusing that multiple results for the lung segmentation U-net were reported at different places, that also did not seem to be consistent. It remains unclear to whether the same U-net was used to provide the automated segmentations for the proposed approach as for the baseline using a sequence of segmentation and classification.”*

We must clarify that there are multiple U-Nets. We produced our ground-truth segmentation targets (for the COVID-19 detection and the tuberculosis detection datasets) with a U-Net trained for lung segmentation in another study [7]. We did not train this model and, in the Datasets section, we only report its previously published results. We did not place them in the Results section, because we did not generate the reported performances, and we only employed the model to produce and store segmentation targets for the X-rays. We altered the Datasets section to better explain these aspects, please refer to **lines 1055-1070** and **1121-1125**.

Our alternative segmentation-classification pipeline does not employ the U-Net from [7]. It uses another one, which we trained. Accordingly, we state its performance in the Results section (as explained in the last question). **Lines 1055-1070** elucidate why we did not utilize the U-Net from [7] in the alternative segmentation-classification pipeline. We thank the reviewer for this comment, which improved the organization and clarity of the manuscript.

- 4- *“Conceptually, it is unclear why the latter should perform so much worse than the proposed method that does only learn implicit segmentation. Intuitively, I would have expected the explicit segmentation followed by classification to yield an upper bound of the performance. This somewhat surprising result should be investigated further.”*

The reviewer’s reasoning is very coherent, and we agree that the result is somewhat surprising. The alternative segmentation-classification pipeline performance in COVID-19 detection exposed limitations of the model, which have become even more apparent in the new task of tuberculosis detection. In the new application, the model was again overtaken by the ISNet, and its generalization capability was less promising than in COVID-19 detection. Taking into consideration the alternative model’s numerical results and heatmaps, we have formulated a logical explanation for its limitations, elucidated in **lines 313-324** and **446-460**. We thank the reviewer for requesting this in-depth analysis of the alternative model.

- 5- *“The qualitative results for the CelebA data in Fig. 1 raises some questions about the claims of the paper. In fact, I thought that the heatmaps are much more convincing for the sequential segmentation/classification approach, where the positive association appears to better located around the cheeks compared to the proposed method.”*

Again, the reviewer's comment is coherent, and we have performed a more profound analysis of the models’ heatmaps. The ISNet heatmaps show a more concentrated/sparse attention profile, which can also be observed in the new attention gated network heatmaps. This may indicate that the ISNet tends to focus on fewer, more relevant features inside the region of interest, as the attention gated model is designed to do. This attention profile made the ISNet photograph heatmaps visualization more challenging, as is the case for the attention gated network. However, the ISNet’s performance metrics in all tasks (and the attention gated network result in facial attribute estimation) indicate that this attention profile does not hinder accuracy. Furthermore, the heatmaps confirm the main claim of the ISNet: its capacity to ignore the background. Finally, we also note that visualization of the heatmaps could be improved by utilizing LRP-gamma and LRP-0 rules in our LRP block. However, such rules are not used in the ISNet, since they were not adequate for background relevance minimization. Our LRP procedure is mainly based on the LRP-epsilon rule, whose heatmaps tend to be grainier and sparser. However, these characteristics do not compromise their capacity to reveal background attention, their main objective in our study. The reviewer may refer to figure 10.4 in [8] for examples of heatmaps using different LRP rules. Furthermore, even with the sparsity problem, we can observe red regions over the cheeks in all ISNet heatmaps. We have included this analysis in our results section, please refer to **lines 473-481**.

- 6- *“Because of the issues mentioned about COVID-19 applications and the impact of biases from various data sources, I would also suggest the authors consider an in-domain application of disease detection using CheXpert or MIMIC-CXR datasets. Here, the authors could consider a subset of the 14 disease conditions to further compare the performance of different approaches, avoiding the issue of confounding factors introduced by pooling COVID and non-COVID data from different sources.”*

We thank the reviewer for this comment. We have performed the suggested test, using a subset of the CheXpert database. The ISNet has shown a small test performance increment (0.029 maF1) concerning the baseline DenseNet121 (without segmentation), but confidence intervals of the performance metrics significantly overlapped. Therefore, we cannot claim an accuracy benefit in the proposed in-domain application.

When we use the official CheXpert train and test databases, we evaluate the neural network on a test dataset that is independent and identically distributed (i.i.d.) in relation to the training database. In such cases, ignoring the background (with the ISNet or the alternative classification-segmentation pipeline) can produce two effects. First, especially when a standard DNN would find it difficult to focus on the relevant image features, ignoring the background may help the model pay attention to the critical features, improving accuracy. But the dataset may present background features that are correlated to the classes (i.e., background bias). In this case, ignoring the background can reduce performance on the i.i.d. test dataset, because the model is now ignoring bias that artificially helped classification. Accordingly, models affected by shortcut learning perform well on standard benchmarks (i.e., i.i.d. test datasets), but fail to generalize to real-world scenarios (or out-of-distribution databases) [1][2][9]. We have now added this analysis in **lines 342-348**, and used it to understand our results in the facial attribute estimation task, which is an in-domain application. The results with CheXpert are like the ones in facial attribute estimation, where we found no significant performance increment with the ISNet. However, the facial attribute estimation task has the additional benefit of demonstrating that the ISNet can work with natural images and with a wide variety of regions of interest shapes. Therefore, to avoid increasing the size of our manuscript, we have not included the in-domain CheXpert classification task.

The major objective of the ISNet is to avoid shortcut learning caused by background bias. Therefore, its main benefits cannot be properly demonstrated by an in-domain application. Instead, evaluation with an out-of-distribution (o.o.d.) database is the best way of displaying our proposal's generalization benefits. We selected the COVID-19 and tuberculosis detection applications (with testing on an o.o.d. dataset) because they exemplify how mixing databases can promote background bias and shortcut learning, the problems that our architecture addresses. Moreover, we have evaluated the DNNs in both an i.i.d. and an o.o.d. test dataset for tuberculosis detection, which allowed us to compare the ISNet behavior in both an in-domain and a cross-dataset scenario.

As in COVID-19 detection, we have observed that the largest tuberculosis X-ray database does not have control/healthy cases. Thus, to use it for tuberculosis detection, researchers must also resort to dataset mixing. Therefore, we have studied the task of tuberculosis detection too. In **lines 146-164** of the Introduction, we now explain that X-rays are essential for the early detection of TB, that many studies propose DNNs to classify TB-positive and normal cases, and that generalization capability is a known concern for the task. **Section 2.2** explains the new application results. A standard classifier, multi-task DNN, and attention gated network showed exceptionally high performance in a test dataset that was independent and identically distributed (i.i.d.) in relation to the training data, but they failed to properly generalize to an external, o.o.d. dataset. As in COVID-19 detection, the ISNet was once more the best generalizing model, it minimized background attention and hindered shortcut learning, unlike the other tested DNNs (see **lines 434-460**, Table 3 and Figure 1). Showing a behavior coherent with a reduction in shortcut learning [9], the ISNet performance on the i.i.d.

tuberculosis evaluation database was worse than a standard classifier, while it was the best performing model on the o.o.d. dataset (**lines 329-332**).

We thank the reviewer for his/her comment. With the manuscript revision, we hope to have better elucidated the advantages of our model, explaining its behavior with both in-domain scenarios and o.o.d. datasets. Additionally, we expect to have clarified why we chose tasks that are prone to shortcut learning and utilized evaluation on o.o.d. databases.

7- *“The claim that the proposed approach does implicit segmentation is somewhat problematic given that no segmentation outputs are generated”*

We have better contextualized the term implicit segmentation. Our model is based on background relevance minimization in explanation heatmaps, a new concept, which makes a classifier focus on a region of interest that it learns from segmentation targets. Our proposal can be mostly seen as a novel spatial attention mechanism for classifiers. The ISNet is not a segmenter, as it does not produce segmentation outputs. We created the term implicit segmentation to refer to the ISNet ability to implicitly distinguish between the input image foreground and background, and focus its attention on the foreground only, as can be seen in its heatmaps (Figures 1 and 2). We have explained the term’s meaning in detail in our revised introduction, please refer to **lines 71-79**. To avoid any misinterpretation, we have removed all mentions of the expression “implicit segmentation” before the aforementioned explanation, including in our title and abstract.

8- *“Result tables are easier to read if the same metric for different methods are reported next to each other, rather than grouping the sub tables by method”*

We thank the reviewer for the suggestion and we modified the tables accordingly.

9- *“The very tight line spacing of the manuscript made reading very tiring and difficult. Please consider larger line spacing when preparing a manuscript for review”*

We are sorry for the inconvenience, and we have increased the line spacing.

10- *“The released codebase only contains parts of the code needed to replicate the results. Ideally, the implementations of the baseline methods should be added. For example, I could not verify the implementation of the U-net.”*

We have added the source code for the baseline DNNs in the released codebase. Moreover, the complete code is available to the reviewers, including demos, reproduction instructions, the trained models, and details such as the training loop, statistical evaluation, and image processing (which are also described in the manuscript). Please find the non-public code at <https://drive.google.com/drive/folders/1pSkn3Nf1W9U-yrWRZj6fn8JtLxEpKexh?usp=sharing>. We shall include this detailed code in the released codebase upon publication.

We thank the reviewer again for his/her time and valuable comments.

References:

- [1] López-Cabrera, J., Portal Diaz, J., Orozco, R., Lovelle, O. & Perez-Diaz, M. Current limitations to identify covid-19 using artificial intelligence with chest x -ray imaging (part ii). the shortcut learning problem. *Heal. Technol.* 11, DOI: 10.1007/s12553-021-00609-8 (2021).
- [2] DeGrave, A. J., Janizek, J. D. & Lee, S.-I. Ai for radiographic covid-19 detection selects shortcuts over signal. *Nat Mach Intell* 3, 610–619, DOI: 10.1038/s42256-021-00338-7 (2021).
- [3] Eitel, F. & Ritter, K. Testing the robustness of attribution methods for convolutional neural networks in mri-based alzheimer’s disease classification. In Suzuki, K. et al. (eds.) *Interpretability of Machine Intelligence in Medical Image Computing and Multimodal Learning for Clinical Decision Support*, 3–11 (Springer International Publishing, Cham, 2019)
- [4] Sun, J., Lopuschkin, S., Samek, W. & Binder, A. Explain and improve: Lrp-inference fine-tuning for image captioning models. *Inf. Fusion* 77, 233–246, DOI: <https://doi.org/10.1016/j.inffus.2021.07.008> (2022).
- [5] de la Iglesia Vayá, M. et al. Bimcv covid-19+: a large annotated dataset of rx and ct images from covid-19 patients (2020). 2006.01174.
- [6] Ronneberger, O., P.Fischer & Brox, T. U-net: Convolutional networks for biomedical image segmentation. In *Medical Image Computing and Computer-Assisted Intervention (MICCAI)*, vol. 9351 of LNCS, 234–241 (Springer, 2015).
- [7] Bassi, P. R. A. S. & Attux, R. Covid-19 detection using chest x-rays: is lung segmentation important for generalization? (2021). 2104.06176.
- [8] Montavon, G., Binder, A., Lopuschkin, S., Samek, W. & Müller, K.-R. Layer-wise relevance propagation: An overview. In *Explainable AI: Interpreting, Explaining and Visualizing Deep Learning*, 193–209 (Springer International Publishing, 2019). Available at: https://www.researchgate.net/publication/335708351_Layer-Wise_Relevance_Propagation_An_Overview
- [9] Geirhos, R. et al. Shortcut learning in deep neural networks. *Nat. Mach. Intell.* 2, 665–673, DOI: 10.1038/s42256-020-00257-z (2020).

Reviewers' Comments:

Reviewer #2:

Remarks to the Author:

The authors have prepared a very comprehensive and solid revision. I appreciate the careful discussion and engagement with the reviewers' suggestions. The key changes include the comparison with important baselines such as attention-gated networks and a multi-task learning approach.

The revision is quite extensive and does address most of my previous concerns. The text changes are overall helpful in clarifying the contributions and better supporting the claims. The experiments add significant value. I am now much more convinced that the proposed method can lead to more robust image classification.

My main concern is the exclusion of the new CheXpert results. I am not entirely convinced that these should be excluded. The authors state that these results are somewhat 'negative' in a sense that no significant improvement could be found for the proposed method. It is argued that this may be due to the i.i.d. nature of the test set. This could be tested by using an external chest X-ray dataset such as MIMIC-CXR. In any case, I believe reporting such negative findings is important as it may provide valuable information about the possible limitation and use case for the proposed method.

Chest X-ray disease detection has been a prime example in the past where deep learning methods have exploited shortcuts for making predictions (see DeGrave et al. 2021 <https://www.nature.com/articles/s42256-021-00338-7>). Given that ISNet is specifically designed for this issue, I feel it would be good to better investigate this. My suggestion would be to add MIMIC-CXR as an external test set and report the results in the supplementary material.

Reviewer #3:

Remarks to the Author:

Many parts of the abstract and introduction in the Revised paper have been revised to answer the reviewers' questions.

However, it did not provide a clear answer on what differs from the existing segmentation + classification.

As an additional test, the authors included TB detection. The explanation that the proposed method in Table 3 is the best model does not explain everything. It's hard to find an explanation as to why you're showing these results.

Only the proposed ISNet model

"testing the proposed multi-task model in the COVID-19 detection and in the new tuberculosis detection task, we have observed that it does not behave like an ISNet and cannot replace it." It isn't easy to accept such an explanation.

"The only methodological similarity between the models is the use of a loss function with multiple terms, a minor aspect of our study. "

In addition, it is difficult to accept that a loss function that minimizes multiple terms is similar but plays an entirely different role.

This is a critical claim and very poorly explained.

Reviewer #4:

Remarks to the Author:

My review mainly focuses on the authors' responses to concerns by Reviewer #1. Even after revision, there are still some major issues remaining.

1. Although authors tried to clarify difference between their method and multitasking networks, actually many similar works have already proposed. For example, two works in [1][2] proposed a very similar idea, by not only building a multitask framework but also using the segmentation mask to guide the network attention. Authors are encouraged to use these two methods as the benchmark for comparison. Meanwhile, it will be also good if comparing the LRP explanation technique with CAM/Grad-CAM technique when considering these two works as the benchmark for comparison. Note that CAM/Grad-CAM is more widely used to explain the attention for deep models.

[1] Li, Kunpeng, et al. "Tell me where to look: Guided attention inference network." Proceedings of the IEEE Conference on Computer Vision and Pattern Recognition. 2018.

[2] Ouyang, Xi, et al. "Learning hierarchical attention for weakly-supervised chest X-ray abnormality localization and diagnosis." IEEE Transactions on medical imaging 40.10 (2020): 2698-2710.

2. Although COVID-19 detection is a good example for demonstrating the efficiency and general attention mechanism of proposed method, it is important to indicate that this application is not clinically significant. In early stage of COVID-19, it is critical to build a deep network for detecting COVID-19 from chest x-ray images, since COVID-19 causes severe lung infection. But, current COVID-19 mostly does not cause any lung infection.

3. It is important to evaluate whether the learned attention regions by the classification network match well with the abnormal regions that radiologists found. To do this, authors can provide several testing cases for some experienced radiologists to indicate the abnormal regions, and then compare with the attention findings by classification network. It is important to prove that the attention findings match well with actual clinical findings.

We thank the reviewers very much for their time and valuable comments. In this letter, we reply to their concerns and point out how we have changed our paper to address them. We provide a revised manuscript with all modifications marked in red (file Manuscript_R2_Red.pdf). Our novel algorithms are publicly available at <https://github.com/PedroRASB/ISNet>. We updated the repository with the source code for the two new benchmark DNNs. Moreover, a more complete code (with demos and trained network parameters) is available to the reviewers at the link below. We shall include this detailed code in the released codebase upon publication.

<https://drive.google.com/drive/folders/1Zi212Nnmhp-ukdG0dNSCcalTQi9K-BfD?usp=sharing>.

Using the link above the reviewers can also find Supplementary Data 1 (high-resolution images, radiologist's annotations, and ISNet heatmaps).

Reviewer #2:

"The authors have prepared a very comprehensive and solid revision. I appreciate the careful discussion and engagement with the reviewers' suggestions. The key changes include the comparison with important baselines such as attention-gated networks and a multi-task learning approach.

The revision is quite extensive and does address most of my previous concerns. The text changes are overall helpful in clarifying the contributions and better supporting the claims. The experiments add significant value. I am now much more convinced that the proposed method can lead to more robust image classification.

My main concern is the exclusion of the new CheXPert results. I am not entirely convinced that these should be excluded. The authors state that these results are somewhat 'negative' in a sense that no significant improvement could be found for the proposed method. It is argued that this may be due to the i.i.d. nature of the test set. This could be tested by using an external chest X-ray dataset such as MIMIC-CXR. In any case, I believe reporting such negative findings is important as it may provide valuable information about the possible limitation and use case for the proposed method.

Chest X-ray disease detection has been a prime example in the past where deep learning methods have exploited shortcuts for making predictions (see DeGrave et al. 2021 <https://www.nature.com/articles/s42256-021-00338-7>). Given that ISNet is specifically designed for this issue, I feel it would be good to better investigate this. My suggestion would be to add MIMIC-CXR as an external test set and report the results in the supplementary material."

First, we thank the reviewer very much for the kind comments regarding our previous revision. Moreover, we are grateful for the current suggestion. Indeed, the inclusion of the CheXPert classification task was important to better delimit the use case of the novel methodology and to show that different X-ray datasets may present diverse challenges. Please find the new application in Appendix B. All manuscript modifications are highlighted in red.

As previously suggested by the reviewer, we have considered a subset of the CheXPert database due to the high computational cost of the experiments using the large multi-label dataset. With the subset, we have trained the ISNet and 5 other benchmark deep neural networks (DNNs). Following the

reviewer's suggestion, we have tested the models on a corresponding subset of the MIMIC-CXR test database (considering the same classes on both datasets).

Surprisingly, as we have previously observed for the i.i.d. evaluation, the performances on the o.o.d. MIMIC database were similar for all the tested DNNs. All macro-average AUC (Area Under the ROC Curve) scores presented some overlap in their 95% confidence intervals (Table 5). This result is thoroughly analyzed in Appendix B.

In summary, all images in CheXPert come from the same hospital (Stanford University Hospital, California, United States of America). This is not the case for large COVID-19 or tuberculosis detection databases, which require dataset mixing for neural network training. Indeed, as previously explained, we chose the two tasks because they exemplify real-world applications that usually rely on dataset mixing. In such mixed databases, images from different classes have diverse sources (hospitals), which may present unique background characteristics, causing background bias and consequent shortcut learning.

The results in Appendix B show that background bias and shortcut learning are much less apparent for CheXPert (a large single-source X-ray dataset). Proving the existence of less shortcut learning, all the DNNs in Appendix B showed a relatively small performance gap when comparing i.i.d. (CheXPert) and o.o.d. (MIMIC) evaluation. For example, the average AUC gap for the standard classifier (DenseNet121) was only 0.076. The paper mentioned by the reviewer shows that such gaps are much larger for COVID-19 detection, and the same DenseNet121 has an AUC gap above 0.42 in tuberculosis detection. The reduced shortcut learning is a clear signal that CheXpert has less background bias.

Further confirming that background bias is not a main issue for CheXPert training, the ISNet and the segmentation-classification pipeline (U-Net followed by DenseNet121) did not surpass the standard DenseNet121 in Appendix B. Meanwhile, our previous experiments show that the two models are the most promising for avoiding attention to background features (as can be clearly seen in Figure 1a).

The CheXPert experiment was important to show that background bias (and the consequent shortcut learning) is not a problem intrinsic to X-ray classification but is mostly caused by dataset mixing, which is commonly necessary for X-ray classification tasks (please refer to Appendix B.1). When training in large single-source databases, o.o.d. generalization gaps are much smaller, as can also be seen in other studies [1]. Moreover, these works concluded that such gaps could be caused by factors other than shortcut learning, such as label shift between datasets [1].

We have concluded Appendix B (last paragraph in the manuscript) by clearly defining the ISNet best use case: to reduce attention to background bias and improve o.o.d. generalization when training datasets present background bias. We have also explained that such bias is demonstrated by large gaps between i.i.d. and o.o.d. evaluation, along with strong attention to background features in explanation heatmaps (as seen in Figures 1 and 2). Finally, we have improved our introduction and abstract to better delimit the ISNet use case (lines 28-29).

As a final note to the reviewer, we will briefly explain other changes in this revision. We have improved the ISNet heatmap loss function (section 4.3). We have discovered that the previous loss indirectly

prompted an overall reduction of the heatmap values, which produced a concentrated and sparse attention profile inside the images' region of interest, as the reviewer previously noticed. Such a profile did not hinder accuracy, as we had explained. However, it made heatmap interpretation more challenging. The new loss function ensures that the region of interest contains a more natural attention pattern, which is more in line with human reasoning (please refer to the ISNet smile heatmap in Figure 2 for a clear example). We have also changed the normalization strategy used to display the heatmaps in the paper (lines 469-470), improving their visualization. Due to the loss function modification, we have re-trained the ISNet and updated its results. They have improved for the COVID-19 detection task and are within the margin of error for the other applications. Following the new normalization strategy, we have updated the heatmaps in Figure 1 and 2. Moreover, now every heatmap displayed in the figures is available in high-resolution in the supplementary material (Supplementary Data 1, available at <https://drive.google.com/drive/folders/1Zi212Nnmhp-ukdG0dNSCcaITQi9K-BfD?usp=sharing>), along with the radiologist's annotations (analyzed in Appendix A) and the figure analysis (Supplementary Note 1).

[1] Cohen, J. P., Hashir, M., Brooks, R. & Bertrand, H. On the limits of cross-domain generalization in automated x-ray prediction. In Arbel, T. et al. (eds.) Proceedings of the Third Conference on Medical Imaging with Deep Learning, vol. 12 of Proceedings of Machine Learning Research, 136–155 (PMLR, 2020). Available at: <http://proceedings.mlr.press/v121/cohen20a/cohen20a.pdf>

Reviewer #3:

1- "Many parts of the abstract and introduction in the Revised paper have been revised to answer the reviewers' questions. However, it did not provide a clear answer on what differs from the existing segmentation + classification"

We thank the reviewer for his/her comment and apologize for the lack of clarity. We have improved the manuscript's Introduction and Results section to address the mentioned concerns (please refer to lines 84-106 and 313-351). All manuscript modifications are highlighted in red. Below, we will respond to the reviewer's comment in more detail.

We will start clarifying the differences between our model and multi-task learning for segmentation + classification from a methodological point of view. Existing multi-task learning for segmentation + semantic classification corresponds to minimizing two loss functions, one for each task (classification and segmentation). Both functions compare standard neural network outputs to their respective targets (classification labels or segmentation masks). The proposed ISNet solves only one task, classification. Our methodology also employs a classification loss, as we need to minimize classification error. However, our second cost function does not optimize a standard DNN (Deep Neural Network) segmentation output. Indeed, our neural network does not even produce such segmentation outputs. The second cost function is the new "heatmap loss," which takes LRP (Layer-wise Relevance Propagation) explanation heatmaps as inputs and compares them to segmentation targets. Thus, **instead of optimizing a semantic segmentation output (like multi-task learning), we are optimizing the LRP explanations of a classifier**, forcing its attention to be contained inside a region of interest. The second loss objective is to avoid bias in the classification task. We are the first to create an

attention mechanism based on the optimization of LRP explanations. On the other hand, as shown by our experiments, standard segmentation + classification with multi-task learning **cannot be regarded as an attention mechanism**. I.e., the optimization of its second loss function did not avoid classification bias (Tables 1 and 3, lines 259-262, 282-289, and 414-423). An in-depth explanation for this behavior is provided in lines 313-340, which we have modified for better clarity.

Moreover, the LRP block must not be confused with the segmentation branch in a multi-task DNN. The segmentation branch is composed of standard DNN layers; it is a trainable structure that produces a standard segmentation mask. It has independent trainable parameters. The ISNet does not have a segmentation branch. The LRP block is a temporary structure used only during training. It implements the LRP procedure, which backpropagates a quantity called relevance through all classifier layers, creating high-resolution heatmaps that explain the classifier decisions (Section 4.1). Thus, the LRP block does not have independent trainable parameters, it only utilizes the classifier's parameters. The LRP heatmaps are closely related to the classifier attention pattern, unlike the multi-task DNN segmentation output (as our experiments confirmed, please refer to lines 321-323 and 414-419).

COVID-19 and tuberculosis detection in chest X-rays are two tasks that are prone to shortcut learning due to background bias. This can be observed in the poor generalization performance of a standard DenseNet121 classifier (Tables 1 and 3). Furthermore, background attention is clearly visible in the DenseNet121 X-ray heatmaps (Figure 1). Analyzing the results in Tables 1 and 3, we observe that the multi-task model did not surpass the standard classifier (DenseNet121), and its heatmaps in Figure 1 explain the reason for this behavior: it did not prevent classifier background attention and shortcut learning. We further confirmed such results using artificially biased datasets (lines 282-289 and 414-423).

Accordingly, we observe that standard segmentation + classification with multi-task learning cannot be used to hinder a classifier's attention to background bias. This is a major difference between the presented ISNet and standard multi-task learning for segmentation + classification: the two methodologies have **fundamentally different objectives and use cases**. Our model is used to avoid classifier attention to background bias, thus hindering shortcut learning. Standard segmentation + classification fails at this task. Instead, its objective is to produce classification outputs and segmentation masks simultaneously. On the other hand, our methodology does not produce such segmentation masks.

Segmentation + classification may also be created without multi-task learning, by using a pipeline with two neural networks trained separately. This is represented by another alternative baseline in our manuscript: the U-Net+DenseNet121. This pipeline can be used for the objective of avoiding attention to background bias. However, our methodology proved to be more accurate, faster, and lighter than this traditional method (Section 2.4). Moreover, methodological differences are clear: the traditional method does not use optimization of LRP explanations, and it uses two separately trained neural networks (our model uses one DNN).

In summary, very few papers optimize a classifier's explanations to control attention, as we did. The ones that do utilize gradCAM instead of LRP. However, GradCAM is fundamentally different from LRP.

GradCAM heatmaps are based on a linear combination of the feature maps from a specific DNN layer, while LRP produces a high-resolution heatmap by back-propagating a signal through all DNN layers, capturing high-level semantics and precise localization details. We have included Guided Attention Inference Networks (GAIN) as a benchmark DNN in our manuscript (lines 125-148). Like the other five DNNs used for comparison in our study, GAIN's results could not match the ISNet when training datasets presented background bias, because the model could not avoid background attention (Tables 1 and 3, lines 292-300, 426-430, and Figures 1 and 2). Furthermore, GAIN's results have demonstrated that GradCAM explanations are unreliable for the creation of attention mechanisms, as we explained in Section 2.5.1. This unreliability was confirmed in Appendix B, where we have tested another DNN based on GradCAM optimization, HAM (Hierarchical Attention Mining). Like GAIN, HAM could not avoid attention to background bias.

Overall, after testing the multiple baselines suggested by the reviewers, we can still state that our study is the first to create a classifier architecture that, without a separate segmenter at run-time, reliably ignores the image's background regardless of its sources of bias, hindering shortcut learning. We hope that our response and revision can better explain the differences between the ISNet and the state-of-the-art, and we would be happy to address any specifics that are still unclear.

2- "As an additional test, the authors included TB detection. The explanation that the proposed method in Table 3 is the best model does not explain everything. It's hard to find an explanation as to why you're showing these results."

In the comments of reviewer #1, who is now absent, we observed a very strong focus on the COVID-19 detection task. Therefore, we believe that the first version of our manuscript did not clearly express the idea that we are presenting a general methodology to prevent classifier attention to background bias. It is not a DNN architecture tailored for a specific application. We use the applications in our manuscript to demonstrate the new model's capabilities to avoid attention to background bias. Accordingly, we have introduced the tuberculosis classification task to exemplify that COVID-19 detection is not the only scenario where standard classifiers are prone to focusing on bias and presenting shortcut learning. Thus, the task demonstrates another situation in which the presented ISNet is beneficial, highlighting the applicability of the method to problems beyond COVID-19 detection. We have included lines 391-393 to highlight this purpose.

Moreover, the tuberculosis detection task was crucial for us to better understand sources of bias in the standard segmentation-classification pipeline (lines 513-528). These findings were important to answer comments made by reviewer #2.

3- "testing the proposed multi-task model in the COVID-19 detection and in the new tuberculosis detection task, we have observed that it does not behave like an ISNet and cannot replace it." It isn't easy to accept such an explanation."

We are sorry for the unclear explanation, and we thank the reviewer for bringing it to our attention. As we have elucidated in the answer to the reviewer's first comment, the ISNet reduced attention to background bias in COVID-19 and TB detection, the tasks that were naturally prone to such attention. On the other hand, the multi-task model could not reduce the focus on background bias. Background

attention is clearly visible in the model's explanations (Figures 1 and 2), and we proved that it can strongly influence the network's decisions (lines 282-284, and 414-416). Accordingly, the multi-task model cannot hinder shortcut learning, thus performing worse than the ISNet in the out-of-distribution tests for tasks prone to background bias (Tables 1 and 3). For this reason, multi-task learning cannot replace the ISNet in its main use case: to be an attention mechanism that effectively avoids classifier background bias attention, hindering shortcut learning. We have better contextualized and explained the sentence you mentioned; please find it in lines 330-332.

4- "The only methodological similarity between the models is the use of a loss function with multiple terms, a minor aspect of our study. " In addition, it is difficult to accept that a loss function that minimizes multiple terms is similar but plays an entirely different role. This is a critical claim and very poorly explained."

We apologize for the unclear explanation, and we expect the modifications made to lines 84-106 to have better explained these aspects, especially considering the text in lines 313-351. Moreover, we also hope that our detailed answer to the reviewer's first comment could shed some light on this lack of clarity.

Naturally, both the ISNet and the multi-task model have one loss term whose function is to minimize classification error (e.g., the standard cross-entropy loss), as is necessary for training virtually any supervised classifier. However, the ISNet's second loss term (the heatmap loss) and the multi-task model's second loss (the segmentation loss) do indeed play entirely dissimilar roles. The heatmap loss takes as input the classifier LRP explanation. The optimization of such explanations has the objective of constraining the classifier's attention to stay inside a region of interest. Meanwhile, in multi-task learning, the segmentation loss optimizes the model's segmentation branch outputs to match segmentation targets. Unlike the ISNet, this training procedure does not constrain the multi-task model's classification attention to a region of interest (please refer to Figures 1 and 2, lines 501-505, and lines 548-553). Thus, it cannot avoid attention to background bias and hinder shortcut learning, as the ISNet can. This capability explains the ISNet's superior performances in the tasks with mixed datasets, which contain background bias (Tables 1 and 3). Due to the losses' different purposes, background bias affects the decisions and classification scores of the multi-task model, but it does not affect the ISNet's (lines 277-291 and 410-425). In summary, the multi-task DNN segmentation loss' role is to produce segmentation masks; meanwhile, the ISNet heatmap loss role is to act as an attention mechanism, optimizing the classifier attention pattern to avoid focus on background bias. Our experiments prove that the multi-task DNN segmentation loss is not useful for controlling classification attention.

We thank the reviewer again for the comments, which greatly improved the clarity of our paper.

Reviewer #4:

1- "Although authors tried to clarify difference between their method and multitasking networks, actually many similar works have already proposed. For example, two works in [1][2] proposed a very similar idea, by not only building a multitask framework but also using the segmentation mask to guide the network attention. Authors are encouraged to use these two methods as the benchmark for

comparison. Meanwhile, it will be also good if comparing the LRP explanation technique with CAM/Grad-CAM technique when considering these two works as the benchmark for comparison. Note that CAM/Grad-CAM is more widely used to explain the attention for deep models.

[1] Li, Kunpeng, et al. "Tell me where to look: Guided attention inference network." Proceedings of the IEEE Conference on Computer Vision and Pattern Recognition. 2018.

[2] Ouyang, Xi, et al. "Learning hierarchical attention for weakly-supervised chest X-ray abnormality localization and diagnosis." IEEE Transactions on medical imaging 40.10 (2020): 2698-2710."

Firstly, we are incredibly grateful for the reviewer's suggestions. The proposed comparisons certainly improved the quality of our paper and better contextualized our model. With the guidance of the reviewer's comment, the manuscript now explains in detail why six alternative DNNs could not match the ISNet's capability of controlling attention and avoiding focus on background bias. Besides displaying our architecture's differences and advantages in relation to the state-of-the-art, we believe that the explanations may help future research address the indicated weaknesses. All manuscript modifications are highlighted in red.

At first glance, our proposed neural network (ISNet) may look similar to those introduced by the two papers the reviewer mentioned. Both models also optimize loss functions based on neural network explanations. However, both are based on GradCAM explanations, while the ISNet is the first attention mechanism based on Layer-wise Relevance Propagation (LRP). We must clarify that LRP and GradCAM are fundamentally different methodologies. While GradCAM produces heatmaps with a linear combination of the activations from one DNN layer, LRP produces high-definition heatmaps by back-propagating a quantity called relevance. The process captures information from all DNN parameters and activations, embedding both high-level semantics and precise spatial information in the heatmaps (Section 4.1). Thanks to the reviewer's suggestion, our new experiments reveal fundamental weaknesses in the GradCAM procedure, which undermine its ability to detect and reduce background attention. Such weaknesses are not shared by LRP, explaining the ISNet's superior capability of ignoring background bias. Indeed, in all experiments where background bias was verifiable, the ISNet's generalization performance strongly surpassed GAIN's and HAM's (and the other 4 baseline DNNs'). An in-depth analysis of the GradCAM's flaws is presented in the new Section 2.5.1.

We have tested the two mentioned neural networks, Guided Attention Inference Networks (GAIN) [1], and Hierarchical Attention Mining models (HAM) [2]. The main difference between the methods is that GAIN's losses are general-purpose, while HAM's are tailored for the task of X-ray classification in datasets with available lesion annotations (bounding boxes).

HAM is not very adequate for our applications of COVID-19 and TB detection. Firstly, our previous experiments pointed out that attention mechanisms that did not consider segmentation targets were ineffective against background bias (lines 301-312). The only HAM loss function that considers segmentation targets (AMSE loss) is based on lesion annotations, not lung masks. Such annotations are not available for large COVID-19 or tuberculosis databases, and the utilization of such large databases were important for the statistical significance of our study, as previously requested by the editor.

Secondly, HAM's other GradCAM-based losses' objective is to make the DNN attention for the class "not normal" (i.e., some anomalies present in the X-ray) coherent with the attention for the specific X-ray abnormalities. If both attention patterns focus on background bias, these losses (attention bound loss and attention union loss) would be low. Thus, the functions are not able to penalize background attention. Moreover, the losses were designed for multi-label datasets presenting multiple classes, such as CheXPert.

For these reasons, we cannot implement HAM for the applications in the main text of our study. However, we have included a new Appendix (B), which deals with CheXpert classification, as requested by reviewer #2. Thus, we have implemented GAIN in the three applications in the main text (COVID-19 detection, tuberculosis detection, and facial attribute estimation), and HAM in Appendix B.

We have not implemented GAIN in Appendix B because training the model on CheXPert would have taken an exceedingly long time. CheXPert classification is a large multi-label classification problem, greatly increasing GAIN's training time (see lines 443-446). Moreover, the model's training losses were unstable with the learning rate being used with the other DNNs in the appendix, considering a mixed-precision implementation. Ditching mixed precision or reducing the learning rate would have made training time unreasonable (even considering multiple GPUs). Please find more details about the included HAM and GAIN models in lines 125-148, 162-163, 1430-1452, and Appendix B.

In the tasks in which demonstrated background bias (COVID-19 and TB detection), GAIN could not improve o.o.d. test accuracy. Unlike the ISNet, its performance did not significantly surpass the standard classifier (DenseNet121). These results are shown in Tables 1 and 3. Supporting the numerical results, LRP heatmaps for GAIN clearly show background attention (Figure 1). As we did for the other DNNs, we have also tested GAIN with the datasets artificially biased with polygons in their backgrounds. Unlike the ISNet, GAIN could not ignore the polygons, and its classification performance was affected by the artificial bias (lines 292-300, and 426-430). Again, the quantitative change in classification results supported GAIN's LRP explanations, which clearly showed attention to the polygons (Figure 2). However, GAIN's GradCAM heatmaps did not display attention over bias (Figure 1 and 2). We have analyzed the reasons for this discrepancy in detail (please refer to the new Section, 2.5.1), and we have concluded that GradCAM has fundamental flaws. Especially when the GradCAM heatmaps are directly optimized (GAIN), such weaknesses allow the creation of unfaithful explanations, which do not show background attention, while background features clearly influence the classifier outputs (lines 292-300, and 426-430).

In summary, our analysis of GAIN's results (Section 2.5.1) shows that GradCAM is not reliable for the creation of attention mechanisms. Meanwhile, ISNet's performances indicate that LRP is a much more trustworthy foundation for creating attention mechanisms that avoid focus on background bias. The ISNet classification results were not affected by artificial bias (lines 277-280, and 410-412). Moreover, its external (o.o.d.) test accuracy strongly surpassed the other 5 baselines in the tasks presenting natural background bias (Table 1 and Table 3).

Following the reviewer's suggestion, we now display GradCAM explanation heatmaps for the ISNet and GAIN (Figure 1 and Figure 2). The ISNet's GradCAM heatmaps do not show background attention.

Moreover, they agree with the ISNet's LRP explanations, considering that GradCAM has less resolution and worse class selectivity (please refer to lines 476-485).

In Appendix B, we explain that CheXPert classification is not strongly affected by background bias. Unlike the COVID-19 and TB datasets, CheXPert was constructed from a single source hospital (Stanford University Hospital, California, United States of America). For the dataset, common classifiers show more natural focus on the lungs, the performance gap between i.i.d. and o.o.d. testing is much smaller than for COVID-19 and TB detection, and the standard segmentation-classification pipeline could not improve results on the task. These factors indicate much less background bias in CheXPert. For this reason, as in facial attribute estimation, none of the implemented neural networks could significantly surpass the standard DenseNet121 classifier. Accordingly, HAM and ISNet also produced comparable results.

Due to their lack of lesion annotations, we could not test HAM in the datasets naturally presenting background bias (COVID-19 and tuberculosis). Thus, we have added artificial background bias to the CheXPert images (lines 1876-1912). Like in facial attribute estimation (lines 410-430), we have included polygons in the figures' corners, and associated each polygon to a possible label. After training HAM on the biased dataset, we have tested it on MIMIC. When we added the artificial bias to the test samples, HAM macro-average AUC score was 0.951 +/-0.009. However, after removing the polygons, the score dropped to 0.627 +/-0.032. A similar drop was observed for a common DenseNet121. The large AUC gap proves that HAM paid much attention to the artificial bias. The ISNet did not. With the same experiment, our proposed model's macro-average AUC was 0.7 +/-0.031 in the presence of the polygons, and it remained 0.7 +/-0.031 after removing them. Thus, background bias could not influence the ISNet.

The ISNet LRP and GradCAM heatmaps for the biased CheXPert images show no attention to the polygons (Figure 8). Meanwhile, HAM's LRP explanations reveal strong focus on the background bias, and are similar to the standard DenseNet121 LRP heatmaps. As seen with GAIN, HAM's GradCAM explanations hid the attention to the polygons (Figure 8). For this reason, they could not explain how the background bias strongly influenced HAM's AUC, thus being unrealistic explanations of the model's decision rules. Overall, HAM and GAIN similar results and behavior indicate that fundamental GradCAM weaknesses are the cause of the different architectures' failure to avoid bias attention (Section 2.5.1).

2- "Although COVID-19 detection is a good example for demonstrating the efficiency and general attention mechanism of proposed method, it is important to indicate that this application is not clinically significant. In early stage of COVID-19, it is critical to build a deep network for detecting COVID-19 from chest x-ray images, since COVID-19 causes severe lung infection. But, current COVID-19 mostly does not cause any lung infection."

The reviewer certainly comprehended the purpose of the COVID-19 related application in the manuscript, which is to demonstrate the efficiency of the proposed attention mechanism in a scenario where background bias is clearly present [1]. We certainly agree with the reviewer's comment, and we have included the suggested observation in the Discussion Section. Please find it in lines 715-720.

[1] DeGrave, A. J., Janizek, J. D. & Lee, S.-I. Ai for radiographic covid-19 detection selects shortcuts over signal. Nat Mach Intell 3, 610–619, DOI: 10.1038/s42256-021-00338-7 (2021).

3- “It is important to evaluate whether the learned attention regions by the classification network match well with the abnormal regions that radiologists found. To do this, authors can provide several testing cases for some experienced radiologists to indicate the abnormal regions and then compare with the attention findings by classification network. It is important to prove that the attention findings match well with actual clinical findings.”

We thank the reviewer very much for this comment. We have started an international collaboration with Dr. Dertkigil, professor of radiology at the Department of Radiology of the University of Campinas (UNICAMP), Campinas, SP, Brazil. The radiologist has been practicing medicine at the Clinics Hospital of the University of Campinas since 2009 and is the Director of the hospital’s Radiology Service. For his contribution, Dr. Dertkigil has been included as an author.

As requested, we have provided him with several X-rays, and we have requested the localization of lung lesions caused by the diseases the image portrayed. We randomly selected 30 images from the o.o.d. test databases to be analyzed, 10 per class (COVID-19, pneumonia, and tuberculosis). Before completing his task, the radiologist had no access to the neural network’s heatmaps or classification outputs.

Afterward, we have compared the images annotated by the specialist to the ISNet LRP heatmaps, as GradCAM explanations had worse resolution and class selectivity (lines 476-485). Per-image findings, and all the 30 heatmaps superposed over the annotated X-rays are available in the supplementary material (Supplementary Note 1, and Supplementary Data 1, available at <https://drive.google.com/drive/folders/1Zi212Nnmhp-ukdG0dNSCcaITQi9K-BfD?usp=sharing>). Figure 7 presents some examples, explained in lines 1738-1749. Appendix A displays a detailed qualitative and quantitative comparison between the annotated X-rays and the ISNet heatmaps. In summary, we have observed that the ISNet attention is clearly correlated with the radiologist’s annotations. As expected, the correlation is stronger for COVID-19, as the class has the highest F1-Score (as shown in Figure 7).

As a final note, we are happy that the reviewer’s comment also prompted us to improve the ISNet heatmap loss function (Section 4.3). The previous loss indirectly enforced an overall reduction of the heatmap values, which produced a concentrated and sparse attention profile inside the images’ region of interest. As previously explained to Reviewer #2, such a profile did not hinder accuracy, but it made heatmap interpretation more challenging. The new loss function ensures that the region of interest contains a more natural attention pattern (Figures 1, 2, and 7), which could be better compared to the human specialist’s annotations. We have also changed the normalization strategy used to display the heatmaps in the paper (lines 469-470). Due to the loss function modification, we have re-trained the ISNet and updated its results. They have improved for the COVID-19 detection task and are within the margin of error for the other applications. Following the new normalization strategy, we have updated the heatmaps in Figures 1 and 2. Moreover, now every heatmap displayed in the figures is available in high-resolution in the supplementary material.

Reviewers' Comments:

Reviewer #2:

Remarks to the Author:

I would like to thank the authors for an excellent revision and careful consideration of the reviewers' comments and request. This revision has addressed all my previous concerns, and I believe the manuscript is substantially improved over the revisions.

Reviewer #3:

Remarks to the Author:

The key claim of this work is that the ISNet, a deep neural network (DNN) architecture with an attention mechanism, can effectively focus on the region of interest in an image, minimizing background distractions and improving generalization to external test datasets. This approach is versatile, efficient, and can replace the traditional segmentation-classification pipeline, outperforming other methods in tasks such as COVID-19 and tuberculosis detection in chest X-rays, as well as facial attribute estimation.

The primary distinction between ISNet and the segmentation-classification approach lies in their combined energy minimization function. A potential drawback of this joint minimization is that if one component malfunctions, the entire system may be compromised. The question arises: how can the system remain stable if the segmentation part experiences errors?

Line 745-750

This is because they are complex and nonlinear structures with millions of
746 trainable parameters. Layer-wise relevance propagation¹⁰ (LRP) is an explanation technique
tailored for deep models. It was
747 created to solve the challenging task of providing heatmaps to interpret DNNs. Furthermore, it
is one of the most robust
748 explanation methodologies to date¹⁴. Heatmaps are graphics that show how a neural network
distributes its attention in the input
749 space. For each class, a heatmap explains if an image region has a positive or negative,
strong or weak influence (relevance) on
750 the classifier confidence for that class.

\ What if LRP isn't the most effective method? Would the proposed approach still be valid?

Heatmaps are not always accurate, and there are numerous alternatives and variations. Building a theory on such a weak assumption may be questionable.

Lines 756-757

Furthermore, high (absolute) values of relevance show input features that were important for the classifier decision,
757 i.e., the heatmap makes the classifier's attention explicit.

\ It is difficult to assert that heatmaps are accurate indicators of attention. It might be preferable to compare Visual Transform (a direct attention mechanism) with heatmaps. One may wonder why there isn't a direct comparison if this paper claims the attention mechanism is crucial and improves overall system performance.

As mentioned in lines 330-332, one of the key innovations of the ISNet approach is to prevent background bias attention or distraction, thereby avoiding shortcut learning.

This paper fails to prove its proposed innovations systematically.

The evidence for the proposed ideas is not clearly presented, and the combined energy minimization function (classification and segmentation) has only been applied to several datasets.

To thoroughly explain the proposal, it is crucial to provide concrete examples of shortcut learning, which typical algorithms struggle to address. The various examples merely compare results from applying the method to existing datasets.

It is a priority to demonstrate the following claims made by the paper, even through simulation data if necessary.

"The key claim of this work is that the ISNet, a deep neural network (DNN) architecture with an attention mechanism, can effectively focus on the region of interest in an image, minimizing background distractions and improving generalization to external test datasets."

Reviewer #4:

Remarks to the Author:

The proposed segmentation-guided classification training is simple, but the paper was nicely written. The following are some detailed comments.

(1) It would be better to provide AUC values in Tables 1, 3, 4.

(2) Object regions for both COVID-19 and tuberculosis detection and facial attribute estimation are subjective annotations of experts, thus not easy to evaluate quantitatively. It is suggested to evaluate ISNet in some object classification datasets, such as Stanford Dogs (<http://vision.stanford.edu/aditya86/ImageNetDogs/>), with examples provided in doi: 10.1109/TIP.2019.2908795.

(3) It is good to show some failed generalization in MIMIC. But, for successful cases, the wide differences shown in Tables 1, 3 and 4 make reviewer too confusing to believe that ISNet is trustworthy. Replacing batch normalization in Networks to instance normalization or layer normalization may lead to a better generalization in Tables 1 and 3.

(4) Figure 4 looks like a duplicate of Figure 5. It may be more helpful for readers to understand if LRP block layers could be expanded.

(5) Some typos:

Page 11: a NVidia-> an NVIDIA

Page 11: ISNet training time -> ISNet's training time

We are grateful for the reviewers' time and valuable comments, which improved the quality of our manuscript. Our novel algorithms are publicly available at <https://github.com/PedroRASB/ISNet>. We updated the repository with the source code for the new benchmark DNNs. Moreover, a more complete code (with demos and trained network parameters) is available to the reviewers at the link below. We shall include this detailed code in the released codebase upon publication.

<https://drive.google.com/drive/folders/1Zi212Nnmhp-ukdG0dNSCcalTQi9K-BfD?usp=sharing>.

As before, using the link above the reviewers can also find Supplementary Data 1 (high-resolution images, radiologist's annotations, and ISNet heatmaps).

Reviewer #2:

"I would like to thank the authors for an excellent revision and careful consideration of the reviewers' comments and requests. This revision has addressed all my previous concerns, and I believe the manuscript has substantially improved over the revisions."

We are grateful for the reviewer's time and attention during all the revisions. Her/his comments were very important for improving the manuscript.

Reviewer #3:

We would like to thank the reviewer very much for her/his comments, which we have addressed in the revised manuscript. Following her/his suggestions, we have reformulated the Results and Discussion Sections, improving clarity and objectivity; we have included new experiments and benchmark models, more systematically proving our neural network's advantages over the state-of-the-art; and we have added an entire theoretical Section, which mathematically justifies the success of our model. In summary, the reviewer's suggestions have added value to the manuscript, which now presents clearer and more comprehensive empirical results, which are theoretically justified. Below, we report the point-by-point reply to her/his comments. Please note that, in the revised manuscript, changes are in red.

1- "The key claim of this work is that the ISNet, a deep neural network (DNN) architecture with an attention mechanism, can effectively focus on the region of interest in an image, minimizing background distractions and improving generalization to external test datasets. This approach is versatile, efficient, and can replace the traditional segmentation-classification pipeline, outperforming other methods in tasks such as COVID-19 and tuberculosis detection in chest X-rays, as well as facial attribute estimation.

The primary distinction between ISNet and the segmentation-classification approach lies in their combined energy minimization function. A potential drawback of this joint minimization is that if one component malfunctions, the entire system may be compromised. The question arises: how can the system remain stable if the segmentation part experiences errors?"

We thank the reviewer for the insightful comment, and we provide a detailed response below. In summary, we show that not only the ISNet, but also the state-of-the-art segmentation-classification approach she/he mentioned relies on the proper operation of its segmentation component. Thus, the cited potential drawback is not specific to our proposed model. Moreover, we demonstrate that, even though the ISNet and the alternative segmentation-classification rely on the proper function of their segmentation part, they are much more reliable and generalizable than standard classifiers in the presence of background bias.

First, we must clarify that the mentioned segmentation-classification approach (which we used as a baseline) does not remain stable if its segmentation component fails. The commonly employed methodology comprises a semantic segmenter, which identifies the image's background, followed by a background removal step, and,

finally, the classification of the segmented image. If the segmenter malfunctions and removes the image's foreground, the classifier scores will be unreliable. Moreover, if the segmenter does not remove the background properly, the classification accuracy will significantly drop. Indeed, we have demonstrated that the segmentation-classification pipeline's accuracy falls dramatically when it is tested without its segmenter and background removal step (i.e., simulating a malfunction where the segmenter selects the entire image as foreground) (**lines 568-573 and 629-631**). We hypothesize that the accuracy drop is caused by the fact that a segmenter malfunction represents a strong distributional shift in the classifier's input space, hindering generalization. In other words, badly segmented test images are vastly different from the samples that the classifier saw during training. In summary, our experiments proved that a segmenter malfunction could compromise the entire system in the state-of-the-art segmentation-classification approach.

If we assume a failure in the ISNet segmentation component, we cannot ensure that the system will remain reliable. However, our results demonstrate that this is not a drawback specific to our proposed joint minimization; it is also characteristic of state-of-the-art systems using segmentation and classification. However, as shown in our experiments, systems considering segmentation and classification are necessary to hinder shortcut learning when datasets present background bias (**Section 2.1 and Table 1**).

Although the dependence on segmentation may seem like a substantial drawback, a failure in the segmentation system would be easily detectable for the ISNet, problematic segmentation would result in a poor heatmap loss, inadequate LRP (Layer wise Relevance Propagation) heatmaps, and weak o.o.d. test performance. Moreover, with proper minimization of the ISNet loss, we expect the ISNet to generalize well, and its segmentation ability to be reliable at run-time. In our multiple experiments, we have not observed an ISNet having adequate training losses and heatmaps, but failing at segmenting test images, even when considering o.o.d. data. This phenomenon's explanation is that, especially in the presence of background bias, segmentation tends to generalize better than classification. First, segmenters are resistant to background bias, because, as biasing features are correlated with classification labels, attention to bias only improves classification loss. Second, recent studies have shown that segmenters can have a surprising generalization capacity, performing well even when tested with classes never seen during training [1]. Meanwhile, in the presence of background bias, classifiers are known to focus on shortcuts and have limited generalizability [2][3].

In summary, in a scenario with background bias, a properly trained system using segmentation and classification (e.g., the ISNet) will be more reliable than a standard classifier, even though it depends on the correct operation of two parts. The segmentation part's strong generalization capacity will help the classifier to evaluate the correct image features, and to generalize better. This superior reliability was demonstrated by our experiments, where the ISNet and the state-of-the-art segmentation-classification pipeline significantly surpassed all other neural networks in scenarios with concrete synthetic background bias and the tendency for shortcut learning (please refer to **Section 2.1**, and to our response to the reviewer's last comment). Previous works also pointed out the importance of considering segmentation when classifying datasets with background bias [4].

Following these comments, the revised manuscript now clarifies the ISNet dependence on segmentation and emphasizes the requirements to ensure reliability and generalizability (**lines 1394-1407**). In summary, to ensure reliability in segmentation-classification systems, verifying that both system parts are properly trained is crucial. For the ISNet, this means checking the convergence of all loss functions, verifying the heatmaps, and performing an o.o.d. evaluation. If the trained model fails at such tests, the researcher should retrain the DNN from a different initialization point, consider changing hyper-parameters, and evaluate if the DNN and dataset sizes are compatible with the classification and segmentation tasks at hand.

[1] Birodkar, V., Lu, Z., Li, S., Rathod, V. & Huang, J. The surprising impact of mask-head architecture on novel class segmentation (2021). 2104.00613.

[2] Geirhos, R. et al. Shortcut learning in deep neural networks. *Nature Machine Intelligence* 2, 665–673, DOI: 10.1038/2073s42256-020-00257-z (2020)

[3] DeGrave, A. J., Janizek, J. D. & Lee, S. I. Ai for radiographic covid-19 detection selects shortcuts over signal. *Nature Machine Intelligence* 3, 610–619, DOI: 10.1038/s42256-021-00338-7 (2021).

[4] López-Cabrera, J., Portal Diaz, J., Orozco, R., Lovelle, O. & Perez-Diaz, M. Current limitations to identify covid-19 using artificial intelligence with chest x-ray imaging (part ii). the shortcut learning problem. *Heal. Technol.* 11, DOI:10.1007/s12553-021-00609-8 (2021).

2- “Line 745-750

This is because they are complex and nonlinear structures with millions of

746 trainable parameters. Layer-wise relevance propagation¹⁰ (LRP) is an explanation technique tailored for deep models. It was

747 created to solve the challenging task of providing heatmaps to interpret DNNs. Furthermore, it is one of the most robust

748 explanation methodologies to date¹⁴. Heatmaps are graphics that show how a neural network distributes its attention in the input

749 space. For each class, a heatmap explains if an image region has a positive or negative, strong or weak influence (relevance) on

750 the classifier confidence for that class.

What if LRP isn't the most effective method? Would the proposed approach still be valid? Heatmaps are not always accurate, and there are numerous alternatives and variations. Building a theory on such a weak assumption may be questionable.”

We thank the reviewer for raising this point, which has helped improve the theoretical aspect of our paper. If LRP is not the most effective explanation method, our proposed approach would still be valid because, in essence, LRP has mathematical fundamentals that allow the ISNet optimization process to achieve robust resistance to background attention. The same fundamentals theoretically justify the ISNet’s advantages over the state-of-the-art. The analysis behind these statements constitutes an entirely new **Section, 4.9**, and the **Appendix C**. In summary, the new section explains that the LRP’s roots on Taylor expansions allow the efficient convergence of the ISNet’s optimization process and lead to superior resistance to background bias while maintaining high overall accuracy. The section also theoretically justifies why more popular explanation techniques (e.g., GradCAM, input gradients, and Gradient*Input) lack fundamental qualities found in LRP, leading to the optimization of models with inferior generalization performance in the presence of background bias. To further support these theoretical comparisons, we have included a new benchmark network and an ablation model in our manuscript (Right for the Right Reasons, or RRR, and the ISNet Grad*Input). Experiments with them, like the past tests with GAIN (Guided Attention Inference Networks) and HAM (Hierarchical Attention Mining), empirically demonstrate the advantages of the ISNet over models optimizing popular alternative choices of heatmaps (**Section 2.1**). In the following, we briefly summarize our findings and analysis.

A first-order Taylor expansion decomposes a function’s output, producing one term for each of its input variables. Consider we choose a function’s root near its input (i.e., the data point) as the expansion’s reference point. From a classification standpoint, a root indicates a point of maximal uncertainty, as its zero output locates the ReLU hinge, or the point of 50% probability for the sigmoid activation. Therefore, a Taylor expansion performed at a nearby root will decompose the function into terms that represent the differential contribution (in relation to a state of maximal uncertainty) of the input elements to the function output [1]. Consequently, the minimization of the expansion terms associated with bias reduces the contribution of bias to making the function output different from zero. However, finding an adequate root and decomposing the complex functions implemented by deep neural networks is a complicated and analytically intractable problem [1].

Fortunately, LRP- ϵ is a fast procedure that decomposes the neural network's outputs by approximating a sequence of local Taylor expansions (please refer to **Section 4.9.1** for details). For this reason, LRP- ϵ optimization is an efficient and justifiable alternative to minimize the contribution of biased input elements (background bias) to the network's output class confidence. Moreover, the technique also meets the basic requirements for explanation optimization: it is differentiable and fast. Speed is important because the ISNet must generate multiple heatmaps during training, while differentiability is necessary for neural network optimization. Please refer to **Section 4.9** for a more complete theoretical analysis of the ISNet's LRP optimization and its roots in Taylor expansions.

As the reviewer mentioned, there are indeed multiple techniques to produce heatmaps. Some of them also satisfy the aforementioned basic requirements for heatmap optimization. Popular and viable alternative explanation methods are GradCAM, input gradients, and Gradient*Input. For a more comprehensive evaluation of the advantages of LRP optimization, we have compared the ISNet to benchmark models optimizing these alternative heatmaps. GAIN is a network optimizing GradCAM, which we implemented in the last revision round. Now, we have also included RRR, a model considering input gradients, and the ISNet Grad*Input is an ablation experiment, where we have substituted the ISNet's LRP heatmaps with Gradient*Input explanations. The alternative models are introduced in **lines 127-169**, described in **Section 4.8 (lines 1535-1579)**, and training parameters are provided in **Section 4.12**.

We have empirically demonstrated that the ISNet produces higher resistance to background attention than all the alternative methodologies, according to quantitative results obtained with simulation data, as the reviewer requested in her/his last comment. Such results are in **Table 1**, and they are analyzed in **lines 375-437**. They quantitatively prove that background bias could not influence the ISNet's decisions, but they did influence the alternative networks, causing shortcut learning. These findings are also supported by results in real-world datasets naturally presenting background bias (COVID-19 and tuberculosis detection), where the ISNet better hindered shortcut learning, leading to superior generalization. Scores for COVID-19 and tuberculosis detection appear in **Tables 2 and 4**, respectively, and they are discussed in **lines 555-567 and 601-616**. The LRP heatmaps (**Figures 1 and 2**) show background attention for the alternative methodologies and not for the ISNet, agreeing with the quantitative results in Tables 1, 2 and 4. Moreover, we have justified why LRP optimization is advantageous over the optimization of the alternative techniques. Theoretical comparisons between LRP and GradCAM optimization are in **Section 2.6.1**. Meanwhile, **Section 4.9** constitutes an in-depth theoretical analysis of LRP optimization, with comparisons to the optimization of GradCAM, input gradients and Gradient*Input. Further analysis is presented in **Appendix C**. Conclusions regarding the alternative models are presented in **lines 950-979**.

We are grateful to the reviewer because her/his comment allowed us to provide a much more profound theoretical analysis of the ISNet, which justifies our empirical findings. Therefore, her/his comments have added great value to the study.

[1] Bach, S. et al. On pixel-wise explanations for non-linear classifier decisions by layer-wise relevance propagation. PLOS ONE 10, 1–46, DOI: 10.1371/journal.pone.0130140 (2015).

3- *“Lines 756-757*

Furthermore, high (absolute) values of relevance show input features that were important for the classifier decision,

757 i.e., the heatmap makes the classifier's attention explicit.

It is difficult to assert that heatmaps are accurate indicators of attention. It might be preferable to compare Visual Transform (a direct attention mechanism) with heatmaps. One may wonder why there isn't a direct comparison if this paper claims the attention mechanism is crucial and improves overall system performance."

We thank the reviewer for the suggestion, and we have implemented the requested network. Vision Transformers are an increasingly popular choice of neural network, and they are built upon an attention mechanism. However, the transformer's self-attention does not learn from segmentation targets. Without this external guide, it cannot effectively differentiate background bias features (e.g., text in an X-ray's background) from important class features (e.g., lung disease signs). Both the biasing features and the foreground features are correlated with the classification labels. Accordingly, paying attention to both can improve the transformer's training loss, making the model unreliable in avoiding background bias attention.

This analysis was confirmed by our experiments. In the datasets that naturally had background bias (COVID-19 detection and tuberculosis detection), the ISNet significantly outperformed the Vision Transformer (**Tables 2 and 4**). In COVID-19 detection, the ISNet has 0.773 +/-0.009 average F1-Score, and the transformer 0.46 +/-0.009. In tuberculosis detection, the ISNet achieved a score of 0.738 +/-0.044, and the vision transformer 0.526 +/-0.05. Corroborating the quantitative results, heatmaps show that the vision transformer paid significant attention to the background in both tasks, while the ISNet focused on the lungs (**Figure 2**).

As requested by the reviewer in her/his comment 4, we have used simulation data to create experiments that clearly and quantitatively show shortcut learning, producing an ideal scenario to compare the different neural networks' capacities to avoid background attention. We will explain the experiments in detail in our response to the next comment. Their results, summarized in **Section 2.1**, prove that background features significantly influenced the vision transformer outputs but did not affect the ISNet.

The overall conclusion regarding the transformer's heatmaps and its quantitative performance metrics is that, in relation to a standard convolutional classifier, the vision transformer presented a more diffuse attention profile in all tasks. I.e., it paid significant attention to both the image foreground (i.e., dogs, faces, or lungs) and background. Meanwhile, in the presence of strong background bias, the common convolutional network concentrated more attention on the background (**Figure 1**). This finding explains why, in highly biased environments (**Section 2.1 and Table 1**), the vision transformer surpassed the standard convolutional network. However, because it could not ignore the background, the transformer was significantly surpassed by the ISNet in all experiments.

Please find details about our transformer implementation in **Section 4.8 (lines 1526-1534)**. **Section 4.12** describes training procedures. The model's behavior is analyzed in **lines 480-486 and 496-506**. Its quantitative results are presented in **Tables 1 to 5**. Finally, its heatmaps appear in **Figures 1 and 2**, and are analyzed in **lines 694-698 and 762-766**. Overall conclusions, which consider the vision transformer, are presented in **lines 897-907, 911-914, 928-933 and 942-946**.

4- "As mentioned in lines 330-332, one of the key innovations of the ISNet approach is to prevent background bias attention or distraction, thereby avoiding shortcut learning.

This paper fails to prove its proposed innovations systematically.

The evidence for the proposed ideas is not clearly presented, and the combined energy minimization function(classification and segmentation) has only been applied to several datasets.

To thoroughly explain the proposal, it is crucial to provide concrete examples of shortcut learning, which typical algorithms struggle to address. The various examples merely compare results from applying the method to existing datasets.

It is a priority to demonstrate the following claims made by the paper, even through simulation data if necessary.

“The key claim of this work is that the ISNet, a deep neural network (DNN) architecture with an attention mechanism, can effectively focus on the region of interest in an image, minimizing background distractions and improving generalization to external test datasets.””

We apologize for the lack of clarity in the previous version of the manuscript. During the past revision rounds, multiple experiments and benchmark models were included in the paper, following the suggestions of reviewers and the editor. To improve the tracking of changes, we avoided modifications to the entire Results Section. Unfortunately, this decision was prejudicial to the clarity and objectivity of the manuscript. We thank the reviewer for her/his comment, which prompted a major reformulation of the **Results and Discussion Sections**. Moreover, as requested, we have utilized simulation data to create concrete examples of shortcut learning, where we could quantitatively demonstrate the ISNet advantages over the state-of-the-art.

We begin by explaining the new experiments. The past version of the manuscript had some experiments utilizing synthetic background bias, but they were reported only in text, in sparse locations throughout the **Results Section**. We have expanded such experiments, creating scenarios where the effect of shortcut learning is clearly visible and quantifiable. Furthermore, the experiments now contain more benchmark models and a new application. All experiments using simulation data are aggregated in **Section 2.1**, and all quantitative results are summarized in **Table 1**.

Shortcut learning induced by background bias designates a scenario where irrelevant background features influence the outputs of neural networks [1]. In other words, classifiers learn to pay attention to the background and consider it in its predictions. The biasing features are correlated with the classification problem labels, but only in the training dataset and i.i.d. (independent and identically distributed) data [1]. For this reason, classifiers that pay attention to them and suffer shortcut learning will not generalize well, showing a performance drop in real-world test data (out-of-distribution), as it does not contain the same background bias present in training [1].

Exemplifying a scenario of background bias, the simulation data experiments consist of digitally including biasing features in the background of real images. Such features are geometrical shapes, positioned at the image’s corners (examples are provided in **Figure 1**, first row). As background bias is correlated with the classification labels, we choose the geometrical shapes according to the image’s class. For example, in COVID-19 detection, we have added a triangle to all X-rays displaying COVID-19, a circle to all pneumonia images, and a square to all samples with healthy patients. An advantage of working with synthetic background bias is that it allows the creation of controlled experiments. We have trained the neural networks with the synthetically biased images, and we have created three evaluation scenarios to assess the models’ resistance to shortcut learning: the biased test, the unbiased test, and the deceiving bias test (which was not present in previous versions of the manuscript). The biased test is a test dataset containing the same background bias as the training data. The standard test has no geometrical shape, typifying external, out-of-distribution (o.o.d.) data. Finally, the deceiving bias evaluation is an extreme case of o.o.d. data, it contains the synthetic geometrical shapes, but their correlation with the classification categories is changed. For example, in a test X-ray showing a COVID-19 patient, a square will now be present (instead of a triangle). Accordingly, the deceiving bias will try to fool the neural network, making it predict the wrong class. The synthetic bias experiments are described in **lines 249-272** and reported in **Section 2.1**. Further details about the inclusion of geometrical shapes are in **Section 4.10 (lines 1844-1858, 1942-1950, and 1976-1983)**.

If a neural network can effectively focus on the region of interest and prevent the shortcut learning caused by background bias, it will perform equally on the three evaluation scenarios, proving that the synthetic bias could not influence its outputs. Meanwhile, a model that cannot hinder background attention will display the behavior that characterizes shortcut learning: it will show the highest accuracy in the biased test, followed by an accuracy

drop in the standard test, and a further performance reduction in the deceiving bias evaluation. The three testing scenarios are represented in the columns of **Table 1**. Reductions in F1-Score, when columns are read from left to right, are proof of shortcut learning and the influence of background bias over a classifier’s decisions. Thus, differences between the three evaluation performances represent quantitative evidence of shortcut learning.

We have conducted experiments adding synthetic background bias to 3 different applications. First, COVID-19 classification and facial attribute estimation, applications that were already present in the manuscript, which are also analyzed without synthetic bias. Moreover, we have included a dog breed classification task, utilizing a subset of the Stanford Dogs dataset (described in **Section 4.10.4**). This specific dataset was requested by Reviewer #4, as she/he suggested that the dogs constituted a great choice of image foreground to display the ISNet capacity of focusing on a region of interest. By adding strong synthetic bias to three diverse applications, we can comprehensively demonstrate the ISNet’s capacity to avoid background attention. Accordingly, the 3 tasks consider vastly different dataset sizes (ranging from a few hundreds to tens of thousands of images), image domains (medical images, pictures of people and dogs’ photos), foreground definitions (lungs, dogs, and faces), data augmentation procedures (**Section 4.11**), and classifier backbones (**Section 4.8, and lines 346-353**). By analyzing consistent result patterns in 3 remarkably diverse scenarios, we more systematically prove our conclusions and the ISNet advantages over the state-of-the-art (**Section 2.1**).

We analyze the results of a standard classifier (convolutional neural network without any attention mechanism) to demonstrate that the 3 synthetic bias tasks are concrete examples of shortcut learning, as requested by the reviewer (**Table 1**). In Stanford Dogs, when the artificial geometrical shapes are present in the test dataset, the network achieves 0.926 +/-0.019 average F1-Score, which drops to 0.419 +/-0.034 when they are removed, and to 0.071 +/-0.017 when they are replaced by deceiving bias. The influence of background bias over the classifier outputs is extreme. To put the results into perspective from the 201 evaluation samples, in the biased test the model correctly classifies 192, in the unbiased test, 89, and, in the deceiving test, 10 (**lines 319-320**). The other two applications with synthetic bias also constitute concrete examples of shortcut learning. In COVID-19 detection, the standard classifier average F1-Score is 0.775 +/-0.008 in the biased test, dropping to 0.434 +/-0.01 in the standard evaluation, and 0.195 +/-0.004 with deceiving bias. In facial attribute estimation, the common classifier scores in the three evaluation scenarios are 0.974 +/-0.012, 0.641 +/-0.054 and 0.398 +/-0.019, respectively.

Results are summarized in **Table 1** for all networks. Only the ISNet and the segmentation-classification pipeline display no accuracy drop when the synthetic bias was removed or substituted by deceiving bias. As requested by the reviewer, the experiments constitute scenarios where the ISNet excels, while the alternative models struggle. These results confirm the findings from the previous versions of the manuscript, where we concluded that the ISNet and the alternative pipeline were the networks with the highest resistance to background bias, among the multiple models we tested. The ISNet was initially conceived to substitute the much larger and slower segmentation-classification pipeline, and it could even surpass the alternative model in 4 of the 5 experiments that contained background bias (3 simulation data experiments in **Table 1**, 2 non-synthetic data experiments in **Tables 2 and 4**). The results of the synthetic bias experiments are discussed in detail in **Section 2.1 (and Subsection 2.1.1)**. Furthermore, **Figure 1** shows that LRP heatmaps are coherent with the quantitative evidence of background bias in **Table 1**: only the ISNet’s and the alternative pipeline’s heatmaps show no attention to the synthetic bias.

The quantitative results in the synthetic bias experiments corroborate the findings from the two experiments with non-synthetic background bias, COVID-19 and Tuberculosis detection (**Sections 2.2 and 2.3**). The ISNet could reliably avoid the influence of the artificial geometrical shapes, thus minimizing shortcut learning, which led to its superior accuracy on o.o.d. data (i.e., the standard and deceiving bias experiments in **Table 1**). Accordingly, the proposed model could also minimize shortcut learning caused by non-synthetic bias, leading to

the highest o.o.d. performance in **Tables 2 and 4** (o.o.d. performance improvement is a characteristic sign of shortcut learning reduction [1]). Once more, LRP heatmaps are in accordance with the numerical results, showing no background attention for the ISNet.

Finally, we would like to explain the improvements to clarity and objectivity in the manuscript's **Results Section**. It now begins by concisely explaining the objectives of each experiment (**lines 274-289**). The three synthetic bias applications are presented first, to clearly and quantitatively demonstrate the ISNet's capacity to hinder background bias, and to better compare the multiple neural networks we tested (**Section 2.1**). Afterwards, COVID-19 and tuberculosis detection in X-rays are presented to exemplify contemporary computer vision applications known for having background bias (**Sections 2.2 and 2.3**). With them, we demonstrate that the ISNet is also advantageous in real-world tasks. Finally, facial attribute estimation is presented as an in-domain application, without apparent background bias (**Section 2.4**). It serves to better delimit the ISNet objective and use-case, besides displaying the network's capacity of focusing on the foreground of natural images with diverse regions of interest (faces) shapes, positions, and locations.

In summary, following the reviewer's request, our manuscript now presents concrete and quantifiable examples of shortcut learning, where the proposed ISNet significantly surpasses the state-of-the-art (**Section 2.1**). Moreover, it demonstrates that the model's positive results lead to significant advantages in real-world applications known for shortcut learning (**Sections 2.2 and 2.3**). Finally, the manuscript now contains much deeper theoretical justifications for the ISNet's qualities and study's empirical results (**Section 4.9 and Appendix C**). Thus, we thank the reviewer again for her/his important contribution and her/his time.

[1] Geirhos, R. et al. Shortcut learning in deep neural networks. *Nature Machine Intelligence* 2, 665–673, DOI: 10.1038/2073s42256-020-00257-z (2020)

Reviewer #4:

We thank the reviewer very much for her/his comments. We carefully addressed them, and we provide a point-by-point response below. Please note that, in the revised manuscript, modifications are in red.

1- *"The proposed segmentation-guided classification training is simple, but the paper was nicely written. The following are some detailed comments.*

It would be better to provided AUC values in Tables 1, 3, 4."

We thank the reviewer for the comment, which has helped improve the paper presentation. We are also grateful for the kind words about how the paper was written. As suggested, we have included the AUC scores in Tables 1, 3, and 4, now **Tables 2, 4, and 5**, respectively. Additionally, the tables now present the per-class AUC scores, while the last version of the paper only included the macro-average measurements. All per-class scores have interval estimates. As in the previous version of the manuscript, the macro-average AUCs in the multi-class single-label classification task (i.e., COVID-19 classification, **Table 2**) are presented as point estimates. Once again, the reason for this decision is given in **Section 4.13**, which explains the statistical methods employed in the results' analysis.

2- *"Object regions for both COVID-19 and tuberculosis detection and facial attribute estimation are subjective annotations of experts, thus not easy to evaluate quantitatively. It is suggested to evaluate ISNet in some object classification datasets, such as Stanford Dogs (<http://vision.stanford.edu/aditya86/ImageNetDogs/>), with examples provided in doi: 10.1109/TIP.2019.2908795."*

We thank the reviewer very much for the suggestion. We have included experiments using the proposed dataset, and the ISNet could clearly concentrate its attention on the dogs, as shown in **Figure 1**. Indeed, the experimental results have further confirmed the findings from the previous applications, showing that the ISNet decisions were not based on background features (**Section 2.1**). The dataset is described in **Section 4.10.4**. We would like to explain a few points about the Stanford Dogs experiment in more detail.

First, the Stanford Dogs dataset has no segmentation mask highlighting the dogs. Such masks are required for training the ISNet and many of our benchmark DNNs. However, the database has bounding boxes, which we have fed to a pre-trained segmenter (DeepMAC [1]), to create the necessary ground-truth segmentation masks. DeepMAC is a neural network specialized in the segmentation of object classes not seen during training. Upon visual inspection, we have confirmed that the generated masks were accurate. Thus, as an additional benefit, the experiment proposed by the reviewer showed how segmentation masks for ISNet training can be easily created, with general pre-trained segmenters (**lines 270-272** and **Section 4.10.4**).

Second, like the facial attribute estimation dataset, Stanford Dogs is not a database characterized by background bias. However, the objective of our experiments is to show that the ISNet can avoid classifier attention to background bias. Biasing features pull the classifier's attention away from the region of interest (e.g., the dogs), and produce models that perform very well when the bias is present but that lose much accuracy when it is removed. I.e., background bias causes shortcut learning and hinders generalization [1]. To clearly delimit the ISNet use-case scenario (avoiding biased attention), we already have two experiments showing that, in the absence of background bias, the ISNet and all the benchmark models designed to prevent biased attention (e.g., the segmentation-classification pipeline) are roughly equivalent to a standard classifier. These experiments are facial attribute estimation, reported in **Section 2.4**, and CheXPert/MIMIC classification, analyzed in **Appendix B**.

To demonstrate that the ISNet can focus on the dogs even in the presence of strong background bias, we have added synthetic bias (geometrical shapes correlated with the dogs' breeds, described in **lines 249-272** and **Section 4.10.4, lines 1976-1983**) to the background of the Stanford Dogs' images. With the shapes, the new experiment becomes ideal for comparing the multiple tested neural networks' resistance to background bias. Once more, quantitative results confirm our previous findings, and show the ISNet's superior resistance to bias, as the proposed model maintained its accuracy when the geometrical shapes were removed from the test dataset (**Table 1** and **Section 2.1**). Moreover, with the addition of the synthetic bias, the Stanford Dogs experiment also complied with Reviewer #3's request to include another application (considering synthetic data) demonstrating the ISNet's resistance to strong background bias, in a scenario where other neural networks fail.

The geometrical shapes have a strong tendency to attract classifier attention. However, the ISNet resists and focuses on the dogs (**Section 2.1, Table 1** and **Figure 1**). Besides presenting heatmaps for all models trained with the synthetically biased data, **Figure 1** also shows the heatmaps for a standard classifier trained with Stanford Dogs images without the geometrical shapes (last row). Thus, the reviewer can observe that the ISNet (trained with synthetically biased data) attention is much more concentrated on the dogs, even in relation to a model trained with the original dataset.

Finally, we would like to clarify that we utilized a subset of the Stanford Dogs database, composed of all images displaying the following breeds: Pekingese, Tibetan Mastiff, and Pug (as explained in **lines 266-272**, and **Section 4.10.4**). We employed a random number generator to select these breeds. As expressed in the manuscript (in **Section 2.5** and in the **Discussion Section, lines 980-986**), although the ISNet is fast at run-time, it has the disadvantage of increasing its training time with the number of categories in the classification problem. Therefore, considering all 120 classes from Stanford Dogs would require an extremely long training time. Moreover, it would make the reproduction of our results much more expensive and time-consuming, especially for researchers without access to powerful computational resources. With such consideration in mind, Reviewer

#2 requested experiments using only a subset of the large CheXPert database (**Appendix B**). Nonetheless, the new experiment using 3 dog breeds served our purposes well: its results have confirmed our past findings, and we could successfully show that the ISNet concentrates its attention on the dogs, as requested by the reviewer.

[1] Geirhos, R. et al. Shortcut learning in deep neural networks. *Nature Machine Intelligence* 2, 665–673, DOI: 10.1038/2073s42256-020-00257-z (2020)

3- *“(3) It is good to show some failed generalization in MIMIC. But, for successful cases, the wide differences shown in Tables 1,3 and 4 make reviewer too confusing to believe that ISNet is trustworthy. Replacing batch normalization in Networks to instance normalization or layer normalization may lead to a better generalization in Tables 1 and 3.”*

We thank the reviewer very much for her/his suggestion. As proposed, we have tried replacing batch normalization with instance and layer normalization. Unfortunately, the substitution did not lead to improvements in the generalization results. Using the baseline classifier for reference, in COVID-19 detection (previous Table 1, now **Table 2**) the model achieved 0.416 +/-0.09 average F1-Score with layer normalization and 0.437 +/-0.01 with instance normalization. In tuberculosis detection (previous Table 3, currently **Table 4**), the F1-scores were 0.504 +/-0.05 and 0.561 +/-0.05, considering layer and instance normalization, respectively. The average performance scores did not surpass the values seen with batch normalization: 0.546 +/-0.01 in COVID-19 detection (**Table 2**), and 0.566 +/-0.05 in tuberculosis detection (**Table 4**).

Background bias and shortcut learning are the major causes of the generalization difficulty seen in **Tables 2 and 4** (COVID-19 and Tuberculosis detection). Such difficulty is also pointed out in previous works, which are useful to put our results into perspective (**lines 541-554**) [1][2]. Many methodologies that can improve generalization in other scenarios are not efficient in dealing with shortcut learning and background bias, as demonstrated by the unimpressive generalization results of multiple state-of-the-art neural networks in **Tables 2 and 4**. This scenario may explain why the alternative normalization schemes were also not successful.

To maximize the reliability of our results, the datasets we consider for the tasks of tuberculosis and COVID-19 detection are among the largest open and high-quality data sources for COVID-19 and tuberculosis X-rays (**Sections 4.10.1 and 4.10.2**). Previous works commonly employ smaller test datasets and show larger interval estimates when also utilizing o.o.d. evaluation (**lines 541-554**). Our dataset selection was performed in early revision rounds, satisfying requirements set by the Nature Communications editor. Moreover, our conclusions and analyses consider the interval estimates in **Tables 2 and 4**. For example, taking the intervals into account, we observe that the ISNet performances in **Tables 2 and 4** significantly surpass all the benchmark neural networks.

We acknowledge that the results in **Tables 2 and 4** (previous Tables 1 and 3) are remarkably diverse from those in **Table 5** (previously Table 4). Such a difference is explained by the diverse nature of the experiments expressed in the tables: COVID-19 and tuberculosis detection (**Tables 2 and 4**) are known examples of contemporary tasks that have background bias, causing a strong tendency for shortcut learning and limited generalizability [1][2][3] (please refer to **lines 188-232**). The bias greatly limits the generalization capacity of classifiers, reducing the benchmark DNNs’ performance in Tables 2 and 4. Because it can avoid background attention and the consequent shortcut learning, the ISNet outperformed the multiple benchmark networks in the two tasks. Meanwhile, facial attribute estimation with the Celeb-A dataset is an example of an in-domain application where background bias is not apparent, and there is no strong tendency for shortcut learning (**Section 2.4**). Thus, all benchmark models performed well, and all the tested neural networks led to remarkably similar high classification accuracy (**Table 5**). The first two tasks illustrate ISNet’s capacity to address scenarios of background bias (where the state-of-the-art struggles). In contrast, the last task better delimits our architecture’s objective and use-case scenario. Moreover, **Table 1** (column “Standard Test maF1”, last 9 rows) shows that, when synthetic background bias is

injected in the facial attribute estimation dataset, the results for the application start to resemble those seen in COVID-19 and tuberculosis detection (**Tables 2 and 4**), as the benchmark neural networks lose much generalization performance, making the ISNet performance substantially superior (as it is not affected by background bias).

Regarding the ISNet trustworthiness, we now present many experiments considering both synthetic (**Section 2.1**) and non-synthetic (**Sections 2.3 and 2.4**) background bias. They constitute highly diverse scenarios, including changes in the image domain (natural and medical images), dataset size, the definition of the region of interest (dogs, lungs or faces), and task difficulty. Accordingly, numerical results vary across the experiments. To demonstrate the trustworthiness of the novel model, we have focused on analyzing patterns consistent across all experiments' results. Indeed, in the 6 scenarios considering synthetic or non-synthetic bias in our study (**Sections 2.1 to 2.3, and lines 2441-2467**), the ISNet significantly surpasses all other benchmark neural networks without a dedicated segmenter. Moreover, the ISNet could even surpass the larger and slower (**lines 668-675**) segmentation-classification pipeline in all but the Stanford Dogs experiment. Finally, all experiments with synthetic bias consistently demonstrated that the ISNet's decisions were not influenced by background features, unlike all other single network models (**Section 2.1**). Please note that we reformulated our **Results and Discussion Sections**, more clearly and objectively explaining the differences between the multiple experiments in our study.

[1] DeGrave, A. J., Janizek, J. D. & Lee, S. I. Ai for radiographic covid-19 detection selects shortcuts over signal. *Nature Machine Intelligence* 3, 610–619, DOI: 10.1038/s42256-021-00338-7 (2021).

[2] López-Cabrera, J., Portal Díaz, J., Orozco, R., Lovelle, O. & Perez-Diaz, M. Current limitations to identify covid-19 using artificial intelligence with chest x -ray imaging (part ii). the shortcut learning problem. *Heal. Technol.* 11, DOI:10.1007/s12553-021-00609-8 (2021).

[3] World Health Organization. Chest radiography in tuberculosis detection: summary of current WHO recommendations and guidance on programmatic approaches (World Health Organization, 2016).

4- *“Figure 4 looks like a duplicate of Figure 5. It may be more helpful for readers to understand if LRP block layers could be expanded.”*

We thank the reviewer for the comment. We must agree that the two images looked similar. Following the reviewer's suggestion, we have expanded the LRP block layers in Figure 5, briefly indicating their sequence of mathematical operations. A more detailed description of the LRP block layers (and these operations) is available in **Section 4.4**. We have only expanded the layers in Figure 5, because Figure 4 appears before **Section 4.4**. Moreover, by expanding only one of the figures, we hope to make them more diverse, as Figure 4 is now much more concise than Figure 5. If the reviewer still thinks the two figures are redundant, we will happily substitute them with a single image. In the revised manuscript, Figure 5 became **Figure 6**, and Figure 4 is now **Figure 5**. Please find **Figure 6's** description in **lines 1347-1351**.

5- *“Some typos:*

Page 11: a NVidia-> an NVIDIA

Page 11: ISNet training time -> ISNet's training time”

We have corrected the typos.

Again, we thank the reviewer very much for her/his time and comments, which have helped improve our manuscript's overall quality.

Reviewers' Comments:

Reviewer #4:

Remarks to the Author:

I would like to express my gratitude to the authors for their exceptional revisions and thorough attention to the reviewers' comments and requests. The code that the author has shared on Github, as well as the revisions, appear to be robust.

1. The author has opted to utilize DenseNet121 instead of other recent and more widely used classifiers, as referenced here (<https://paperswithcode.com/sota/image-classification-on-imagenet>). Could the author explain their rationale behind this choice?
2. The newly incorporated Vision Transformer demonstrated significantly inferior performance compared to the "standard classifier" in Tables 1 and 2. Could the author provide some detailed settings? For instance, is it a ViT-base, ViT-Huge, or another variant? Furthermore, were these networks pretrained (like ImageNet supervised, CLIP, MAE) or were they initialized randomly? I am of the belief that larger Vision Transformers generally have superior generalizability, which may reduce "shortcut learning".
3. How does the hyper-parameter P in L_{is} vary across different datasets, especially considering that some datasets don't have a shortcut, while others have a strong one? The author noted in line 1130 that it "does not seem to strongly affect model performance." Is there any quantitative analysis to support this assertion?
4. How is the $\text{abs}(H_{bk})$ backpropagated given that $\text{abs}()$ is non-differentiable? In your code, it's written as `heatmap=torch.abs(heatmap)`.

Reviewer #4:

"I would like to express my gratitude to the authors for their exceptional revisions and thorough attention to the reviewers' comments and requests. The code that the author has shared on Github, as well as the revisions, appear to be robust."

We thank the reviewer very much for her/his kind words about our work. Moreover, we are also grateful for her/his suggestions, which helped improve the quality of the manuscript. We have carefully addressed all new comments and provide a point-by-point answer below.

Moreover, we want to inform the reviewer that we have performed a text revision, which Nature Communications required in this final review stage. Mainly, we have made the language more concise in the main paper (Introduction, Results, Discussion, and Methods) and moved information and sections from the main text to the Supplementary Information. Furthermore, some sections (4.3 ISNet Loss Function and 2.4 Heatmaps) are now summarized in the main manuscript and described in more detail in the Supplementary Information. These changes were necessary to comply with Nature Communications' word limit and to make the main article more accessible to a general audience. However, we clarify that we have not removed any information or detailed analysis from the paper. For example, the Results Section presents and analyzes the performances of all implemented neural networks, but we have moved the specific and in-detail comparison between the ISNet and the 8 benchmark models to Supplementary Note 2. The facial attribute estimation task was grouped with the CheXPert application in Supplementary Note 7 since both experiments serve to delimit the ISNet use-case better (but facial attribute estimation with synthetic bias is still in the main manuscript, Section 2.1). In summary, everything that was removed from the main text (due to space limitations) can be found in the Supplementary Information. As requested by the journal, the paper's title was changed to remove the punctuation, and images displaying faces were anonymized for privacy reasons.

We marked in red the manuscript modifications we have made to address the reviewer's comments. To avoid confusion, minor language modifications were not marked. The editorial checklist requested the removal of tracked changes in the Supplementary Information file. However, in this file, all changes related to the reviewer's comments are in Supplementary Notes 6.2, 6.3, and 6.4, as will be explained in the point-by-point response below.

1- *"The author has opted to utilize DenseNet121 instead of other recent and more widely used classifiers, as referenced here (<https://paperswithcode.com/sota/image-classification-on-imagenet>). Could the author explain their rationale behind this choice?"*

Layer-wise Relevance Propagation is a general explanation technique that can be applied to multiple neural network architectures. For this reason, we can utilize multiple architectures as the ISNet backbone, making our approach versatile. To display this versatility, we implemented an ISNet based on a VGG-19 backbone in Stanford Dogs classification, and we used a DenseNet121 in the remaining applications. This choice allows us to show that the ISNet works well with a more classical backbone, the VGG containing 19 layers and no skip connections, and with a deeper backbone, considering 121 layers and skip connections.

The VGG was chosen to exemplify less deep neural networks with fewer layers and no skip connections. We selected it precisely because it is a standard, influential, and popular architecture, bearing a simple yet effective design.

Meanwhile, we used DenseNet121 to represent deeper neural networks with skip connections. It was specifically chosen for a few reasons. First, it contains fewer parameters and is more efficient than ResNets with similar depths. This focus on efficiency resonates with the ISNet objective of improving robustness to background bias while keeping low computational cost at run-time. Although more modern architectures are also very deep and efficient, the DenseNet121 is the most popular architecture for lung disease classification in chest X-rays [1]. Because two of our applications fall within this category (Tuberculosis and COVID-19 classification with chest X-rays), the DenseNet121 was a natural choice for the ISNet backbone.

We thank the reviewer for her/his comment. Now, we have better explained the reasons for the DenseNet121 choice in the main manuscript (lines 195 to 203).

[1] Bressemer, K.K., Adams, L.C., Erxleben, C. *et al.* Comparing different deep learning architectures for classification of chest radiographs. *Sci Rep* 10, 13590 (2020). <https://doi.org/10.1038/s41598-020-70479-z>

2- *"The newly incorporated Vision Transformer demonstrated significantly inferior performance compared to the "standard classifier" in Tables 1 and 2. Could the author provide some detailed settings? For instance, is it a ViT-base, ViT-Huge, or another variant? Furthermore, were these networks pretrained (like ImageNet supervised, CLIP, MAE) or were they initialized randomly? I am of the belief that larger Vision Transformers generally have superior generalizability, which may reduce "shortcut learning"."*

We used the ViT-base (ViT-B/16), because, in relation to other ViT variants, its number of parameters (86M) was closer to the other benchmark architectures we used in this work, allowing a fairer comparison. This specification is provided in Supplementary Note 2.1 and copied below.

"The Vision Transformer[2] is another benchmark model, as it represents an increasingly popular choice of attention mechanism, which also does not learn from segmentation masks. The model is based on the transformer, a highly successful structure in natural language processing. In summary, the transformer implements a self-attention mechanism, which relates diverse positions in its input sequence, producing a new representation of the sequence[3]. Initially, the sequence elements are based on linear projections of patches in the input image, which are processed by a series of transformer encoders. Unlike all other tested models, the Vision Transformer is not a convolutional neural network. Specifically, we employed the ViT-base (ViT-B/16) architecture[2], which has 12 layers, breaks the input image into 16x16 patches, and contains 86M parameters. The specific model was chosen because its parameter count scale more closely matches the other neural networks in this study. Moreover, since we do not employ pretraining in this work (a decision justified in Supplementary Note 6.4), utilizing a model with a more conservative size can reduce the risk of overfitting. The network's implementation is the standard PyTorch Vision Transformer model."

For better clarity, we have also specified that we are using ViT-B/16 in all Tables. The ViT network was randomly initialized, like all other models in the study, including the ISNet. The reviewer's comment about pretraining is very relevant. In the new version of the manuscript, we have explained in detail why we have not employed transfer learning in our experiments and why this decision is important for the generality of our study's conclusions. The new analysis, which contains a quantitative example, is presented in Supplementary Note 6.4. For the reviewer's convenience, we copied the explanation below. The decision of not utilizing pretraining, justified below, also enforces our choice of using ViT-B/16 instead of larger vision transformers. Without transfer learning, the larger models could more easily overfit our databases.

"In this study, all DNNs were initialized with PyTorch's standard random weight initialization procedures. In this section, we discuss the possible effects of pretraining on shortcut learning and attention to background bias. Accordingly, we justify why we chose not to utilize transfer learning in this study. In summary, transfer learning consists in training a neural network on one dataset (pretraining), and then training it again (fine-tuning) on another database, which is usually smaller than the first one. Its main objective is to use generalizable knowledge learned in the first dataset to improve performance and generalization in the task represented by the second database.

Pretrained neural networks have already learned to analyze image features that appeared in their pretraining datasets. Therefore, we expect a pretrained model to begin the fine-tuning procedure with a predilection to focus on features that are similar to what it learned in the pretraining dataset. If the pretraining data is more similar to the fine-tuning dataset's background bias than to the database's foreground features, we expect the neural network to start fine-tuning with a predilection to focus on the bias, increasing shortcut learning. Instead, if the pretraining dataset is more related to the fine-tuning database's foreground features, transfer learning could reduce shortcut learning. If the pretraining data features are unrelated to both the foreground and the background in fine-tuning, transfer learning should have little effect over shortcut learning. In case pretraining features are related to both the foreground and the background, pretraining may help the DNN analyze both the foreground features and the background bias, making it difficult to predict what will have the strongest influence over the classifier. In summary, the consequence of pretraining on foreground focus and shortcut learning is highly dependent on the pretraining and fine-tuning datasets' characteristics. Accordingly, if we employ experiments with pretrained models to analyze shortcut learning, the experiments' results and conclusions will lose generality and become more data-dependent. Therefore, pretraining will make it more difficult to draw general conclusions about different architectures' resistance to shortcut learning.

We do not want to draw conclusions that depend on the pretraining data being more similar to the foreground features than to background bias, because we have no guarantee that this assumption will hold in many realistic scenarios. For example, we cannot guarantee that ImageNet (a natural image dataset) pretraining will prompt attention to the lungs (foreground) in X-ray classification. Past studies have shown remarkable attention to background bias (e.g., text and markers) and strong shortcut learning in neural networks pretrained on ImageNet and fine-tuned for COVID-19 classification with chest X-rays[4]. A previous work observed a concerning quantity of LRP relevance outside of the lungs, even though their classifier was pretrained on ImageNet, then trained again on a large single-source

X-ray dataset, and finally fine-tuned for COVID-19 classification in a mixed source database [5]. For example, in the study, LRP revealed that words exclusively present in the COVID-19 class (e.g., Italian words) strongly attracted the classifier's attention. In summary, an ImageNet-pretrained model could show little shortcut learning in an experiment where foreground features are very similar to ImageNet features. This result could lead to the conclusion that the network generalizes well and is resistant to background bias. However, if this strong generalization is not inherent to the architecture but results from pretraining, it may not hold in the COVID-19 detection task. Thus, the conclusion from the first experiment would have limited generality.

Even in the specific cases where pretraining causes foreground focus, we have no guarantee that this predilection will persist throughout the entire fine-tuning procedure. Decision rules based on background bias may represent a much lower classification loss than rules based on foreground features. Thus, without specific strategies to avoid background attention, a pretrained classifier may eventually arrive at the lower loss minima corresponding to biased decision rules. This possibility further limits the generality of shortcut learning analyses in pretrained models: we could conclude that a pretrained network hinders background bias attention, but different fine-tuning hyperparameters (e.g., learning rate) could allow it to find loss minima representing biased decision rules.

In summary, the pretraining influence on shortcut learning depends on the characteristics of the pretraining and fine-tuning datasets. Moreover, we cannot guarantee that any reduction in shortcut learning caused by pretraining will not be lost during the fine-tuning procedure. In this study, we introduced a deep neural architecture and designed experiments to assess its inherent resistance to background bias relative to the state-of-the-art. To maximize the generality of the experiments' conclusions, we initialized all models with standard random parameter initializations, ensuring that the networks begin with no data-dependent predilection to focus on the images' background or foreground. Accordingly, the experiments' results produced consistent patterns from which we could draw more robust and general conclusions about the ISNet. The study of how pretraining influences shortcut learning in multiple deep neural network architectures is an important research topic for future work, which requires careful consideration of the datasets used and the generality of the conclusions drawn.

We finish this section with a quantitative example of how the effect of pretraining on shortcut learning is highly data-dependent. We compare the consequences of pretraining on the Vision Transformer (ViT-B/16) in two tasks, facial attribute estimation and COVID-19 detection, both with synthetic background bias. We consider the standard PyTorch's ImageNet-1K pretrained model. We do not employ dog breed classification because the Stanford Dogs images were originally extracted from ImageNet[6]. Thus, considering the rare scenario where the pretraining and fine-tuning datasets come from the same data distribution would lead to even less general conclusions.

In the synthetically biased facial attribute estimation task, the randomly initialized Vision Transformer obtained macro-average F1-Scores (with 95% confidence intervals, refer to Supplementary Note 10 for details about the statistical analysis) of 0.675 ± 0.023 , 0.645 ± 0.03 , and 0.531 ± 0.023 in the biased, standard and deceiving bias test datasets, respectively (main article Table 1). The same model pretrained on ImageNet (and fine-tuned following the specifications in Supplementary Note 6.2) achieved maF1 of 0.904 ± 0.019 , 0.726 ± 0.04 , and 0.456 ± 0.018 in the three testing scenarios,

respectively. The improvement in the standard test dataset (which has no synthetic bias) result indicates that pretraining allowed the network to better analyze the foreground features (faces). This finding is expected, given that faces are also common in ImageNet. However, the worse result in the deceiving bias evaluation shows that the model's decisions are still strongly influenced by the synthetic background bias. Indeed, the higher F1-Score in the biased test dataset indicates that pretraining even helped the DNN better analyze the background bias.

In synthetically biased COVID-19 detection, the randomly initialized Vision Transformer achieved macro-average F1-Score (with standard deviation) of 0.685 +/-0.009, 0.496 +/-0.009, and 0.327 +/-0.009 in the biased, standard, and deceiving bias test datasets, respectively (main article Table 1). With ImageNet pretraining (and fine-tuning as specified in Supplementary Note 6.2), the scores changed to 0.963 +/-0.004, 0.421 +/-0.009, and 0.031 +/-0.003. As in facial attribute estimation, pretraining helped the Vision Transformer analyze the synthetic background bias, improving the biased test performance and reducing the deceiving bias F1-Score. However, pretraining did not improve the standard test result in COVID-19 detection. Thus, it could not help the network analyze the foreground (lung) features. Conversely, in facial attribute estimation, pretraining improved the analysis of foreground features (faces), increasing the standard test F1-Score. The diverse consequences of pretraining on the two tasks are justified by the fact that ImageNet (a natural image dataset) features are much more similar to the faces in CelebA than to the lung features in the COVID-19 X-ray dataset. Accordingly, these differences exemplify the fact that using pretrained models would make the results of our experiments less general and more data-dependent.”

We thank the reviewer very much for her/his comment, which prompted another improvement of our manuscript.

[2] Dosovitskiy, A. et al. An image is worth 16x16 words: Transformers for image recognition at scale (2020). 2010.11929.

[3] Vaswani, A. et al. Attention is all you need (2017). 1706.03762.

[4] DeGrave, A.J., Janizek, J.D. & Lee, S.I. AI for radiographic COVID-19 detection selects shortcuts over signal. Nat Mach Intell 3, 610–619 (2021). <https://doi.org/10.1038/s42256-021-00338-7>

[5] Bassi, P. R. A. S. & Attux, R. A deep convolutional neural network for covid-19 detection using chest x-rays. Res. on Biomed. Eng. DOI: 10.1007/s42600-021-00132-9 (2021)

[6] Khosla, A., Jayadevaprakash, N., Yao, B. & Fei-Fei, L. Novel dataset for fine-grained image categorization. In First Workshop on Fine-Grained Visual Categorization, IEEE Conference on Computer Vision and Pattern Recognition (Colorado Springs, CO, 2011).

3- *“How does the hyper-parameter P in L_{vis} vary across different datasets, especially considering that some datasets don't have a shortcut, while others have a strong one? The author noted in line 1130 that it “does not seem to strongly affect model performance.” Is there any quantitative analysis to support this assertion?”*

We thank the reviewer for her/his comment. The assertion in line 1130 was too vague, and we have substituted it with a more detailed explanation. In the previous version of the manuscript, we briefly introduced the hyper-parameter tuning procedure in Section 4.3 (which contained line 1130), but we described it in detail only in Section 4.12. Improving objectiveness, we have replaced the sentence the

reviewer mentioned by referencing (lines 395 and 396 of the main manuscript) a new Section, Supplementary Note 6.3, where we explain our hyper-parameter tuning procedure in detail and present the quantitative analysis the reviewer requested. The P and d parameters for all datasets are: P=0.7 and d=0.9 in COVID-19 and tuberculosis detection, P=0.7 and d=1 in dog breed classification, P=0.5 and d=0.996 in facial attribute estimation (parameters stated in Supplementary Note 6.2), and P=0.7 and d=1 in CheXPert (Supplementary Note 7.2). We explain below the content of Supplementary Note 6.3.

In summary, we tuned the ISNet loss P and d hyper-parameters using the experiments with synthetic background bias. We trained on the datasets with the artificial bias and analyzed performance (macro-average F1-Score) in a validation dataset without the synthetic bias. Thus, we tuned the hyper-parameters to maximize the validation dataset's performance while minimizing the performance gap between this validation set and its artificially biased version (containing the same bias as the training data). To find ideal hyper-parameters we resorted to a grid search, which considered the d values of 0.9, 0.996, and 1, and P of 0.2, 0.4, 0.6 and 0.8. Afterward, we employed a finer search for P around the best setting obtained in the grid search. The finer stage kept d constant and tried adding and subtracting 0.1 from P. E.g., if the grid search result was d=0.9, P=0.6, we also tried d=0.9, P=0.7, and d=0.9, P=0.5. When two hyper-parameter settings resulted in very similar validation maF1 (strong confidence interval overlap), we preferred the one with higher P and lower d (prioritizing the higher P). Thus, our search selects the parameters that provide the highest background bias resistance but preserve performance and training stability.

To address the reviewer's second question, we removed the vague sentence "does not seem to strongly affect model performance", replacing it by a more specific affirmation (Supplementary Note 6.3) and a quantitative example. We copied it below:

"After the two stages, further finer searches did not significantly improve performance. For example, in dog breed classification, the two-stage search led to P=0.7 and d=1, which resulted in average F1-Scores (with standard deviation) of 0.548 +/-0.035 in the biased test, 0.553 +/-0.035 in the standard test, and 0.548 +/-0.035 in the deceiving bias test (main article Table 1). Meanwhile, an ISNet with P=0.65 and d=1 obtained maF1 of 0.542 +/-0.034, 0.542 +/-0.034, and 0.537 +/-0.034 in the biased, standard, and deceiving tests, respectively, and the network with P=0.75 and d=1 got 0.539 +/-0.034 maF1 in the three testing scenarios. Accordingly, the results with P=0.65, P=0.7, and P=0.75 were very similar, and no model presented a significant advantage over the others."

Moreover, we created a plot (Supplementary Figure 6, in Supplementary Note 6.3) to analyze the influence of P on the ISNet performance quantitatively. We trained 10 ISNets, with diverse values of P (from 0 to 0.9), in the synthetically biased Stanford Dogs dataset. Supplementary Figure 6 shows the 9 models' test performances. For each network, we plot the average F1-Scores in the three diverse testing datasets (the standard test, which has no synthetic background bias, the biased test, which has the same background bias found in the training images, and the deceiving bias test, which has background bias designed to fool the neural network). The analysis of the figure is in Supplementary Note 6.3, and we copy it here for the reviewer's convenience:

“Supplementary Figure 6 exemplifies how test performances change with P , considering training on the synthetically biased Stanford Dogs dataset, $d=1$, and the training scheme described in Supplementary Note 6.2. P changes in 0.1 increments, from 0 to 0.9. Before $P=0.3$, there is considerable variability in the standard test mAF1 for adjacent data points (representing $P=0$, $P=0.1$, and $P=0.2$), a large gap between the performances on the diverse testing scenarios (standard, biased, and deceiving bias), and low standard and deceiving bias mAF1. These results indicate that, while shortcut learning is still not effectively hindered, the synthetic background bias influences the DNN, and its capacity to analyze the foreground features is unpredictable and subpar. At about $P=0.3$, the gap between the three testing scenarios closed, and the deceiving and standard tests' mAF1 significantly increased, indicating a minimization of shortcut learning. From $P=0.3$ to $P=0.7$, performances are stable, with confidence intervals' overlap and small variation with P . They begin to drop at around $P=0.8$. All networks were trained for the same time, 200 epochs. Thus, the final performance fall reflects that very high P makes training slower.”

Now, we address the reviewer's question about how P changes when datasets have strong and weak shortcuts. We tuned P (and d) using the datasets with synthetic background bias (Table 1), and we used the same hyperparameters in the corresponding datasets without artificial bias. The strategy may seem aggressive, producing high P in datasets with weak or no background bias. However, with quantitative examples, we show that the technique does not cause significant performance reduction. We have presented and justified this hyper-parameter tuning strategy in Supplementary Note 6.3, and we copied the analysis below for the reviewer's convenience.

“In the experiments without synthetic background bias, we employed the same hyper-parameters that we tuned using the artificially biased training data. The idea behind this decision is that hyper-parameters tuned to hinder background attention in a dataset with synthetic background bias should also hinder background focus in the non-synthetically biased version of the same dataset because its background bias is less extreme. Moreover, due to the similarity between the two tasks, we used the hyper-parameters tuned for COVID-19 detection in the tuberculosis detection application. If one does not want to resort to synthetic background bias to define hyper-parameters, P and d can also be tuned to maximize performance on an out-of-distribution validation dataset. This procedure may result in less aggressive hyper-parameter settings (e.g., lower P in the ISNet) since hyper-parameters would not be tuned in a database with extreme synthetic background bias. However, the alternative procedure requires an o.o.d. validation dataset. We tested other hyper-parameter settings using a grid search in the applications without synthetic bias, applying the previously explained two-stages search protocol. Still, they could not significantly improve generalization, confirming the adequacy of our hyper-parameter tuning strategy. As an illustration, we would expect the tuning strategy to overestimate P in datasets without background bias. Supplementary Note 7 shows applications where quantitative results indicate weak or no background bias (facial attribute estimation without synthetic bias and CheXPert classification). The ISNets trained in such applications had their hyper-parameters tuned on synthetically biased versions of the datasets using the procedure described in this section. The resulting parameters were $P=0.7$ and $d=1$ in CheXPert and $P=0.5$ and $d=0.996$ in facial attribute estimation. In both cases, the ISNet's average test performance had confidence interval overlaps with a standard DenseNet121 (Supplementary Tables 1 and 2), equivalent to an ISNet with $P=0$. Thus, the performance gain obtained by reducing P in the dataset without strong background bias was not large.”

We thank the reviewer again for her/his comment. It allowed us to include a new section (Supplementary Note 6.3), where we explained the hyper-parameter tuning procedure more clearly and objectively, providing quantitative analyses.

4- *“How is the $\text{abs}(H_{bk})$ backpropagated given that $\text{abs}()$ is non-differentiable? In your code, it's written as $\text{heatmap}=\text{torch.abs}(\text{heatmap})$.”*

It is possible that in previous versions of PyTorch the function $\text{torch.abs}()$ was non-differentiable; we are not sure. But we guarantee that it is differentiable in current PyTorch versions (we are using 1.11.0, as we now state in the GitHub code page, <https://github.com/PedroRASB/ISNet>, and in Supplementary Note 6.5).

From a theoretical point of view, the absolute value function is differentiable everywhere except when its input is 0. PyTorch considers its derivative 0 at this specific point, as it does for the ReLU function. The $\text{abs}()$ function is commonly used and differentiated in neural network training. For example, we may also backpropagate through it in L1 regularization, also known as Lasso regression. In summary, the ISNet loss is differentiable and PyTorch automatically backpropagates through it. We have included this information in line 397 of the main manuscript, in response to the reviewer's comment.

Moreover, we have written a simple Python script to demonstrate that $\text{torch.abs}()$ is differentiable:

```
import torch
x=torch.tensor(10.0,requires_grad=True)
y=torch.abs(x)
y.backward()
print('The derivative of abs(10) is:', x.grad.item())
x=torch.tensor(-10.0,requires_grad=True)
y=torch.abs(x)
y.backward()
print('The derivative of abs(-10) is:', x.grad.item())
x=torch.tensor(0.0,requires_grad=True)
y=torch.abs(x)
y.backward()
print('The derivative of abs(0) is:', x.grad.item())
```

The script outputs the expected derivatives:

```
The derivative of abs(10) is: 1.0
The derivative of abs(-10) is: -1.0
The derivative of abs(0) is: 0.0
```

We thank the reviewer again for the comments, which helped us improve our manuscript, and for her/his time and attention during the revision rounds.